# LAST-ITERATE CONVERGENCE OF SMOOTH REGRET MATCHING$^+$ VARIANTS IN LEARNING NASH EQUILIBRIA

## ABSTRACT

Regret Matching$^+$ (RM$^+$) variants have been widely developed to superhuman Poker AIs, yet few studies investigate their last-iterate convergence. Their last-iterate convergence has been demonstrated only for games with strong monotonicity or two-player zero-sum matrix games. A primary obstacle in proving the last-iterate convergence for these algorithms is that their feedback is not the loss gradient of the vanilla games. This deviation results in the absence of crucial properties, *e.g.*, monotonicity or the weak Minty variation inequality (MVI), which are pivotal for establishing the last-iterate convergence. To address the absence of these properties, we propose a remarkably succinct yet novel proof paradigm that consists of: (i) recovering these key properties through the equivalence between RM$^+$ and Online Mirror Descent (OMD), and (ii) measuring the the distance to Nash equilibrium (NE) via the tangent residual to show this distance is related to the distance between accumulated regrets. To show the practical applicability of our proof paradigm, we use it to prove the last-iterate convergence of two existing smooth RM$^+$ variants, Smooth Extra-gradient RM$^+$ (SExRM$^+$) and Smooth Predictive RM$^+$ (SPRM$^+$). We show that they achieve last-iterate convergence in learning an NE of games satisfying monotonicity, a weaker condition than the one used in existing proofs for both variants. Then, inspired by our proof paradigm, we propose Smooth Optimistic Gradient RM$^+$ (SOGRM$^+$). We show that SOGRM$^+$ achieves last-iterate convergence in learning an NE of games satisfying the weak MVI, the weakest condition in all known proofs for RM$^+$ variants. The experimental results show that SOGRM$^+$ significantly outperforms other algorithms.

## 1 INTRODUCTION

Nash Equilibrium (NE) is a fundamental concept in the field of game theory. Recent advancements in superhuman game AI, are largely attributed to NE learning (Moravčík et al., 2017; Brown & Sandholm, 2018; 2019; Pérolat et al., 2022). Despite these advancements, the most popular algorithms for learning an NE—no-regret algorithms, typically achieve only average-iterate convergence. Moreover, in two-player zero-sum matrix games, these algorithms are prone to divergence or cyclic behavior (Bailey & Piliouras, 2018; Mertikopoulos et al., 2018b; Pérolat et al., 2021). Average-iterate convergence requires strategy averaging. This averaging poses significant challenges in large-scale games where function approximation is used to represent the strategy since a new function has to be trained to represent the average strategy (Liu et al., 2023).

To address the challenges related to averaging, numerous studies consider the last-iterate convergence, ensuring iterates converge to NE (Mertikopoulos et al., 2018a; Daskalakis & Panageas, 2019; Tatarenko & Kamgarpour, 2020; Wei et al., 2021; Lee et al., 2021; Cen et al., 2021; Liu et al., 2023; Sokota et al., 2023; Abe et al., 2022a;b; 2023; Pérolat et al., 2021; 2022; Cai & Zheng, 2023). These algorithms are based on Online Mirror Descent (OMD) or Follow the Regularized Leader (FTRL). Despite their theoretical appeal, Regret Matching$^+$ (RM$^+$) variants (Bowling et al., 2015; Farina et al., 2021; 2023), are more commonly utilized in solving real-world games. Precisely, they are widely used in superhuman Poker AIs (Bowling et al., 2015; Moravčík et al., 2017; Brown & Sandholm, 2018). The key distinction between RM$^+$ variants and FTRL/OMD based algorithms is that RM$^+$ variants update within the (subset of the) non-negative orthant, whereas FTRL/OMD based algorithms update within the original strategy space of the game.

Table 1: Comparisons between the last-iterate convergence results of this paper and previous studies about RM$^+$ variants. "2p0s Games", "SM", "SN", and "RS" refer to two-player zero-sum matrix games, strong monotonicity, strict NE, and restarting (Cai et al., 2023), respectively. Games with monotonicity cover games with strong monotonicity and two-player zero-sum matrix games. Games with the Weak MVI is a super set of games with monotonicity. Notably, the convergence of " SExRM$^+$ & SPRM$^+$" in Cai et al. (2023) is the convergences to a point of the set of NE, which is stronger than our convergence concept that ensures the iterates converge to the set of NE (see details in Section 2.1).

| | Algorithm | Games with SM | 2p0s Games with SN | 2p0s Games | Games with Monotonicity | Games with Weak MVI |
|---|---|:---:|:---:|:---:|:---:|:---:|
| Meng et al. (2023) | RM$^+$ | ✓ | | | | |
| Cai et al. (2023) | RM$^+$ | | ✓ | | | |
| | SExRM$^+$ & SPRM$^+$ | | ✓ | ✓ | | |
| | RS-SExRM$^+$ & RS-SPRM$^+$ | | ✓ | ✓ | | |
| This paper | SExRM$^+$ & SPRM$^+$ | ✓ | ✓ | ✓ | ✓ | |
| | SOGRM$^+$ | ✓ | ✓ | ✓ | ✓ | ✓ |

Unfortunately, few studies investigate the last-iterate convergence of RM$^+$ variants. To date, the only known results on the last-iterate convergence of RM$^+$ variants are confined to a specific game with strong monotonicity (Meng et al., 2023) or two-player zero-sum matrix games (Cai et al., 2023). In contrast, the OMD/FTRL based algorithms achieve last-iterate convergence in a broader class of games, those that satisfy monotonicity. These games are also called as monotone games (Cai et al., 2022b; Cai & Zheng, 2023; Abe et al., 2023; Pérolat et al., 2021; 2022). They cover several common game types, such as two-player zero-sum matrix games and convex-concave games, along with significant applications like the training of Large Language Models (LLM) (Munos et al., 2023). Moreover, recent studies show that OMD/FTRL based algorithms even achieve last-iterate convergence in learning an NE of games satisfying the weak Minty variation inequality (MVI)) (Cai & Zheng, 2022; Diakonikolas, 2020; Pethick et al., 2023). Weak MVI is weaker and covers more games than monotonicity. It includes applications like Generative Adversarial Networks (GAN) (Cai & Zheng, 2022). Therefore, a key question is:

*Do RM$^+$ variants achieve last-iterate convergence in*
*learning an NE of games satisfying monotonicity or even only the weak MVI?*

Compared to traditional no-regret algorithms, *e.g.*, FTRL/OMD based algorithms, the primary challenge in proving the last-iterate convergence of RM$^+$ variants is that their feedback is not the loss gradients of the vanilla games. This deviation results in the absence of crucial properties, *e.g.*, monotonicity or weak MVI, which are pivotal for establishing the last-iterate convergence.

**Contributions.** (i) To address the absence of crucial properties, *e.g.*, monotonicity or weak MVI, we introduce a novel proof paradigm. Firstly, it recovers these properties by leveraging the equivalence between RM$^+$ and OMD in Liu et al. (2021). Secondly, it measures the distance of RM$^+$ variants to NE via the tangent residual (Cai et al., 2022b) to show that this distance is related to the distance between accumulated regrets. Specifically, in RM$^+$ variants, the feedback does not exhibit monotonicity or weak MVI. However, in their OMD equivalents, the feedback is the loss gradient of the vanilla games, which satisfies monotonicity or weak MVI. Then, We establish the last-iterate convergence of RM$^+$ variants by demonstrating that the distance between accumulated regrets, rather than the strategies in OMD based algorithms, converges to 0. This convergence occurs since, in RM$^+$ variants, the tangent residual converges 0 as this distance converges to 0. (ii) To show the practical applicability of our proof paradigm, we utilize this paradigm to establish that two existing smooth RM$^+$ variants (Farina et al., 2023), Smooth Extra-gradient RM$^+$ (SExRM$^+$) and Smooth Predictive RM$^+$ (SPRM$^+$), achieve last-iterate convergence in learning an NE of games satisfying monotonicity. (iii) Inspired by our proof paradigm, we propose Smooth Optimistic Gradient RM$^+$ (SOGRM$^+$), which combines Optimistic Gradient (OG) (Cai & Zheng, 2022) and smooth RM$^+$ variants. SOGRM$^+$ achieves last-iterate convergence in games satisfying the weak MVI. (iv) Experimental results show that SOGRM$^+$ significantly outperforms other algorithms. (v) Our proof paradigm yields explicit best-iterate convergence rates for SExRM$^+$, SPRM$^+$, and SOGRM$^+$ without any modifications.

**Discussions.** Table 1 shows the comparison between our work and the two most relevant literature (Meng et al., 2023; Cai et al., 2023). (i) Our proof diverges significantly from theirs as they either analyze the dynamics of limit points (Cai et al., 2023) or use the strongly monotonicity (Meng et al., 2023). (ii) The last-iterate convergence results of Cai et al. (2023) and Meng et al. (2023) cannot be

extended to games satisfying monotonicity (let alone the weak MVI). The reason is that their results need more assumptions than monotonicity, *e.g.*, the existence of the strict NE, the interchangeability of NE, the Saddle-Point Metric Subregularity (Cai et al., 2023), or even strong monotonicity (Meng et al., 2023). (iii) Our proof paradigm implies that existing last-iterate convergence results of OMD based algorithms can be applied to RM$^+$ variants. In contrast, Cai et al. (2023)'s proof cannot achieve this goal as their motivation is that the feedback of RM$^+$ variants only satisfies MVI (which is weaker than monotonicity while stronger than weak MVI, and defined in Section 2.1) even when the loss gradient of vanilla games satisfies monotonicity. (iv) Cai et al. (2023)[1] have to use another approach to prove the best-iterate convergence while we employ the same proof paradigm. (v) The best-iterate convergence results of Cai et al. (2023) only hold in two-player zero-sum matrix games as their results depend on the definition of the duality gap of these games. In contrast, our best-iterate convergence results hold in all games satisfying monotonicity or even only the weak MVI.

**Technical Novelty.** We develop a remarkably succinct yet novel proof paradigm via two techniques: the equivalence between RM$^+$ and OMD in Liu et al. (2021) and the tangent residual (Cai et al., 2022b). These techniques have been overlooked in previous works about the last-iterate convergence of RM$^+$ variants (Meng et al., 2023; Cai et al., 2023). To the best of our knowledge, neither of these techniques was used alone to prove the last-iterate convergence of RM+ variants. We combine and extend both techniques, but this process is not straightforward. For example, the proof for SOGRM$^+$ requires additional techniques, *i.e.*, transforming variables via the definition of the inner product to use the weak MVI and tangent residual, rather than directly using equalities as in OG.

## 2 PRELIMINARIES

### 2.1 SMOOTH GAMES AND TANGENT RESIDUAL

**Smooth games.** In this paper, we consider smooth games whose strategy space is simplex. We use $\boldsymbol{x}_i \in \boldsymbol{\mathcal{X}}_i$ to denote the strategy of player $i$ and $\boldsymbol{x} = \{\boldsymbol{x}_i | i \in \mathcal{N}\}$ to represent the strategy profile, where $\boldsymbol{\mathcal{X}}_i$ is an $(|A_i| - 1)$-dimension simplex, $|A_i|$ is the dimension of $\boldsymbol{\mathcal{X}}_i$, and $\mathcal{N}$ is the set of players. The utility of player $i$ if all players follow strategy profile $\boldsymbol{x}$ is $-\infty < u_i(\boldsymbol{x}_i, \boldsymbol{x}_{-i}) < +\infty$, where $-i$ is the players other than $i$. For any $i$ and the fixed $\boldsymbol{x}_{-i} \in \boldsymbol{\mathcal{X}}_{-i}$, $u_i(\boldsymbol{x}_i, \boldsymbol{x}_{-i})$ is a concave function w.r.t. $\boldsymbol{x}_i \in \boldsymbol{\mathcal{X}}_i$. Also, $\boldsymbol{\ell}_i^{\boldsymbol{x}} = -\nabla_{\boldsymbol{x}_i} u_i(\boldsymbol{x}_i, \boldsymbol{x}_{-i})$ is loss gradient. In smooth games,

$$\|\boldsymbol{\ell}^{\boldsymbol{x}} - \boldsymbol{\ell}^{\boldsymbol{x}'}\|_2 \le L\|\boldsymbol{x} - \boldsymbol{x}'\|_2, \forall \boldsymbol{x}, \boldsymbol{x}' \in \boldsymbol{\mathcal{X}},$$

where $\boldsymbol{\ell}^{\boldsymbol{x}} = [\boldsymbol{\ell}_i^{\boldsymbol{x}} : i \in \mathcal{N}]$, and $L > 0$ is a constant. In addition, we assume $\|\boldsymbol{\ell}_i^{\boldsymbol{x}}\|_1 \le P$ for each player $i$ and strategy $\boldsymbol{x}$, where $P$ is a positive constant.

**Nash equilibrium (NE).** In NE, for any player, her strategy is the best-response to the strategies of others. The notation $\boldsymbol{\mathcal{X}}^*$ denotes the set of NE. As $u_i(\boldsymbol{x}_i, \boldsymbol{x}_{-i})$ is a concave function w.r.t. $\boldsymbol{x}_i \in \boldsymbol{\mathcal{X}}_i$, then $\forall \boldsymbol{x}^* \in \boldsymbol{\mathcal{X}}^*, \boldsymbol{x} \in \boldsymbol{\mathcal{X}}, \langle \boldsymbol{\ell}_i^{\boldsymbol{x}^*}, \boldsymbol{x}_i^* - \boldsymbol{x}_i \rangle \le 0$ (Facchinei, 2003).

**Monotonicity.** Smooth games with monotonicity is called smooth monotone games, which include many common and well-studied classes of games, such as two-player zero-sum matrix games and convex-concave games. The most important property of monotone games is monotonicity

$$\langle \boldsymbol{\ell}^{\boldsymbol{x}} - \boldsymbol{\ell}^{\boldsymbol{x}'}, \boldsymbol{x} - \boldsymbol{x}' \rangle \ge 0, \forall \boldsymbol{x}, \boldsymbol{x}' \in \boldsymbol{\mathcal{X}}.$$

Monotonicity is the most widely used assumption in existing works about the last-iterate convergence.

**Minty variation inequality (MVI).** From $\langle \boldsymbol{\ell}^{\boldsymbol{x}^*}, \boldsymbol{x}^* - \boldsymbol{x} \rangle \le 0$ and monotonicity, we get $\langle \boldsymbol{\ell}^{\boldsymbol{x}}, \boldsymbol{x} - \boldsymbol{x}^* \rangle \ge 0, \forall \boldsymbol{x} \in \boldsymbol{\mathcal{X}}, \exists \boldsymbol{x}^* \in \boldsymbol{\mathcal{X}}^*$, also called MVI. MVI is weaker than monotonicity as MVI holds if monotonicity holds, but not vice versa. We provide an example of games that satisfy the weak MVI but not monotonicity in Appendix G.

**Weak MVI.** Some recent works consider a weaker assumption than monotonicity and MVI called weak MVI (Cai & Zheng, 2022; Diakonikolas, 2020; Pethick et al., 2023; Cai et al., 2022a), which covers more game types. Formally, weak MVI with $\rho \le 0$ implies there exists $\boldsymbol{x}^* \in \boldsymbol{\mathcal{X}}^*$ that ensures

$$\langle \boldsymbol{\ell}^{\boldsymbol{x}} + \boldsymbol{z}, \boldsymbol{x} - \boldsymbol{x}^* \rangle \ge \rho \|\boldsymbol{\ell}^{\boldsymbol{x}} + \boldsymbol{z}\|_2^2, \forall \boldsymbol{z} \in \mathcal{N}_{\boldsymbol{\mathcal{X}}}(\boldsymbol{x}), \boldsymbol{x} \in \boldsymbol{\mathcal{X}},$$

where $\mathcal{N}_{\boldsymbol{\mathcal{X}}}(\boldsymbol{x}) = \{\boldsymbol{v} \in \mathbb{R}^{|\boldsymbol{\mathcal{X}}|} : \langle \boldsymbol{v}, \boldsymbol{x}' - \boldsymbol{x} \rangle \le 0, \forall \boldsymbol{x}' \in \boldsymbol{\mathcal{X}}\}$ is the normal cone of $\boldsymbol{x}$. If $\rho \to -\infty$, intuitively, any smooth games satisfy the weak MVI. In Appendix G, we provide a smooth game that satisfies the weak MVI with $0 > \rho > -\infty$ and does not satisfy the MVI. The relations between monotonicity, the MVI, and the weak MVI, is that monotonicity $\subseteq$ MVI $\subseteq$ weak MVI.

---

[1]Meng et al. (2023) do not investigate best-iterate convergence.

**Tangent residual.** To measure the distance from a strategy profile to the set of NE, we employ the tangent residual provided by Cai et al. (2022b). Formally, $\forall \boldsymbol{x} \in \boldsymbol{\mathcal{X}}$, its tangent residual is

$$r^{tan}(\boldsymbol{x}) = \min_{\boldsymbol{z} \in \mathcal{N}_{\boldsymbol{\mathcal{X}}}(\boldsymbol{x})} \|\boldsymbol{\ell}^{\boldsymbol{x}} + \boldsymbol{z}\|_2.$$

If $r^{tan}(\boldsymbol{x}) = 0$, then $\boldsymbol{x}$ is an NE in smooth games. Also, if $\boldsymbol{x}$ is an NE in smooth games, $r^{tan}(\boldsymbol{x}) = 0$.

**Last-iterate convergence.** In this paper, this convergence refers to the behavior where the sequence of strategy profiles converges to the set of NEs. As previously discussed, if $\lim_{t \to \infty} r^{tan}(\boldsymbol{x}^t) = 0$, then $\boldsymbol{x}^t$ is an NE, which implies that $\boldsymbol{x}^t$ converges to the set of NE. However, it is important to note that Cai et al. (2023) define a stronger concept of convergence, known as last-iterate convergence of the iterates. In contrast to the last-iterate convergence discussed in our paper, their definition implies that $\boldsymbol{x}^t$ converges to a specific point within the set of NE. Our results do not pertain to the last-iterate convergence of the iterates, whereas the results regarding SExRM$^+$ and SPRM$^+$ in Cai et al. (2023) do.

## 2.2 REGRET MATCHING$^+$

**Online convex optimization.** Each player $i$ selects a decision $\boldsymbol{x}_i^t$ via the feedback in this framework. Such feedback is the loss gradient $\boldsymbol{\ell}_i^{t-1} = \boldsymbol{\ell}_i^{\boldsymbol{x}^{t-1}}$ in solving smooth games. No-regret algorithms are the algorithms, which ensures the regret $R_i^T(\boldsymbol{x}) = \max_{\boldsymbol{x}_i \in \boldsymbol{\mathcal{X}}_i} \sum_{t=1}^T \langle \boldsymbol{\ell}_i^t, \boldsymbol{x}_i^t - \boldsymbol{x}_i \rangle$ to grow sublinearly, where $\boldsymbol{x}_i^t$ is the decision at iteration $t$.

**Online mirror descent (OMD).** OMD is a traditional no-regret algorithm (Nemirovskij & Yudin, 1983). Let $q_i^t(\cdot) : \boldsymbol{\mathcal{X}}_i \to \mathbb{R}, \forall t \geq 0$, OMD generates the decisions via the prox-mapping operator

$$\boldsymbol{x}_i^{t+1} \in \arg\min_{\boldsymbol{x}_i \in \boldsymbol{\mathcal{X}}_i} \{ \langle \boldsymbol{\ell}_i^t, \boldsymbol{x}_i \rangle + q_i^t(\boldsymbol{x}_i) + D_{q_i^{0:t-1}}(\boldsymbol{x}_i, \boldsymbol{x}_i^t) \},$$

where $q_i^{0:t-1}(\cdot) = q_i^0(\cdot) + q_i^1(\cdot) + \cdots + q_i^{t-1}(\cdot)$, and $D_{q_i^{0:t-1}}(\boldsymbol{x}, \boldsymbol{y}) = q_i^{0:t-1}(\boldsymbol{x}) - q_i^{0:t-1}(\boldsymbol{y}) - \langle \nabla q_i^{0:t-1}(\boldsymbol{y}), \boldsymbol{x} - \boldsymbol{y} \rangle$ is the Bregman divergence associated with $q_i^{0:t-1}(\cdot)$. Notably, we employ the definition of OMD in Joulani et al. (2017) and Liu et al. (2021), which represents a generalization of the standard OMD, to demonstrate the equivalence between RM$^+$ and OMD as proposed by Liu et al. (2021). To recover the standard OMD, we can set $q_i^0 = \phi(\cdot)/\eta$ and $q_i^t = 0$ for all $t \geq 1$, where $\phi(\cdot)$ is a 1-strongly convex regularizer with respect to some norm in the decision space $\boldsymbol{\mathcal{X}}$, and $\eta > 0$.

**Blackwell approachability framework.** RM$^+$ variants are from this framework whose core insight lies in reframing the problem of regret minimization within $\boldsymbol{\mathcal{X}}_i$ as regret minimization within $\text{cone}(\boldsymbol{\mathcal{X}}_i)$ (Abernethy et al., 2011). Specifically, a regret minimization algorithm is instantiated in $\text{cone}(\boldsymbol{\mathcal{X}}_i)$, where its output at iteration $t$ is $\boldsymbol{\theta}_i^t$. This corresponds to the strategy $\boldsymbol{x}_i^t = \boldsymbol{\theta}_i^t / \langle \boldsymbol{\theta}_i^t, \mathbf{1} \rangle$ within $\boldsymbol{\mathcal{X}}_i$. Given the loss $\boldsymbol{\ell}_i^t$ at iteration $t$, the algorithm observes the transformed loss $\boldsymbol{F}_i(\boldsymbol{\theta}^t) = \langle \boldsymbol{\ell}_i^t, \boldsymbol{x}_i^t \rangle \mathbf{1} - \boldsymbol{\ell}_i^t$ and subsequently generates $\boldsymbol{\theta}_i^{t+1}$.

**Regret Matching$^+$ (RM$^+$)** (Bowling et al., 2015). RM$^+$ keeps track of the accumulated regret $\boldsymbol{\theta}_i^t$. In RM$^+$, the strategy $\boldsymbol{x}_i^t$ at each iteration $t$ is denoted by $\boldsymbol{x}_i^t = \boldsymbol{\theta}_i^t / \|\boldsymbol{\theta}_i^t\|_1$. It updates its accumulated regret $\boldsymbol{\theta}_i^t$ via the regret matching$^+$ operator (Bowling et al., 2015)

$$\boldsymbol{\theta}_i^{t+1} = [\boldsymbol{\theta}_i^t + \eta \boldsymbol{F}_i^t(\boldsymbol{\theta}^t)]^+,$$

where $\eta > 0$ is the step-size, $\boldsymbol{F}_i(\boldsymbol{\theta}^t) = \langle \boldsymbol{\ell}_i^t, \boldsymbol{x}_i^t \rangle \mathbf{1} - \boldsymbol{\ell}_i^t$ ( $\{\boldsymbol{\theta}^t = [\boldsymbol{\theta}_i^t : i \in \mathcal{N}]$). As analyzed in Farina et al. (2021), RM$^+$ is closely connected to an OMD instance which updates in $\text{cone}(\boldsymbol{\mathcal{X}}_i)$ and faces a sequence of loss $[\boldsymbol{F}_i^t(\boldsymbol{\theta}^t)]_{t \geq 1}$. Formally, RM$^+$ can be rewritten as

$$\boldsymbol{\theta}_i^{t+1} \in \arg\min_{\boldsymbol{\theta}_i \in \mathbb{R}_{\geq 0}^{|A_i|}} \{ \langle -\boldsymbol{F}_i(\boldsymbol{\theta}^t), \boldsymbol{\theta}_i \rangle + \frac{1}{\eta} D_\psi(\boldsymbol{\theta}_i, \boldsymbol{\theta}_i^t) \}, \ \boldsymbol{x}_i^{t+1} = \frac{\boldsymbol{\theta}_i^{t+1}}{\|\boldsymbol{\theta}_i^{t+1}\|_1},$$

where $\psi(\boldsymbol{a}) = \|\boldsymbol{a}\|_2^2 / 2$ is the quadratic regularizer, and $\mathbb{R}_{\geq 0}^d = \{\boldsymbol{y} | \boldsymbol{y} \in \mathbb{R}^d, \boldsymbol{y} \geq \mathbf{0}\}$.

**Equivalence between RM$^+$ and OMD in Liu et al. (2021).** The analysis in Farina et al. (2021) is the main approach for proving the last-iterate convergence of RM$^+$ variants (Meng et al., 2023; Cai et al., 2023). However, in this analysis, the feedback ($-\boldsymbol{F}(\boldsymbol{\theta}^t) = [-\boldsymbol{F}_i(\boldsymbol{\theta}^t) : i \in \mathcal{N}]$) does not enjoy monotonicity or the weak MVI, crucial for proving the last-iterate convergence in existing works.

To recover monotonicity or the weak MVI, we use the equivalence provided by Liu et al. (2021) to rewrite RM$^+$ as

$$\boldsymbol{x}_i^{t+1} \in \underset{\boldsymbol{x}_i \in \mathcal{X}_i}{\arg\min}\{\langle \boldsymbol{\ell}_i^t, \boldsymbol{x}_i \rangle + q_i^t(\boldsymbol{x}_i) + D_{q_i^{0:t-1}}(\boldsymbol{x}_i, \boldsymbol{x}_i^t)\}, q_i^{0:t}(\boldsymbol{x}_i) = \frac{\|\boldsymbol{\theta}_i^{t+1}\|_1}{\eta}\psi(\boldsymbol{x}_i). \tag{1}$$

The feedback ($\boldsymbol{\ell}^{t+1} = [\boldsymbol{\ell}_i^{t+1} : i \in \mathcal{N}]$) is the loss gradient of the vanilla game, which enjoys monotonicity or the weak MVI. Therefore, we recover monotonicity or the weak MVI via this equivalence. Notably, this equivalence indicates that, given $\boldsymbol{\theta}_i^{t+1}$ at iteration $t$, the update of RM$^+$ can be expressed in the form of Eq. (1). However, utilizing Eq. (1) to derive $\boldsymbol{\theta}_i^{t+1}$ is impossible.

## 2.3 SMOOTH REGRET MATCHING$^+$ VARIANTS

Smooth RM$^+$ variants (Farina et al., 2023) are designed to address the instability of Predictive RM$^+$ (PRM$^+$) (Farina et al., 2021). To do that, they enable the decision $\boldsymbol{\theta}_i^t$ in $\mathbb{R}_{\geq 1}^d$ instead of $\mathbb{R}_{\geq 0}^d$ in other RM$^+$ variants to obtain the smoothness of $\boldsymbol{F}_i(\boldsymbol{\theta}^t)$, where $\mathbb{R}_{\geq 1}^d = \{\boldsymbol{y}|\boldsymbol{y} \in \mathbb{R}^d, \boldsymbol{y} \geq \boldsymbol{0}, \|\boldsymbol{y}\|_1 \geq 1\}$. We consider two existing smooth RM$^+$ variants, Smooth Extra-gradient RM$^+$ (SExRM$^+$) and Smooth Predictive RM$^+$ (SPRM$^+$). SExRM$^+$ and SPRM$^+$ are respectively related to instances of two OMD variants, Optimistic Gradient Descent Ascent (OGDA) (Wei et al., 2021) and Extra-Gradient (EG) (Korpelevich, 1976), which updates in $\mathbb{R}_{\geq 1}^d$, the subset of cone($\mathcal{X}_i$). The update rule of SExRM$^+$ is

$$\boldsymbol{\theta}_i^{t+\frac{1}{2}} \in \underset{\boldsymbol{\theta}_i \in \mathbb{R}_{\geq 1}^{|A_i|}}{\arg\min}\{\langle -\boldsymbol{F}_i(\boldsymbol{\theta}^t), \boldsymbol{\theta}_i \rangle + \frac{1}{\eta}D_\psi(\boldsymbol{\theta}_i, \boldsymbol{\theta}_i^t)\}, \ \boldsymbol{x}_i^{t+\frac{1}{2}} = \frac{\boldsymbol{\theta}_i^{t+\frac{1}{2}}}{\|\boldsymbol{\theta}_i^{t+\frac{1}{2}}\|_1},$$

$$\boldsymbol{\theta}_i^{t+1} \in \underset{\boldsymbol{\theta}_i \in \mathbb{R}_{\geq 1}^{|A_i|}}{\arg\min}\{\langle -\boldsymbol{F}_i(\boldsymbol{\theta}^{t+\frac{1}{2}}), \boldsymbol{\theta}_i \rangle + \frac{1}{\eta}D_\psi(\boldsymbol{\theta}_i, \boldsymbol{\theta}_i^t)\}, \ \boldsymbol{x}_i^{t+1} = \frac{\boldsymbol{\theta}_i^{t+1}}{\|\boldsymbol{\theta}_i^{t+1}\|_1}, \tag{2}$$

and the update rule of SPRM$^+$ is

$$\boldsymbol{\theta}_i^{t+\frac{1}{2}} \in \underset{\boldsymbol{\theta}_i \in \mathbb{R}_{\geq 1}^{|A_i|}}{\arg\min}\{\langle -\boldsymbol{F}_i(\boldsymbol{\theta}^{t-\frac{1}{2}}), \boldsymbol{\theta}_i \rangle + \frac{1}{\eta}D_\psi(\boldsymbol{\theta}_i, \boldsymbol{\theta}_i^t)\}, \ \boldsymbol{x}_i^{t+\frac{1}{2}} = \frac{\boldsymbol{\theta}_i^{t+\frac{1}{2}}}{\|\boldsymbol{\theta}_i^{t+\frac{1}{2}}\|_1},$$

$$\boldsymbol{\theta}_i^{t+1} \in \underset{\boldsymbol{\theta}_i \in \mathbb{R}_{\geq 1}^{|A_i|}}{\arg\min}\{\langle -\boldsymbol{F}_i(\boldsymbol{\theta}^{t+\frac{1}{2}}), \boldsymbol{\theta}_i \rangle + \frac{1}{\eta}D_\psi(\boldsymbol{\theta}_i, \boldsymbol{\theta}_i^t)\}, \ \boldsymbol{x}_i^{t+1} = \frac{\boldsymbol{\theta}_i^{t+1}}{\|\boldsymbol{\theta}_i^{t+1}\|_1}, \tag{3}$$

where $\eta > 0$ and $\psi(\cdot)$ is the quadratic regularizer.

## 3 OUR PROOF PARADIGM

We now introduce our proof paradigm that includes: (i) recovering monotonicity or the weak MVI by leveraging the equivalence between RM$^+$ and OMD proposed by Liu et al. (2021), and (ii) measuring the distance of RM$^+$ variants to NE via the tangent residual to show that this distance is related to the distance between accumulated regrets rather than the strategies in OMD based algorithms. We considers smooth RM$^+$ variants, since other RM$^+$ variants, *e.g.*, vanilla RM$^+$ and PRM$^+$, are experimentally shown to diverge in two-player zero-sum matrix games (Cai et al., 2023).

**Phase 1.** To recover monotonicity or the weak MVI of smooth RM$^+$ variants via the equivalence in Liu et al. (2021), we prove this equivalence holds between smooth RM$^+$ variants with OMD. To do that, it is sufficient to show the update rule in Eq. (4) can be written as the form in Eq. (5).

$$\boldsymbol{x}_i^{t_2} = \frac{\boldsymbol{\theta}_i^{t_2}}{\|\boldsymbol{\theta}_i^{t_2}\|_1}, \boldsymbol{\theta}_i^{t_2} \in \underset{\boldsymbol{\theta}_i \in \mathbb{R}_{\geq 1}^{|A_i|}}{\arg\min}\{\langle -\boldsymbol{F}_i(\boldsymbol{\theta}^{t_1}), \boldsymbol{\theta}_i \rangle + \frac{1}{\eta}D_\psi(\boldsymbol{\theta}_i, \boldsymbol{\theta}_i^{t_0})\}, \quad \boldsymbol{F}_i(\boldsymbol{\theta}^{t_1}) = \langle \frac{\boldsymbol{\theta}_i^{t_1}}{\|\boldsymbol{\theta}_i^{t_1}\|_1}, \boldsymbol{\ell}_i^{\boldsymbol{\theta}^{t_1}} \rangle \boldsymbol{1} - \boldsymbol{\ell}_i^{\boldsymbol{\theta}^{t_1}}, \tag{4}$$

$$\boldsymbol{x}_i^{t_2} \in \underset{\boldsymbol{x}_i \in \mathcal{X}_i}{\arg\min}\{\langle \boldsymbol{\ell}_i^{\boldsymbol{\theta}^{t_1}}, \boldsymbol{x}_i \rangle + f_i(\boldsymbol{x}_i) + D_{h_i}(\boldsymbol{x}_i, \boldsymbol{x}_i^{t_0})\}, h_i(\boldsymbol{x}_i) + f_i(\boldsymbol{x}_i) = \frac{\|\boldsymbol{\theta}_i^{t_2}\|_1}{\eta}\psi(\boldsymbol{x}_i), h_i(\boldsymbol{x}_i) = \frac{\|\boldsymbol{\theta}_i^{t_0}\|_1}{\eta}\psi(\boldsymbol{x}_i), \tag{5}$$

where $t_0, t_1, t_3$ refer to different iterations, $\eta > 0$, $\boldsymbol{x}_i^{t_0} = \boldsymbol{\theta}_i^{t_0}/\|\boldsymbol{\theta}_i^{t_0}\|_1$, $\boldsymbol{\theta}_i^{t_0}$ with $\boldsymbol{\theta}_i^{t_1} \in \mathbb{R}_{>0}^{|A_i|}$, $\boldsymbol{\ell}_i^{\boldsymbol{\theta}^{t_1}}$ is the loss gradient of player $i$ induced by $\boldsymbol{x}^{t_1} = [\boldsymbol{x}_i^{t_1} = \boldsymbol{\theta}_i^{t_1}/\|\boldsymbol{\theta}_i^{t_1}\|_1 : i \in \mathcal{N}]$, and $\psi(\cdot)$ is the quadratic regularizer. As shown in Section 3.1 of Liu et al. (2021), Eq. (5) can be written as

$$\boldsymbol{x}_i^{t_2} = \frac{[\boldsymbol{\theta}_i^{t_0} + \alpha\boldsymbol{1} - \eta\boldsymbol{\ell}_i^{\boldsymbol{\theta}^{t_1}}]_+}{\|\boldsymbol{\theta}_i^{t_2}\|_1},$$

where $\alpha$ is a unique constant to ensure $\|\boldsymbol{x}_i^{t_2}\|_1 = 1$ ($\alpha$ always exists). Then, considering Eq. (4), with the analysis in Section K of Farina et al. (2023), if

$$\|[\boldsymbol{\theta}_i^{t_0} + \eta\langle\frac{\boldsymbol{\theta}_i^{t_1}}{\|\boldsymbol{\theta}_i^{t_1}\|_1}, \boldsymbol{\ell}_i^{\boldsymbol{\theta}^{t_1}}\rangle\mathbf{1} - \eta\boldsymbol{\ell}_i^{\boldsymbol{\theta}^{t_1}}]_+\|_1 \geq 1,$$

$\boldsymbol{\theta}_i^{t_2}$ in Eq. (4) can be obtained via

$$\boldsymbol{\theta}_i^{t_2} = [\boldsymbol{\theta}_i^{t_0} + \eta\langle\frac{\boldsymbol{\theta}_i^{t_1}}{\|\boldsymbol{\theta}_i^{t_1}\|_1}, \boldsymbol{\ell}_i^{\boldsymbol{\theta}^{t_1}}\rangle\mathbf{1} - \eta\boldsymbol{\ell}_i^{\boldsymbol{\theta}^{t_1}}]_+.$$

Therefore, in this case, $\alpha = \eta\langle\frac{\boldsymbol{\theta}_i^{t_1}}{\|\boldsymbol{\theta}_i^{t_1}\|_1}, \boldsymbol{\ell}_i^{\boldsymbol{\theta}^{t_1}}\rangle$. Similarly, if

$$\|[\boldsymbol{\theta}_i^{t_0} + \eta\langle\frac{\boldsymbol{\theta}_i^{t_1}}{\|\boldsymbol{\theta}_i^{t_1}\|_1}, \boldsymbol{\ell}_i^{\boldsymbol{\theta}^{t_1}}\rangle\mathbf{1} - \eta\boldsymbol{\ell}_i^{\boldsymbol{\theta}^{t_1}}]_+\|_1 < 1,$$

$\boldsymbol{\theta}_i^{t_2}$ in Eq. (4) can be obtained via

$$\boldsymbol{\theta}_i^{t_2} = [\boldsymbol{\theta}_i^{t_0} + \eta\langle\frac{\boldsymbol{\theta}_i^{t_1}}{\|\boldsymbol{\theta}_i^{t_1}\|_1}, \boldsymbol{\ell}_i^{\boldsymbol{\theta}^{t_1}}\rangle\mathbf{1} - \eta\boldsymbol{\ell}_i^{\boldsymbol{\theta}^{t_1}} + \beta\mathbf{1}]_+,$$

where $\beta$ exists and is unique to ensure $\|\boldsymbol{\theta}_i^{t_2}\|_1 = 1$. Therefore, in this case, $\alpha = \eta\langle\frac{\boldsymbol{\theta}_i^{t_1}}{\|\boldsymbol{\theta}_i^{t_1}\|_1}, \boldsymbol{\ell}_i^{\boldsymbol{\theta}^{t_1}}\rangle + \beta$.

As $\alpha$ is unique, we have that

$$\alpha = \begin{cases} \eta\langle\frac{\boldsymbol{\theta}_i^{t_1}}{\|\boldsymbol{\theta}_i^{t_1}\|_1}, \boldsymbol{\ell}_i^{\boldsymbol{\theta}^{t_1}}\rangle, & \|[\boldsymbol{\theta}_i^{t_0} + \eta\langle\frac{\boldsymbol{\theta}_i^{t_1}}{\|\boldsymbol{\theta}_i^{t_1}\|_1}, \boldsymbol{\ell}_i^{\boldsymbol{\theta}^{t_1}}\rangle\mathbf{1} - \eta\boldsymbol{\ell}_i^{\boldsymbol{\theta}^{t_1}}]_+\|_1 \geq 1, \\ \eta\langle\frac{\boldsymbol{\theta}_i^{t_1}}{\|\boldsymbol{\theta}_i^{t_1}\|_1}, \boldsymbol{\ell}_i^{\boldsymbol{\theta}^{t_1}}\rangle + \beta, & \|[\boldsymbol{\theta}_i^{t_0} + \eta\langle\frac{\boldsymbol{\theta}_i^{t_1}}{\|\boldsymbol{\theta}_i^{t_1}\|_1}, \boldsymbol{\ell}_i^{\boldsymbol{\theta}^{t_1}}\rangle\mathbf{1} - \eta\boldsymbol{\ell}_i^{\boldsymbol{\theta}^{t_1}}]_+\|_1 < 1. \end{cases}$$

These complete the proof. Due to page limitations, a detailed proof is in Appendix A. This equivalence is the inherent property of smooth RM$^+$ variants and does not involve the game types. It also implies that smooth RM$^+$ variants can be represented by OMD based algorithms whose feedback is the loss gradient of vanilla games. We recover monotonicity or the weak MVI since the feedback in Eq. (5) is the loss gradient of the vanilla game.

**Phase 2.** As analyzed in Cai & Zheng (2022), if the loss gradient of the vanilla game enjoys monotonicity or the weak MVI, we can use the tangent residual to denote the distance to NE. Formally, from the first-order optimality of the prox-mapping operator in Eq. (5), we have

$$\langle\boldsymbol{\ell}_i^{\boldsymbol{\theta}^{t_1}} + \nabla_{\boldsymbol{x}_i^{t_2}}f_i(\boldsymbol{x}_i^{t_2}) + \nabla_{\boldsymbol{x}_i^{t_2}}D_{h_i}(\boldsymbol{x}_i^{t_2}, \boldsymbol{x}_i^{t_0}), \boldsymbol{x}_i - \boldsymbol{x}_i^{t_2}\rangle \geq 0$$

$$\Leftrightarrow \sum_{i\in\mathcal{N}}\langle\boldsymbol{\ell}_i^{\boldsymbol{\theta}^{t_1}} + \nabla_{\boldsymbol{x}_i^{t_2}}h_i(\boldsymbol{x}_i^{t_2}) + \nabla_{\boldsymbol{x}_i^{t_2}}f_i(\boldsymbol{x}_i^{t_2}) - \nabla_{\boldsymbol{x}_i^{t_0}}h_i(\boldsymbol{x}_i^{t_0}), \boldsymbol{x}_i - \boldsymbol{x}_i^{t_2}\rangle \geq 0$$

$$\Leftrightarrow \sum_{i\in\mathcal{N}}\langle\boldsymbol{\ell}_i^{\boldsymbol{\theta}^{t_1}} - \frac{\boldsymbol{\theta}_i^{t_0} - \boldsymbol{\theta}_i^{t_2}}{\eta}, \boldsymbol{x}_i - \boldsymbol{x}_i^{t_2}\rangle \geq 0 \Leftrightarrow -\boldsymbol{\ell}^{\boldsymbol{\theta}^{t_1}} + \frac{\boldsymbol{\theta}^{t_0} - \boldsymbol{\theta}^{t_2}}{\eta} \in \mathcal{N}_{\boldsymbol{\mathcal{X}}}(\boldsymbol{x}^{t_2}) \tag{6}$$

$$\Leftrightarrow r^{tan}(\boldsymbol{x}^{t_2}) \leq \|\boldsymbol{\ell}^{\boldsymbol{\theta}^{t_2}} - \boldsymbol{\ell}^{\boldsymbol{\theta}^{t_1}} + \frac{\boldsymbol{\theta}^{t_0} - \boldsymbol{\theta}^{t_2}}{\eta}\|_2,$$

where $\boldsymbol{x}_i^{t_1} = \boldsymbol{\theta}_i^{t_1}/\|\boldsymbol{\theta}_i^{t_1}\|_1$, where the third line comes from $\nabla_{\boldsymbol{x}_i^{t_2}}h_i(\boldsymbol{x}_i^{t_2}) + \nabla_{\boldsymbol{x}_i^{t_2}}f_i(\boldsymbol{x}_i^{t_2}) = \boldsymbol{\theta}_i^{t_2}/\eta$ and $\nabla_{\boldsymbol{x}_i^{t_0}}h_i(\boldsymbol{x}_i^{t_0}) = \boldsymbol{\theta}_i^{t_0}/\eta$, the last line is from the definition of the tangent residual. Therefore, if we can prove $\|\boldsymbol{\ell}^{\boldsymbol{\theta}^{t_2}} - \boldsymbol{\ell}^{\boldsymbol{\theta}^{t_1}}\|_2 \to 0$ and $\|\boldsymbol{\theta}^{t_2} - \boldsymbol{\theta}^{t_0}\|_2 \to 0$, we can get that $r^{tan}(\boldsymbol{x}^{t_2}) \to 0$, which implies $\boldsymbol{x}^{t_2}$ is an NE. In smooth RM$^+$ variants, we have that $\|\boldsymbol{\ell}^{\boldsymbol{\theta}^{t_2}} - \boldsymbol{\ell}^{\boldsymbol{\theta}^{t_1}}\|_2 \leq O(\|\boldsymbol{\theta}^{t_2} - \boldsymbol{\theta}^{t_1}\|_2)$ ($\|\boldsymbol{\ell}^{\boldsymbol{\theta}^{t_2}} - \boldsymbol{\ell}^{\boldsymbol{\theta}^{t_1}}\|_2 \leq O(\|\boldsymbol{\theta}^{t_2} - \boldsymbol{\theta}^{t_1}\|_2)$ does not hold in other RM$^+$ variants, which is the reason why we consider smooth RM$^+$ variants). Thus, in smooth RM$^+$ variants, if we can prove $\|\boldsymbol{\theta}^{t_2} - \boldsymbol{\theta}^{t_1}\|_2 \to 0$ and $\|\boldsymbol{\theta}^{t_2} - \boldsymbol{\theta}^{t_0}\|_2 \to 0$, we can get that $\boldsymbol{x}^{t_2}$ converges to NE.

# 4 APPLICATION OF OUR PROOF PARADIGM: CONVERGENCE RESULTS OF SExRM$^+$ AND SPRM$^+$

To show the practical applicability of our paradigm, we use it to prove the last-iterate and best-iterate convergence of two existing smooth RM$^+$ variants, *e.g.*, SExRM$^+$ and SPRM$^+$. Note that our convergence results in this section cover all games that existing works about the last-iterate convergence of RM$^+$ variants investigate.

**Theorem 4.1.** *SExRM$^+$ with $0 < \eta < \frac{1}{DL_u}$ or SPRM$^+$ with $0 < \eta < \frac{1}{8DL_u}$ achieves asymptotic last-iterate convergence and $O(\frac{1}{\sqrt{t}})$ best-iterate convergence rate in learning an NE of games satisfying monotonicity, where $D = \max_{i \in \mathcal{N}} |A_i|$ and $L_u = \sqrt{2P^2 + 4L^2}$. Specifically, if all players follow the update rule of SExRM$^+$ or SPRM$^+$, then $r^{tan}(\boldsymbol{x}^{t+\frac{1}{2}}) \to 0$ and $\min_{\tau \in [t]} r^{tan}(\boldsymbol{x}^{\tau+\frac{1}{2}}) \le O(\frac{1}{\sqrt{t}})$ as $t \to \infty$.*

To prove Theorem 4.1, we introduce the Theorem 4.2, Theorem 4.3, and Lemma 4.4 (the proof of Theorems 4.2 and 4.3 are in Appendix B and C, respectively).

**Theorem 4.2.** *SExRM$^+$ with $0 < \eta < \frac{1}{DL_u}$ ensures $\|\boldsymbol{\theta}^{t+\frac{1}{2}} - \boldsymbol{\theta}^t\|_2 \to 0$ and $\|\boldsymbol{\theta}^{t+1} - \boldsymbol{\theta}^{t+\frac{1}{2}}\|_2 \to 0$ as $t \to \infty$, and $\min_{\tau \in [t]} \left( \|\boldsymbol{\theta}^{\tau+\frac{1}{2}} - \boldsymbol{\theta}^\tau\|_2^2 + \|\boldsymbol{\theta}^{\tau+1} - \boldsymbol{\theta}^{\tau+\frac{1}{2}}\|_2^2 \right) \le O(\frac{1}{t})$, $\forall t \ge 1$.*

**Theorem 4.3.** *SPRM$^+$ with $0 < \eta < \frac{1}{8DL_u}$ ensures $\|\boldsymbol{\theta}^{t+\frac{1}{2}} - \boldsymbol{\theta}^t\|_2 \to 0$ and $\|\boldsymbol{\theta}^{t+1} - \boldsymbol{\theta}^{t+\frac{1}{2}}\|_2 \to 0$ as $t \to \infty$, and $\min_{\tau \in [t]} \left( \|\boldsymbol{\theta}^{\tau+\frac{1}{2}} - \boldsymbol{\theta}^\tau\|_2^2 + \|\boldsymbol{\theta}^{\tau+1} - \boldsymbol{\theta}^{\tau+\frac{1}{2}}\|_2^2 \right) \le O(\frac{1}{t})$, $\forall t \ge 1$.*

**Lemma 4.4.** *(Proposition 1 in Farina et al. (2023)) $\forall \boldsymbol{a}, \boldsymbol{b} \in \mathbb{R}^d_{\ge 0}$, $\|\boldsymbol{a}\|_1 \ge 1$, $\|\boldsymbol{b}\|_1 \ge 1$, $\|\frac{\boldsymbol{a}}{\|\boldsymbol{a}\|_1} - \frac{\boldsymbol{b}}{\|\boldsymbol{b}\|_1}\|_2 \le \sqrt{d}\|\boldsymbol{a} - \boldsymbol{b}\|_2$.*

*Proof.* Now, we start to prove the last-iterate and best-iterate convergence of SExRM$^+$ and SPRM$^+$. Firstly, from the analysis in Section 3 that Eq. (4) can be written as the form in Eq. (5), the update rule of SExRM$^+$ can be written as (see details in Appendix F)

$$\boldsymbol{x}_i^{t+\frac{1}{2}} \in \underset{\boldsymbol{x}_i \in \boldsymbol{\mathcal{X}}_i}{\operatorname{argmin}} \{\langle \boldsymbol{\ell}_i^t, \boldsymbol{x}_i \rangle + q_i^{t-\frac{1}{2}}(\boldsymbol{x}_i) + D_{q_i^{0:t-1}}(\boldsymbol{x}_i, \boldsymbol{x}_i^t)\},$$

$$\boldsymbol{x}_i^{t+1} \in \underset{\boldsymbol{x}_i \in \boldsymbol{\mathcal{X}}_i}{\operatorname{argmin}} \{\langle \boldsymbol{\ell}_i^{t+\frac{1}{2}}, \boldsymbol{x}_i \rangle + q_i^t(\boldsymbol{x}_i) + D_{q_i^{0:t-1}}(\boldsymbol{x}_i, \boldsymbol{x}_i^t)\}, \tag{7}$$

$$q_i^{0:t-1}(\boldsymbol{x}_i) = \frac{\|\boldsymbol{\theta}_i^t\|_1}{\eta}\psi(\boldsymbol{x}_i), q_i^{0:t-1}(\boldsymbol{x}_i) + q_i^{t-\frac{1}{2}}(\boldsymbol{x}) = \frac{\|\boldsymbol{\theta}_i^{t+\frac{1}{2}}\|_1}{\eta}\psi(\boldsymbol{x}_i), q_i^{0:t-1}(\boldsymbol{x}_i) + q_i^t(\boldsymbol{x}_i) = \frac{\|\boldsymbol{\theta}_i^{t+1}\|_1}{\eta}\psi(\boldsymbol{x}_i).$$

Similarly, the update rule of SPRM$^+$ can be written as

$$\boldsymbol{x}_i^{t+\frac{1}{2}} \in \underset{\boldsymbol{x}_i \in \boldsymbol{\mathcal{X}}_i}{\operatorname{argmin}} \{\langle \boldsymbol{\ell}_i^{t-\frac{1}{2}}, \boldsymbol{x}_i \rangle + q_i^{t-\frac{1}{2}}(\boldsymbol{x}_i) + D_{q_i^{0:t-1}}(\boldsymbol{x}_i, \boldsymbol{x}_i^t)\},$$

$$\boldsymbol{x}_i^{t+1} \in \underset{\boldsymbol{x}_i \in \boldsymbol{\mathcal{X}}_i}{\operatorname{argmin}} \{\langle \boldsymbol{\ell}_i^{t+\frac{1}{2}}, \boldsymbol{x}_i \rangle + q_i^t(\boldsymbol{x}_i) + D_{q_i^{0:t-1}}(\boldsymbol{x}_i, \boldsymbol{x}_i^t)\}, \tag{8}$$

$$q_i^{0:t-1}(\boldsymbol{x}_i) = \frac{\|\boldsymbol{\theta}_i^t\|_1}{\eta}\psi(\boldsymbol{x}_i), q_i^{0:t-1}(\boldsymbol{x}_i) + q_i^{t-\frac{1}{2}}(\boldsymbol{x}) = \frac{\|\boldsymbol{\theta}_i^{t+\frac{1}{2}}\|_1}{\eta}\psi(\boldsymbol{x}_i), q_i^{0:t-1}(\boldsymbol{x}_i) + q_i^t(\boldsymbol{x}_i) = \frac{\|\boldsymbol{\theta}_i^{t+1}\|_1}{\eta}\psi(\boldsymbol{x}_i).$$

Monotonicity is recovered since the loss gradient $\boldsymbol{\ell}$ is a monotonic operator. Now, we prove the tangent residual of the strategy profiles $\boldsymbol{x}^t$ converges to 0. From the analysis in Phase 2 of Section 3, according to the second prox-mapping operator in Eq. (7) and Eq. (8), $\forall \boldsymbol{x} \in \boldsymbol{\mathcal{X}}$, we have

$$-\boldsymbol{\ell}^{t+\frac{1}{2}} + \frac{\boldsymbol{\theta}^t - \boldsymbol{\theta}^{t+1}}{\eta} \in \mathcal{N}_{\boldsymbol{\mathcal{X}}}(\boldsymbol{x}^{t+1}).$$

From the definition of the tangent residual, we obtain

$$r^{tan}(\boldsymbol{x}^{t+1}) \le \|\boldsymbol{\ell}^{t+1} - \boldsymbol{\ell}^{t+\frac{1}{2}} + \frac{\boldsymbol{\theta}^t - \boldsymbol{\theta}^{t+1}}{\eta}\|_2 \le \|\boldsymbol{\ell}^{t+1} - \boldsymbol{\ell}^{t+\frac{1}{2}}\|_2 + \frac{1}{\eta}\|\boldsymbol{\theta}^t - \boldsymbol{\theta}^{t+1}\|_2$$

$$\le L\|\boldsymbol{x}^{t+1} - \boldsymbol{x}^{t+\frac{1}{2}}\|_2 + \frac{1}{\eta}\|\boldsymbol{\theta}^t - \boldsymbol{\theta}^{t+\frac{1}{2}}\|_2 + \frac{1}{\eta}\|\boldsymbol{\theta}^{t+\frac{1}{2}} - \boldsymbol{\theta}^{t+1}\|_2,$$

where the last inequality is from the smoothness of the smooth games. Then, using Lemma 4.4 with $\boldsymbol{a} = \boldsymbol{\theta}^{t+1}$ and $\boldsymbol{b} = \boldsymbol{\theta}^{t+\frac{1}{2}}$, we have

$$r^{tan}(\boldsymbol{x}^{t+1}) \le L\sqrt{D}\|\boldsymbol{\theta}^{t+1} - \boldsymbol{\theta}^{t+\frac{1}{2}}\|_2 + \frac{1}{\eta}\|\boldsymbol{\theta}^t - \boldsymbol{\theta}^{t+\frac{1}{2}}\|_2 + \frac{1}{\eta}\|\boldsymbol{\theta}^{t+\frac{1}{2}} - \boldsymbol{\theta}^{t+1}\|_2.$$

From Theorem 4.2 and 4.3 ($\|\boldsymbol{\theta}^t - \boldsymbol{\theta}^{t+\frac{1}{2}}\|_2 \to 0$ and $\|\boldsymbol{\theta}^{t+\frac{1}{2}} - \boldsymbol{\theta}^{t+1}\|_2 \to 0$), we get $r^{tan}(\boldsymbol{x}^t) \to 0$ as $t \to \infty$. Similarly, we get

$$
\begin{aligned}
(r^{tan}(\boldsymbol{x}^{t+1}))^2 &\leq \|\boldsymbol{\ell}^{t+1} - \boldsymbol{\ell}^{t+\frac{1}{2}} + \frac{\boldsymbol{\theta}^t - \boldsymbol{\theta}^{t+1}}{\eta}\|_2^2 \leq 2\|\boldsymbol{\ell}^{t+1} - \boldsymbol{\ell}^{t+\frac{1}{2}}\|_2^2 + \frac{2}{\eta^2}\|\boldsymbol{\theta}^t - \boldsymbol{\theta}^{t+1}\|_2^2 \\
&\leq 2L^2\|\boldsymbol{x}^{t+1} - \boldsymbol{x}^{t+\frac{1}{2}}\|_2 + \frac{2}{\eta^2}\|\boldsymbol{\theta}^t - \boldsymbol{\theta}^{t+1}\|_2^2 \\
&\leq 2L^2D^2\|\boldsymbol{\theta}^{t+1} - \boldsymbol{\theta}^{t+\frac{1}{2}}\|_2^2 + \frac{4}{\eta^2}\|\boldsymbol{\theta}^t - \boldsymbol{\theta}^{t+\frac{1}{2}}\|_2^2 + \frac{4}{\eta^2}\|\boldsymbol{\theta}^{t+\frac{1}{2}} - \boldsymbol{\theta}^{t+1}\|_2^2 \\
&\leq \left(2L^2D^2 + \frac{4}{\eta^2}\right)\left(\|\boldsymbol{\theta}^{t+1} - \boldsymbol{\theta}^{t+\frac{1}{2}}\|_2^2 + \|\boldsymbol{\theta}^t - \boldsymbol{\theta}^{t+\frac{1}{2}}\|_2^2\right).
\end{aligned}
$$

Therefore, from Theorem 4.2 and 4.3 ($\min_{\tau \in [t]} \left(\|\boldsymbol{\theta}^{\tau+\frac{1}{2}} - \boldsymbol{\theta}^\tau\|_2^2 + \|\boldsymbol{\theta}^{\tau+1} - \boldsymbol{\theta}^{\tau+\frac{1}{2}}\|_2^2\right) \leq O(\frac{1}{t})$), we get that for $\tau = \arg\min_{\tau \in [t]} \left(\|\boldsymbol{\theta}^{\tau+\frac{1}{2}} - \boldsymbol{\theta}^\tau\|_2^2 + \|\boldsymbol{\theta}^{\tau+1} - \boldsymbol{\theta}^{\tau+\frac{1}{2}}\|_2^2\right)$,

$$
r^{tan}(\boldsymbol{x}^{\tau+1}) \leq \sqrt{O\left(\left(\|\boldsymbol{\theta}^{\tau+\frac{1}{2}} - \boldsymbol{\theta}^\tau\|_2^2 + \|\boldsymbol{\theta}^{\tau+1} - \boldsymbol{\theta}^{\tau+\frac{1}{2}}\|_2^2\right)\right)} \leq O(\frac{1}{\sqrt{t}}).
$$

These complete the proof. $\qquad\square$

## 5 OUR ALGORITHM: SOGRM$^+$

By using our proof paradigm, we prove that SExRM$^+$ and SPRM$^+$ achieve last-iterate convergence in games satisfying monotonicity. However, OMD/FTRL based algorithms even achieve last-iterate convergence in games satisfying the weak MVI, covering games satisfying monotonicity.

Inspired by our paradigm, we propose a new smooth RM$^+$ variant called Smooth Optimistic Gradient Regret Matching$^+$ (SOGRM$^+$). We prove that SOGRM$^+$ achieves last-iterate convergence in games satisfying monotonicity via our paradigm. SOGRM$^+$ is connected to an OG instance which updates at $\mathbb{R}_{\geq 1}^d$, the subset of cone($\boldsymbol{\mathcal{X}}_i$). Note that the proof of SOGRM$^+$ needs additional techniques, such as transforming variables using the definition of the inner product to employ the weak MVI and tangent residual (details are in Eq. (11), (12), (35), (36), and (38)), rather than directly transforming variables using equalities as in OG. Formally, the update rule of SOGRM$^+$ at iteration $t$ is

$$
\begin{aligned}
\boldsymbol{\theta}_i^{t+\frac{1}{2}} &\in \arg\min_{\boldsymbol{\theta}_i \in \mathbb{R}_{\geq 1}^{|A_i|}} \{\langle -\boldsymbol{F}_i(\boldsymbol{\theta}^{t-\frac{1}{2}}), \boldsymbol{\theta}_i \rangle + \frac{1}{\eta}D_\psi(\boldsymbol{\theta}_i, \boldsymbol{\theta}_i^t)\}, \quad \boldsymbol{x}_i^{t+\frac{1}{2}} = \frac{\boldsymbol{\theta}_i^{t+\frac{1}{2}}}{\|\boldsymbol{\theta}_i^{t+\frac{1}{2}}\|_1}, \\
\boldsymbol{\theta}_i^{t+1} &= \boldsymbol{\theta}_i^{t+\frac{1}{2}} - \eta\boldsymbol{F}_i(\boldsymbol{\theta}^{t-\frac{1}{2}}) + \eta\boldsymbol{F}_i(\boldsymbol{\theta}^{t+\frac{1}{2}}).
\end{aligned} \tag{9}
$$

**Theorem 5.1.** *In smooth games satisfying the weak MVI with $\rho > -\frac{1}{12\sqrt{3}DL_u}$, there always exists $0 < \eta < \frac{1}{2DL_u}$ that ensures SOGRM$^+$ achieves asymptotic last-iterate convergence and $O(\frac{1}{\sqrt{t}})$ best-iterate convergence rate in learning an NE of these games, where $D = \max_{i \in \mathcal{N}} |A_i|$ and $L_u = \sqrt{2P^2 + 4L^2}$. Specifically, if all players follow the update rule of SOGRM$^+$, then $r^{tan}(\boldsymbol{x}^{t+\frac{1}{2}}) \to 0$ and $\min_{\tau \in [t]} r^{tan}(\boldsymbol{x}^{\tau+\frac{1}{2}}) \leq O(\frac{1}{\sqrt{t}})$ as $t \to \infty$.*

To prove the convergence of SOGRM$^+$, we introduce Theorem 5.2 and Lemma 5.3, whose proofs are in Appendix D and E, respectively.

**Theorem 5.2.** *If $\rho > -\frac{1}{12\sqrt{3}DL_u}$, there always exist $0 < \eta < \frac{1}{2DL_u}$ that ensures $\|\boldsymbol{\theta}^{t+1} - \boldsymbol{\theta}^t\|_2 \to 0$ as $t \to \infty$ and $\min_{\tau \in [t]} \|\boldsymbol{\theta}^{\tau+1} - \boldsymbol{\theta}^\tau\|_2^2 \leq O(\frac{1}{t})$, $\forall t \geq 1$.*

**Lemma 5.3.** *If all players follow the update rule of SOGRM$^+$, $\forall \boldsymbol{x} \in \boldsymbol{\mathcal{X}}$,*

$$
\sum_{i \in \mathcal{N}} \langle \boldsymbol{\ell}_i^{t-\frac{1}{2}} - \frac{\boldsymbol{\theta}^t - \boldsymbol{\theta}^{t+\frac{1}{2}}}{\eta}, \boldsymbol{x}_i - \boldsymbol{x}_i^{t+\frac{1}{2}} \rangle = \sum_{i \in \mathcal{N}} \langle \boldsymbol{\ell}_i^{t+\frac{1}{2}} - \frac{\boldsymbol{\theta}^t - \boldsymbol{\theta}^{t+1}}{\eta}, \boldsymbol{x}_i - \boldsymbol{x}_i^{t+\frac{1}{2}} \rangle.
$$

*Proof.* Now, we prove Theorem 5.1 via our proof paradigm. Firstly, from the equivalence in Section 3 (Eq. (4) can be written as the form in Eq. (5)), the update rule of SOGRM+ can be written as (see details in Appendix F)

$$
\boldsymbol{x}_i^{t+\frac{1}{2}} \in \arg\min_{\boldsymbol{x}_i \in \boldsymbol{\mathcal{X}}_i} \{\langle \boldsymbol{\ell}_i^{t-\frac{1}{2}}, \boldsymbol{x}_i \rangle + q_i^{t-\frac{1}{2}}(\boldsymbol{x}_i) + D_{q_i^{0:t-1}}(\boldsymbol{x}_i, \boldsymbol{x}_i^t)\}, \boldsymbol{\theta}_i^{t+1} = \boldsymbol{\theta}_i^{t+\frac{1}{2}} - \eta\boldsymbol{F}_i(\boldsymbol{\theta}^{t-\frac{1}{2}}) + \eta\boldsymbol{F}_i(\boldsymbol{\theta}^{t+\frac{1}{2}}),
$$

$$
q_i^{0:t-1}(\boldsymbol{x}_i) = \frac{\|\boldsymbol{\theta}_i^t\|_1}{\eta}\psi(\boldsymbol{x}_i), q_i^{0:t-1}(\boldsymbol{x}_i) + q_i^{t-\frac{1}{2}}(\boldsymbol{x}) = \frac{\|\boldsymbol{\theta}_i^{t+\frac{1}{2}}\|_1}{\eta}\psi(\boldsymbol{x}_i), \tag{10}
$$

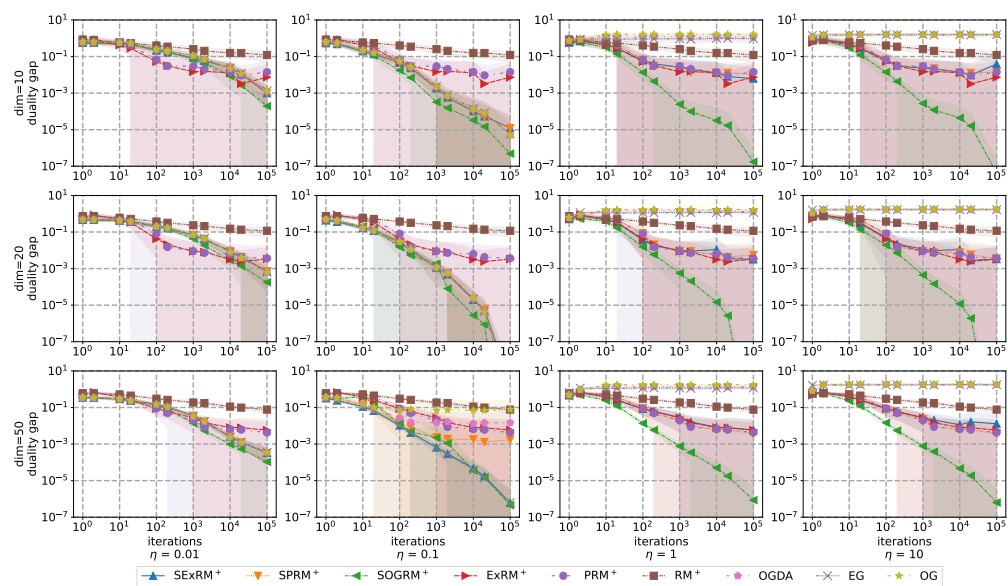

Figure 1: Performance of different algorithms in $10 \times 10$ (top), $20 \times 20$ (middle), $50 \times 50$ (bottom) randomly generated two-player zero-sum matrix games.

From the first-order optimality of the first prox-mapping operator in Eq. (10), $\forall \boldsymbol{x} \in \mathcal{X}$, we have

$$\langle \boldsymbol{\ell}_i^{t-\frac{1}{2}} + \nabla_{\boldsymbol{x}_i^{t+\frac{1}{2}}} q_i^{t-\frac{1}{2}}(\boldsymbol{x}_i^{t+\frac{1}{2}}) + \nabla_{\boldsymbol{x}_i^{t+\frac{1}{2}}} D_{q_i^{0:t-1}}(\boldsymbol{x}_i^{t+\frac{1}{2}}, \boldsymbol{x}_i^t), \boldsymbol{x}_i - \boldsymbol{x}_i^{t+\frac{1}{2}} \rangle \geq 0$$

$$\Leftrightarrow \sum_{i \in \mathcal{N}} \langle \boldsymbol{\ell}_i^{t-\frac{1}{2}} + \nabla_{\boldsymbol{x}_i^{t+\frac{1}{2}}} q_i^{0:t-1}(\boldsymbol{x}^{t+\frac{1}{2}}) + \nabla_{\boldsymbol{x}_i^{t+\frac{1}{2}}} q_i^{t-\frac{1}{2}}(\boldsymbol{x}^{t+\frac{1}{2}}) - \nabla_{\boldsymbol{x}_i^t} q_i^{0:t-1}(\boldsymbol{x}^t), \boldsymbol{x}_i - \boldsymbol{x}_i^{t+\frac{1}{2}} \rangle \geq 0.$$

Then, we have

$$\sum_{i \in \mathcal{N}} \langle \boldsymbol{\ell}_i^{t-\frac{1}{2}} - \frac{\boldsymbol{\theta}^t - \boldsymbol{\theta}^{t+\frac{1}{2}}}{\eta}, \boldsymbol{x}_i - \boldsymbol{x}_i^{t+\frac{1}{2}} \rangle \geq 0 \text{ and } \sum_{i \in \mathcal{N}} \langle \boldsymbol{\ell}_i^{t+\frac{1}{2}} - \frac{\boldsymbol{\theta}^t - \boldsymbol{\theta}^{t+1}}{\eta}, \boldsymbol{x}_i - \boldsymbol{x}_i^{t+\frac{1}{2}} \rangle \geq 0, \quad (11)$$

where the left-hand side is from $\nabla_{\boldsymbol{x}_i^{t+\frac{1}{2}}} q_i^{0:t-1}(\boldsymbol{x}^{t+\frac{1}{2}}) + \nabla_{\boldsymbol{x}_i^{t+\frac{1}{2}}} q_i^{t-\frac{1}{2}}(\boldsymbol{x}^{t+\frac{1}{2}}) = \boldsymbol{\theta}_i^{t+\frac{1}{2}}/\eta$ with $\nabla_{\boldsymbol{x}_i^t} q_i^{0:t-1}(\boldsymbol{x}_i^t) = \boldsymbol{\theta}_i^t/\eta$, and the right-hand side is from Lemma 5.3. According to Eq. (11) and the definition of the normal cone, we have

$$-\boldsymbol{\ell}^{t+\frac{1}{2}} + \frac{\boldsymbol{\theta}^t - \boldsymbol{\theta}^{t+1}}{\eta} \in \mathcal{N}_{\mathcal{X}}(\boldsymbol{x}^{t+\frac{1}{2}}), \quad (12)$$

where $\boldsymbol{\ell}^{t+\frac{1}{2}} = [\boldsymbol{\ell}_i^{t+\frac{1}{2}} : i \in \mathcal{N}]$. From the definition of the tangent residual, we obtain

$$r^{tan}(\boldsymbol{x}^{t+\frac{1}{2}}) \leq \|\boldsymbol{\ell}^{t+\frac{1}{2}} - \boldsymbol{\ell}^{t+\frac{1}{2}} + \frac{\boldsymbol{\theta}^t - \boldsymbol{\theta}^{t+1}}{\eta}\|_2 \leq \frac{1}{\eta} \|\boldsymbol{\theta}^t - \boldsymbol{\theta}^{t+1}\|_2. \quad (13)$$

Combining Eq. (13), Theorem 5.2 ($\|\boldsymbol{\theta}^t - \boldsymbol{\theta}^{t+1}\|_2 \to 0$), we get $r^{tan}(\boldsymbol{x}^{t+\frac{1}{2}}) \to 0$ as $t \to \infty$. Similarly, from Theorem 5.2 ($\min_{\tau \in [t]} \|\boldsymbol{\theta}^{\tau+1} - \boldsymbol{\theta}^\tau\|_2^2 \leq O(\frac{1}{t})$), we get that for $\tau = \arg\min_{\tau \in [t]} \|\boldsymbol{\theta}^\tau - \boldsymbol{\theta}^{\tau+1}\|_2^2$,

$$r^{tan}(\boldsymbol{x}^{\tau+\frac{1}{2}}) \leq \frac{1}{\eta} \|\boldsymbol{\theta}^\tau - \boldsymbol{\theta}^{\tau+1}\|_2 \leq O(\frac{1}{\sqrt{t}}).$$

$\square$

## 6 EXPERIMENTS

We conduct experiments on randomly generated two-player zero-sum matrix games with sizes $[10, 20, 50]$, where learning an NE is defined as $\min_{\boldsymbol{x}_0 \in \mathcal{X}_0} \max_{\boldsymbol{x}_1 \in \mathcal{X}_1} \boldsymbol{x}_0^{\mathrm{T}} \boldsymbol{A} \boldsymbol{x}_1$. Each element of the payoff matrix $\boldsymbol{A}$ is uniformly sampled from $[-1, 1]$. For each game size, we generate 20 instances and report the average duality gaps with variances. The duality gap, $r^{dg}(\boldsymbol{x})$, is used to evaluate the

distance to NE, defined as $r^{dg}(\boldsymbol{x}) = \sum_{i \in \mathcal{N}} \max_{\boldsymbol{x}'_i} \langle \boldsymbol{\ell}^{\boldsymbol{x}}_i, \boldsymbol{x}_i - \boldsymbol{x}'_i \rangle$. As analyzed in Cai et al. (2022b), the duality gap involves a lower bound of the tangent residual, $r^{dg}(\boldsymbol{x}) \leq C_0 r^{tan}(\boldsymbol{x})$, where $C_0$ is a game-dependent constant. Thus, if the tangent residual converges to $0$, the duality gap also converges to $0$. Due to the difficulty in precisely calculating the tangent residual, we do not use it as the metric. We compare smooth RM$^+$ variants (SExRM$^+$, SPRM$^+$, and SOGRM$^+$) with existing RM$^+$ variants (ExRM$^+$, PRM$^+$, and RM$^+$), as well as traditional last-iterate convergence OMD based algorithms—Optimistic Gradient Descent (OGDA) (Wei et al., 2021), Extra-Gradient (EG) (Korpelevich, 1976), and Optimistic Gradient (OG) (Hsieh et al., 2019; Cai & Zheng, 2022)[2]. For initialization, we set $\boldsymbol{\theta}_i$ to $\mathbf{1}_{|\boldsymbol{\mathcal{X}}_i|}/|\boldsymbol{\mathcal{X}}_i|$ and $\mathbf{0}$ for smooth and other RM$^+$ variants, respectively. For OGDA, EG, and OG, the initial strategy is the uniform strategy. For all tested algorithm, we use simultaneous updates since to the best of our knowledge, the theoretical analysis of existing work on last-iterate convergence is based on simultaneous updates. All experiments are performed on a machine with an i9-13900K CPU and 128 GB of memory.

The convergence results are shown in Figure 1, smooth RM$^+$ variants generally achieve at least similar performance compared to other algorithms. Specifically, OGDA, EG, and OG underperform relative to their smooth RM$^+$ counterparts (SPRM$^+$, SExRM$^+$, and SOGRM$^+$, respectively) and are more sensitive to parameters. For larger $\eta$ values ($\eta = 1$ and $\eta = 10$), OGDA, EG, and OG consistently diverge, while smooth RM$^+$ variants maintain last-iterate convergence. Additionally, we observe that SPRM$^+$ and SExRM$^+$ consistently achieve comparable performance to their corresponding non-smooth RM$^+$ variants, namely PRM$^+$ and ExRM$^+$, respectively. Under optimal parameter settings, SPRM$^+$ and SExRM$^+$ significantly outperform PRM$^+$ and ExRM$^+$, respectively. More importantly, we find that our algorithm, SOGRM$^+$, exhibits the fastest convergence rate and shows the least sensitivity to parameter changes. Moreover, for the similar performance of SOGRM$^+$ under $\eta = 1$ and $\eta = 10$, we hypothesize that when $\eta \geq 1$, the term $\eta \boldsymbol{F}_i(\boldsymbol{\theta}^{t-\frac{1}{2}})$ becomes extremely larger than $\boldsymbol{\theta}^1_i$, either positively or negatively. Consequently, the accumulated regret $\boldsymbol{\theta}^{t+\frac{1}{2}}_i$ heavily depends on the feedback $\eta \boldsymbol{F}_i(\boldsymbol{\theta}^{\tau-\frac{1}{2}})$ from iterations $\tau$ ($\tau < t$) rather than $\boldsymbol{\theta}^1_i$. Since the strategies are derived by normalizing the accumulated regret $\boldsymbol{\theta}^{t+\frac{1}{2}}$, the resulting strategies exhibit only minor differences. Therefore, we can observe the similar performance of SOGRM$^+$ under $\eta = 1$ and $\eta = 10$.

## 7 CONCLUSIONS

We study the last-iterate convergence of RM$^+$ variants in learning an NE for games that satisfy monotonicity or only the weak MVI. We introduce a novel proof paradigm to analyze the last-iterate convergence of RM$^+$ variants. Using this paradigm, we show that two existing variants, SExRM$^+$ and SPRM$^+$, exhibit last-iterate convergence in games with monotonicity. Building on this, we propose a new variant, SOGRM$^+$, which achieves last-iterate convergence in games satisfying the weak MVI. To our knowledge, this is the first last-iterate convergence results for RM$^+$ variants in such games. Our paradigm stands out for its simplicity and innovation, and we believe this approach can extend to proving last-iterate convergence for additional RM$^+$ variants in broader game classes.

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

## A   DETAILED PROOF OF SECTION 3

In this section, we provide a detailed proof for Section 3. Firstly, as shown in Section 3, the update rule in Eq. (5) can be written as (more details about how to get this form is in Appendix A.1)

$$\boldsymbol{x}_i^{t_2} = \frac{[\boldsymbol{\theta}_i^{t_0} + \alpha\mathbf{1} - \eta\boldsymbol{\ell}_i^{\boldsymbol{\theta}^{t_1}}]_+}{\|\boldsymbol{\theta}_i^{t_2}\|_1}, \tag{14}$$

where $\alpha$ is a unique constant to ensure $\|\boldsymbol{x}_i^{t_2}\|_1 = 1$ ($\alpha$ always exists) (see the reason why $\alpha$ is a unique constant in the proof of Theorem 2.2 of Chen & Ye (2011)) (note that $\boldsymbol{\theta}_i^{t_2}, \boldsymbol{\theta}_i^{t_0}$, and $\boldsymbol{\ell}_i^{\boldsymbol{\theta}^{t_1}}$ are the same between Eq. (4) and Eq. (5)).

Now, considering Eq. (4), as we show in Section 3, $\boldsymbol{\theta}_i^{t_2}$ in Eq. (4) can be obtained via (more details about how to get this form is in Appendix A.2)

$$\boldsymbol{\theta}_i^{t_2} = \begin{cases} [\boldsymbol{\theta}_i^{t_0} + \eta\langle\frac{\boldsymbol{\theta}_i^{t_1}}{\|\boldsymbol{\theta}_i^{t_1}\|_1}, \boldsymbol{\ell}_i^{\boldsymbol{\theta}^{t_1}}\rangle\mathbf{1} - \eta\boldsymbol{\ell}_i^{\boldsymbol{\theta}^{t_1}}]_+ & \text{if } \|[\boldsymbol{\theta}_i^{t_0} + \eta\langle\frac{\boldsymbol{\theta}_i^{t_1}}{\|\boldsymbol{\theta}_i^{t_1}\|_1}, \boldsymbol{\ell}_i^{\boldsymbol{\theta}^{t_1}}\rangle\mathbf{1} - \eta\boldsymbol{\ell}_i^{\boldsymbol{\theta}^{t_1}}]_+\|_1 \geq 1, \\ [\boldsymbol{\theta}_i^{t_0} + \eta\langle\frac{\boldsymbol{\theta}_i^{t_1}}{\|\boldsymbol{\theta}_i^{t_1}\|_1}, \boldsymbol{\ell}_i^{\boldsymbol{\theta}^{t_1}}\rangle\mathbf{1} - \eta\boldsymbol{\ell}_i^{\boldsymbol{\theta}^{t_1}} + \beta\mathbf{1}]_+ & \text{if } \|[\boldsymbol{\theta}_i^{t_0} + \eta\langle\frac{\boldsymbol{\theta}_i^{t_1}}{\|\boldsymbol{\theta}_i^{t_1}\|_1}, \boldsymbol{\ell}_i^{\boldsymbol{\theta}^{t_1}}\rangle\mathbf{1} - \eta\boldsymbol{\ell}_i^{\boldsymbol{\theta}^{t_1}}]_+\|_1 < 1. \end{cases} \tag{15}$$

where $\beta$ exists and is unique to ensure $\|\boldsymbol{\theta}_i^{t_2}\|_1 = 1$ (we can get $\beta$ if we know $\boldsymbol{\theta}_i^{t_0} + \eta\langle\frac{\boldsymbol{\theta}_i^{t_1}}{\|\boldsymbol{\theta}_i^{t_1}\|_1}, \boldsymbol{\ell}_i^{\boldsymbol{\theta}^{t_1}}\rangle\mathbf{1} - \eta\boldsymbol{\ell}_i^{\boldsymbol{\theta}^{t_1}}$) (see the reason why $\beta$ is a unique constant in the proof of Theorem 2.2 of Chen & Ye (2011)). Assume $\alpha$ in Eq. (14) is

$$\alpha = \begin{cases} \eta\langle\frac{\boldsymbol{\theta}_i^{t_1}}{\|\boldsymbol{\theta}_i^{t_1}\|_1}, \boldsymbol{\ell}_i^{\boldsymbol{\theta}^{t_1}}\rangle & \text{if } \|[\boldsymbol{\theta}_i^{t_0} + \eta\langle\frac{\boldsymbol{\theta}_i^{t_1}}{\|\boldsymbol{\theta}_i^{t_1}\|_1}, \boldsymbol{\ell}_i^{\boldsymbol{\theta}^{t_1}}\rangle\mathbf{1} - \eta\boldsymbol{\ell}_i^{\boldsymbol{\theta}^{t_1}}]_+\|_1 \geq 1, \\ \eta\langle\frac{\boldsymbol{\theta}_i^{t_1}}{\|\boldsymbol{\theta}_i^{t_1}\|_1}, \boldsymbol{\ell}_i^{\boldsymbol{\theta}^{t_1}}\rangle + \beta & \text{if } \|[\boldsymbol{\theta}_i^{t_0} + \eta\langle\frac{\boldsymbol{\theta}_i^{t_1}}{\|\boldsymbol{\theta}_i^{t_1}\|_1}, \boldsymbol{\ell}_i^{\boldsymbol{\theta}^{t_1}}\rangle\mathbf{1} - \eta\boldsymbol{\ell}_i^{\boldsymbol{\theta}^{t_1}}]_+\|_1 < 1. \end{cases} \tag{16}$$

Then, $[\boldsymbol{\theta}_i^{t_0} + \alpha\mathbf{1} - \eta\boldsymbol{\ell}_i^{\boldsymbol{\theta}^{t_1}}]_+ = \boldsymbol{\theta}_i^{t_2}$. Therefore, substituting Eq. (16) into Eq. (14), we get that the update rule in Eq. (14) (or Eq. (5)) can be written as (since $[\boldsymbol{\theta}_i^{t_0} + \alpha\mathbf{1} - \eta\boldsymbol{\ell}_i^{\boldsymbol{\theta}^{t_1}}]_+ = \boldsymbol{\theta}_i^{t_2}$)

$$\boldsymbol{x}_i^{t_2} = \frac{[\boldsymbol{\theta}_i^{t_0} + \alpha\mathbf{1} - \eta\boldsymbol{\ell}_i^{\boldsymbol{\theta}^{t_1}}]_+}{\|\boldsymbol{\theta}_i^{t_2}\|_1} = \frac{\boldsymbol{\theta}_i^{t_2}}{\|\boldsymbol{\theta}_i^{t_2}\|_1}, \tag{17}$$

which enables $\|\boldsymbol{x}_i^{t_2}\|_1 = 1$ (the $\boldsymbol{x}_i^{t_2}$ in Eq. (5)). In addition, we have $\alpha$ is unique. Hence, the value of $\alpha$ must be same as Eq. (16) shown! Also, note the $\boldsymbol{x}_i^{t_2}$ in Eq. (4) is obtained via

$$\boldsymbol{x}_i^{t_2} = \frac{\boldsymbol{\theta}_i^{t_2}}{\|\boldsymbol{\theta}_i^{t_2}\|_1}. \tag{18}$$

Therefore, combining Eq. (17) with Eq. (18), we get that the update rule in Eq. (4) is the same as Eq. (5). It completes the proof[3].

### A.1   DETAILS ABOUT GETTING EQ. (14)

From Eq. (5), we have

$$\boldsymbol{x}_i^{t_2} \in \underset{\boldsymbol{x}_i\in\boldsymbol{\mathcal{X}}_i}{\arg\min}\{\langle\boldsymbol{\ell}_i^{\boldsymbol{\theta}^{t_1}}, \boldsymbol{x}_i\rangle + f_i(\boldsymbol{x}_i) + D_{h_i}(\boldsymbol{x}_i, \boldsymbol{x}_i^{t_0})\}$$

$$\Leftrightarrow \boldsymbol{x}_i^{t_2} \in \underset{\boldsymbol{x}_i\in\boldsymbol{\mathcal{X}}_i}{\arg\min}\{\langle\boldsymbol{\ell}_i^{\boldsymbol{\theta}^{t_1}}, \boldsymbol{x}_i\rangle + \frac{\|\boldsymbol{\theta}_i^{t_2}\|_1}{\eta}\psi(\boldsymbol{x}_i) - \frac{\|\boldsymbol{\theta}_i^{t_0}\|_1}{\eta}\psi(\boldsymbol{x}_i) + D_{\frac{\|\boldsymbol{\theta}_i^{t_0}\|_1}{\eta}\psi}(\boldsymbol{x}_i, \boldsymbol{x}_i^{t_0})\}.$$

---

[3]To verify our proof, we experimented $10^6$ times and did not find a counterexample.

Since $\psi(\cdot)$ is the quadratic regularizer (in other words, $\forall \boldsymbol{a}, \boldsymbol{b} \in \mathbb{R}^d, c \in \mathbb{R}, c\psi(\boldsymbol{a}) = c\|\boldsymbol{a}\|_2^2/2$, $D_{c\psi}(\boldsymbol{a}, \boldsymbol{b}) = c\|\boldsymbol{a} - \boldsymbol{b}\|_2^2/2$), we have

$$
\begin{aligned}
\boldsymbol{x}_i^{t_2} &\in \operatorname*{arg\,min}_{\boldsymbol{x}_i \in \boldsymbol{\mathcal{X}}_i} \{\langle \boldsymbol{\ell}_i^{\boldsymbol{\theta^{t_1}}}, \boldsymbol{x}_i \rangle + f_i(\boldsymbol{x}_i) + D_{h_i}(\boldsymbol{x}_i, \boldsymbol{x}_i^{t_0})\} \\
&\Leftrightarrow \boldsymbol{x}_i^{t_2} \in \operatorname*{arg\,min}_{\boldsymbol{x}_i \in \boldsymbol{\mathcal{X}}_i} \{\langle \boldsymbol{\ell}_i^{\boldsymbol{\theta^{t_1}}}, \boldsymbol{x}_i \rangle + \frac{\|\boldsymbol{\theta}_i^{t_2}\|_1}{2\eta}\|\boldsymbol{x}_i\|_2^2 - \frac{\|\boldsymbol{\theta}_i^{t_0}\|_1}{2\eta}\|\boldsymbol{x}_i\|_2^2 + \frac{\|\boldsymbol{\theta}_i^{t_0}\|_1}{2\eta}\|\boldsymbol{x}_i - \boldsymbol{x}_i^{t_0}\|_2^2\} \\
&\Leftrightarrow \boldsymbol{x}_i^{t_2} \in \operatorname*{arg\,min}_{\boldsymbol{x}_i \in \boldsymbol{\mathcal{X}}_i} \{\langle 2\eta\boldsymbol{\ell}_i^{\boldsymbol{\theta^{t_1}}}, \boldsymbol{x}_i \rangle + \|\boldsymbol{\theta}_i^{t_2}\|_1\|\boldsymbol{x}_i\|_2^2 - \|\boldsymbol{\theta}_i^{t_0}\|_1\|\boldsymbol{x}_i\|_2^2 + \|\boldsymbol{\theta}_i^{t_0}\|_1\|\boldsymbol{x}_i - \boldsymbol{x}_i^{t_0}\|_2^2\} \\
&\Leftrightarrow \boldsymbol{x}_i^{t_2} \in \operatorname*{arg\,min}_{\boldsymbol{x}_i \in \boldsymbol{\mathcal{X}}_i} \{\langle 2\eta\boldsymbol{\ell}_i^{\boldsymbol{\theta^{t_1}}}, \boldsymbol{x}_i \rangle + \|\boldsymbol{\theta}_i^{t_2}\|_1\|\boldsymbol{x}_i\|_2^2 - \|\boldsymbol{\theta}_i^{t_0}\|_1\|\boldsymbol{x}_i\|_2^2 + \|\boldsymbol{\theta}_i^{t_0}\|_1\|\boldsymbol{x}_i\|_2^2 + \\
&\qquad\qquad \|\boldsymbol{\theta}_i^{t_0}\|_1\|\boldsymbol{x}_i^{t_0}\|_2^2 - 2\|\boldsymbol{\theta}_i^{t_0}\|_1\langle \boldsymbol{x}_i, \boldsymbol{x}_i^{t_0}\rangle\} \\
&\Leftrightarrow \boldsymbol{x}_i^{t_2} \in \operatorname*{arg\,min}_{\boldsymbol{x}_i \in \boldsymbol{\mathcal{X}}_i} \{\langle 2\eta\boldsymbol{\ell}_i^{\boldsymbol{\theta^{t_1}}}, \boldsymbol{x}_i \rangle + \|\boldsymbol{\theta}_i^{t_2}\|_1\|\boldsymbol{x}_i\|_2^2 - 2\|\boldsymbol{\theta}_i^{t_0}\|_1\langle \boldsymbol{x}_i, \boldsymbol{x}_i^{t_0}\rangle\} \\
&\Leftrightarrow \boldsymbol{x}_i^{t_2} \in \operatorname*{arg\,min}_{\boldsymbol{x}_i \in \boldsymbol{\mathcal{X}}_i} \{\langle 2\eta\boldsymbol{\ell}_i^{\boldsymbol{\theta^{t_1}}} - 2\|\boldsymbol{\theta}_i^{t_0}\|_1\boldsymbol{x}_i^{t_0}, \boldsymbol{x}_i \rangle + \|\boldsymbol{\theta}_i^{t_2}\|_1\|\boldsymbol{x}_i\|_2^2\} \\
&\Leftrightarrow \boldsymbol{x}_i^{t_2} \in \operatorname*{arg\,min}_{\boldsymbol{x}_i \in \boldsymbol{\mathcal{X}}_i} \{2\frac{\langle \eta\boldsymbol{\ell}_i^{\boldsymbol{\theta^{t_1}}} - \|\boldsymbol{\theta}_i^{t_0}\|_1\boldsymbol{x}_i^{t_0}, \boldsymbol{x}_i \rangle}{\|\boldsymbol{\theta}_i^{t_2}\|_1} + \|\boldsymbol{x}_i\|_2^2\} \\
&\Leftrightarrow \boldsymbol{x}_i^{t_2} \in \operatorname*{arg\,min}_{\boldsymbol{x}_i \in \boldsymbol{\mathcal{X}}_i} \{2\frac{\langle \eta\boldsymbol{\ell}_i^{\boldsymbol{\theta^{t_1}}} - \|\boldsymbol{\theta}_i^{t_0}\|_1\boldsymbol{x}_i^{t_0}, \boldsymbol{x}_i \rangle}{\|\boldsymbol{\theta}_i^{t_2}\|_1} + \|\boldsymbol{x}_i\|_2^2 + \|\frac{\eta\boldsymbol{\ell}_i^{\boldsymbol{\theta^{t_1}}} - \|\boldsymbol{\theta}_i^{t_0}\|_1\boldsymbol{x}_i^{t_0}}{\|\boldsymbol{\theta}_i^{t_2}\|_1}\|_2^2\} \\
&\Leftrightarrow \boldsymbol{x}_i^{t_2} \in \operatorname*{arg\,min}_{\boldsymbol{x}_i \in \boldsymbol{\mathcal{X}}_i} \|\frac{\|\boldsymbol{\theta}_i^{t_0}\|_1\boldsymbol{x}_i^{t_0} - \eta\boldsymbol{\ell}_i^{\boldsymbol{\theta^{t_1}}}}{\|\boldsymbol{\theta}_i^{t_2}\|_1} - \boldsymbol{x}_i\|_2^2 \\
&\Leftrightarrow \boldsymbol{x}_i^{t_2} \in \operatorname*{arg\,min}_{\boldsymbol{x}_i \in \boldsymbol{\mathcal{X}}_i} \|\frac{\boldsymbol{\theta}_i^{t_0} - \eta\boldsymbol{\ell}_i^{\boldsymbol{\theta^{t_1}}}}{\|\boldsymbol{\theta}_i^{t_2}\|_1} - \boldsymbol{x}_i\|_2^2,
\end{aligned}
\tag{19}
$$

where the last line is from $\boldsymbol{\theta}_i^{t_0} = \|\boldsymbol{\theta}_i^{t_0}\|_1\boldsymbol{x}_i^{t_0}$ ($\boldsymbol{x}_i^{t_0} = \frac{\boldsymbol{\theta}_i^{t_0}}{\|\boldsymbol{\theta}_i^{t_0}\|_1}$). Since $\boldsymbol{\mathcal{X}}_i$ is simplex, Eq. (19) indicates getting the orthogonal projection of $\frac{\boldsymbol{\theta}_i^{t_0} - \eta\boldsymbol{\ell}_i^{\boldsymbol{\theta^{t_1}}}}{\|\boldsymbol{\theta}_i^{t_2}\|_1}$ on simplex. Therefore, as analyzed in Chen & Ye (2011), the closed-form solution of Eq. (19) is Eq. (20), and $\alpha$ exists and is unique to ensure $\|\boldsymbol{x}_i^{t_2}\|_1 = 1$ (see the reason why $\alpha$ is a unique constant in the proof of Theorem 2.2 of Chen & Ye (2011)).

$$
\boldsymbol{x}_i^{t_2} = [\frac{\boldsymbol{\theta}_i^{t_0} - \eta\boldsymbol{\ell}_i^{\boldsymbol{\theta^{t_1}}}}{\|\boldsymbol{\theta}_i^{t_2}\|_1} + \alpha'\mathbf{1}]_+ = \frac{[\boldsymbol{\theta}_i^{t_0} + \alpha\mathbf{1} - \eta\boldsymbol{\ell}_i^{\boldsymbol{\theta^{t_1}}}]_+}{\|\boldsymbol{\theta}_i^{t_2}\|_1},
\tag{20}
$$

where $\alpha' = \frac{\alpha}{\|\boldsymbol{\theta}_i^{t_2}\|_1}$.

## A.2 Details about getting Eq. (15)

From Eq. (4) and $\psi(\cdot)$ is the quadratic regularizer (in other words, $\forall \boldsymbol{a}, \boldsymbol{b} \in \mathbb{R}^d, c \in \mathbb{R}, c\psi(\boldsymbol{a}) = c\|\boldsymbol{a}\|_2^2/2$, $D_{c\psi}(\boldsymbol{a}, \boldsymbol{b}) = c\|\boldsymbol{a} - \boldsymbol{b}\|_2^2/2$), we have

$$
\begin{aligned}
\boldsymbol{\theta}_i^{t_2} &\in \operatorname*{arg\,min}_{\boldsymbol{\theta}_i \in \mathbb{R}_{\geq 1}^{|A_i|}} \{\langle -\boldsymbol{F}_i(\boldsymbol{\theta}^{t_1}), \boldsymbol{\theta}_i \rangle + \frac{1}{\eta}D_\psi(\boldsymbol{\theta}_i, \boldsymbol{\theta}_i^{t_0})\} \\
&\Leftrightarrow \boldsymbol{\theta}_i^{t_2} \in \operatorname*{arg\,min}_{\boldsymbol{\theta}_i \in \mathbb{R}_{\geq 1}^{|A_i|}} \{\langle -\boldsymbol{F}_i(\boldsymbol{\theta}^{t_1}), \boldsymbol{\theta}_i \rangle + \frac{1}{2\eta}\|\boldsymbol{\theta}_i - \boldsymbol{\theta}_i^{t_0}\|_2^2\} \\
&\Leftrightarrow \boldsymbol{\theta}_i^{t_2} \in \operatorname*{arg\,min}_{\boldsymbol{\theta}_i \in \mathbb{R}_{\geq 1}^{|A_i|}} \{\langle -\boldsymbol{F}_i(\boldsymbol{\theta}^{t_1}), \boldsymbol{\theta}_i \rangle + \frac{1}{2\eta}\|\boldsymbol{\theta}_i\|_2^2 + \frac{1}{2\eta}\|\boldsymbol{\theta}_i^{t_0}\|_2^2 - 2\frac{1}{2\eta}\langle \boldsymbol{\theta}_i, \boldsymbol{\theta}_i^{t_0}\rangle\}
\end{aligned}
$$

Then, we have

$$\boldsymbol{\theta}_i^{t_2} \in \underset{\boldsymbol{\theta}_i \in \mathbb{R}_{\geq 1}^{|A_i|}}{\arg\min}\{\langle -\boldsymbol{F}_i(\boldsymbol{\theta}^{t_1}), \boldsymbol{\theta}_i\rangle + \frac{1}{\eta}D_\psi(\boldsymbol{\theta}_i, \boldsymbol{\theta}_i^{t_0})\}$$

$$\Leftrightarrow \boldsymbol{\theta}_i^{t_2} \in \underset{\boldsymbol{\theta}_i \in \mathbb{R}_{\geq 1}^{|A_i|}}{\arg\min}\{\langle -2\eta\boldsymbol{F}_i(\boldsymbol{\theta}^{t_1}), \boldsymbol{\theta}_i\rangle + \|\boldsymbol{\theta}_i\|_2^2 + \|\boldsymbol{\theta}_i^{t_0}\|_2^2 - 2\langle \boldsymbol{\theta}_i, \boldsymbol{\theta}_i^{t_0}\rangle\}$$

$$\Leftrightarrow \boldsymbol{\theta}_i^{t_2} \in \underset{\boldsymbol{\theta}_i \in \mathbb{R}_{\geq 1}^{|A_i|}}{\arg\min}\{\langle -2\eta\boldsymbol{F}_i(\boldsymbol{\theta}^{t_1}), \boldsymbol{\theta}_i\rangle + \|\boldsymbol{\theta}_i\|_2^2 - 2\langle \boldsymbol{\theta}_i, \boldsymbol{\theta}_i^{t_0}\rangle\}$$

$$\Leftrightarrow \boldsymbol{\theta}_i^{t_2} \in \underset{\boldsymbol{\theta}_i \in \mathbb{R}_{\geq 1}^{|A_i|}}{\arg\min}\{\langle -2\eta\boldsymbol{F}_i(\boldsymbol{\theta}^{t_1}) - 2\boldsymbol{\theta}_i^{t_0}, \boldsymbol{\theta}_i\rangle + \|\boldsymbol{\theta}_i\|_2^2\}$$

$$\Leftrightarrow \boldsymbol{\theta}_i^{t_2} \in \underset{\boldsymbol{\theta}_i \in \mathbb{R}_{\geq 1}^{|A_i|}}{\arg\min}\{\langle -2\eta\boldsymbol{F}_i(\boldsymbol{\theta}^{t_1}) - 2\boldsymbol{\theta}_i^{t_0}, \boldsymbol{\theta}_i\rangle + \|\boldsymbol{\theta}_i\|_2^2 + \|\eta\boldsymbol{F}_i(\boldsymbol{\theta}^{t_1}) + \boldsymbol{\theta}_i^{t_0}\|_2^2\}$$

$$\Leftrightarrow \boldsymbol{\theta}_i^{t_2} \in \underset{\boldsymbol{\theta}_i \in \mathbb{R}_{\geq 1}^{|A_i|}}{\arg\min}\|\eta\boldsymbol{F}_i(\boldsymbol{\theta}^{t_1}) + \boldsymbol{\theta}_i^{t_0} - \boldsymbol{\theta}_i\|_2^2.$$

Since $\boldsymbol{F}_i(\boldsymbol{\theta}^{t_1}) = \langle \frac{\boldsymbol{\theta}_i^{t_1}}{\|\boldsymbol{\theta}_i^{t_1}\|_1}, \boldsymbol{\ell}_i^{\boldsymbol{\theta}^{t_1}}\rangle \mathbf{1} - \boldsymbol{\ell}_i^{\boldsymbol{\theta}^{t_1}}$, we have

$$\boldsymbol{\theta}_i^{t_2} \in \underset{\boldsymbol{\theta}_i \in \mathbb{R}_{\geq 1}^{|A_i|}}{\arg\min}\{\langle -\boldsymbol{F}_i(\boldsymbol{\theta}^{t_1}), \boldsymbol{\theta}_i\rangle + \frac{1}{\eta}D_\psi(\boldsymbol{\theta}_i, \boldsymbol{\theta}_i^{t_0})\}$$

$$\Leftrightarrow \boldsymbol{\theta}_i^{t_2} \in \underset{\boldsymbol{\theta}_i \in \mathbb{R}_{\geq 1}^{|A_i|}}{\arg\min}\|\boldsymbol{\theta}_i^{t_0} + \eta\langle \frac{\boldsymbol{\theta}_i^{t_1}}{\|\boldsymbol{\theta}_i^{t_1}\|_1}, \boldsymbol{\ell}_i^{\boldsymbol{\theta}^{t_1}}\rangle \mathbf{1} - \eta\boldsymbol{\ell}_i^{\boldsymbol{\theta}^{t_1}} - \boldsymbol{\theta}_i\|_2^2. \tag{21}$$

Eq. (21) indicates getting the orthogonal projection of $\boldsymbol{\theta}_i^{t_0} + \eta\langle \frac{\boldsymbol{\theta}_i^{t_1}}{\|\boldsymbol{\theta}_i^{t_1}\|_1}, \boldsymbol{\ell}_i^{\boldsymbol{\theta}^{t_1}}\rangle \mathbf{1} - \eta\boldsymbol{\ell}_i^{\boldsymbol{\theta}^{t_1}}$ on $\mathbb{R}_{\geq 1}^{|A_i|}$.
As analyzed in Section K of Farina et al. (2023), $\forall \boldsymbol{x} \in \mathbb{R}^d$, projecting $\boldsymbol{x}$ to $\mathbb{R}_{\geq 1}^{|d|}$, if $\|[\boldsymbol{x}]_+\|_1 \geq 1$,
then the solution of the projection is $[\boldsymbol{x}]_+$. If $\|[\boldsymbol{x}]_+\|_1 < 1$, return the orthogonal projection of $\boldsymbol{x}$ on
simplex. Let $\boldsymbol{x} = \boldsymbol{\theta}_i^{t_0} + \eta\langle \frac{\boldsymbol{\theta}_i^{t_1}}{\|\boldsymbol{\theta}_i^{t_1}\|_1}, \boldsymbol{\ell}_i^{\boldsymbol{\theta}^{t_1}}\rangle \mathbf{1} - \eta\boldsymbol{\ell}_i^{\boldsymbol{\theta}^{t_1}}$, we have

$$\boldsymbol{\theta}_i^{t_2} = \begin{cases} [\boldsymbol{\theta}_i^{t_0} + \eta\langle \frac{\boldsymbol{\theta}_i^{t_1}}{\|\boldsymbol{\theta}_i^{t_1}\|_1}, \boldsymbol{\ell}_i^{\boldsymbol{\theta}^{t_1}}\rangle \mathbf{1} - \eta\boldsymbol{\ell}_i^{\boldsymbol{\theta}^{t_1}}]_+, & \text{if } \|[\boldsymbol{\theta}_i^{t_0} + \eta\langle \frac{\boldsymbol{\theta}_i^{t_1}}{\|\boldsymbol{\theta}_i^{t_1}\|_1}, \boldsymbol{\ell}_i^{\boldsymbol{\theta}^{t_1}}\rangle \mathbf{1} - \eta\boldsymbol{\ell}_i^{\boldsymbol{\theta}^{t_1}}]_+\|_1 \geq 1, \\ [\boldsymbol{\theta}_i^{t_0} + \eta\langle \frac{\boldsymbol{\theta}_i^{t_1}}{\|\boldsymbol{\theta}_i^{t_1}\|_1}, \boldsymbol{\ell}_i^{\boldsymbol{\theta}^{t_1}}\rangle \mathbf{1} - \eta\boldsymbol{\ell}_i^{\boldsymbol{\theta}^{t_1}} + \beta\mathbf{1}]_+, & \text{if } \|[\boldsymbol{\theta}_i^{t_0} + \eta\langle \frac{\boldsymbol{\theta}_i^{t_1}}{\|\boldsymbol{\theta}_i^{t_1}\|_1}, \boldsymbol{\ell}_i^{\boldsymbol{\theta}^{t_1}}\rangle \mathbf{1} - \eta\boldsymbol{\ell}_i^{\boldsymbol{\theta}^{t_1}}]_+\|_1 < 1, \end{cases}$$

where the top means if $\|[\boldsymbol{\theta}_i^{t_0} + \eta\langle \frac{\boldsymbol{\theta}_i^{t_1}}{\|\boldsymbol{\theta}_i^{t_1}\|_1}, \boldsymbol{\ell}_i^{\boldsymbol{\theta}^{t_1}}\rangle \mathbf{1} - \eta\boldsymbol{\ell}_i^{\boldsymbol{\theta}^{t_1}}]_+\|_1 \geq 1$, the
solution is $[\boldsymbol{\theta}_i^{t_0} + \eta\langle \frac{\boldsymbol{\theta}_i^{t_1}}{\|\boldsymbol{\theta}_i^{t_1}\|_1}, \boldsymbol{\ell}_i^{\boldsymbol{\theta}^{t_1}}\rangle \mathbf{1} - \eta\boldsymbol{\ell}_i^{\boldsymbol{\theta}^{t_1}}]_+$, and the bottom implies if
$\|[\boldsymbol{\theta}_i^{t_0} + \eta\langle \frac{\boldsymbol{\theta}_i^{t_1}}{\|\boldsymbol{\theta}_i^{t_1}\|_1}, \boldsymbol{\ell}_i^{\boldsymbol{\theta}^{t_1}}\rangle \mathbf{1} - \eta\boldsymbol{\ell}_i^{\boldsymbol{\theta}^{t_1}}]_+\|_1 < 1$, the solution is the orthogonal projection of
$\boldsymbol{\theta}_i^{t_0} + \eta\langle \frac{\boldsymbol{\theta}_i^{t_1}}{\|\boldsymbol{\theta}_i^{t_1}\|_1}, \boldsymbol{\ell}_i^{\boldsymbol{\theta}^{t_1}}\rangle \mathbf{1} - \eta\boldsymbol{\ell}_i^{\boldsymbol{\theta}^{t_1}}$ on simplex. Hence, as analyzed in Chen & Ye (2011),
the closed-form solution in the case where $\|[\boldsymbol{\theta}_i^{t_0} + \eta\langle \frac{\boldsymbol{\theta}_i^{t_1}}{\|\boldsymbol{\theta}_i^{t_1}\|_1}, \boldsymbol{\ell}_i^{\boldsymbol{\theta}^{t_1}}\rangle \mathbf{1} - \eta\boldsymbol{\ell}_i^{\boldsymbol{\theta}^{t_1}}]_+\|_1 < 1$ is
$[\boldsymbol{\theta}_i^{t_0} + \eta\langle \frac{\boldsymbol{\theta}_i^{t_1}}{\|\boldsymbol{\theta}_i^{t_1}\|_1}, \boldsymbol{\ell}_i^{\boldsymbol{\theta}^{t_1}}\rangle \mathbf{1} - \eta\boldsymbol{\ell}_i^{\boldsymbol{\theta}^{t_1}} + \beta\mathbf{1}]_+$, where $\beta$ exists and is unique to ensure $\|\boldsymbol{\theta}_i^{t_2}\|_1 = 1$ (see
the reason why $\beta$ is a unique constant in the proof of Theorem 2.2 of Chen & Ye (2011)).

## B  PROOF OF THEOREM 4.2

**Lemma B.1.** *(Proof is in Appendix B.1) Let $\boldsymbol{x}^* \in \boldsymbol{\mathcal{X}}^*$ and assume all players follow the update
rule of SExRM$^+$, then for every iteration $t \geq 1$, it holds that $\|\boldsymbol{\theta}^{t+1} - \boldsymbol{x}^*\|_2^2 \leq \|\boldsymbol{\theta}^t - \boldsymbol{x}^*\|_2^2 - (1 - \eta D L_u)\left(\|\boldsymbol{\theta}^{t+\frac{1}{2}} - \boldsymbol{\theta}^t\|_2^2 + \|\boldsymbol{\theta}^{t+1} - \boldsymbol{\theta}^{t+\frac{1}{2}}\|_2^2\right)$, where $D = \max_{i\in\mathcal{N}}|A_i|$ and $L_u = \sqrt{2P^2 + 4L^2}$.*

From Lemma B.1, we have

$$\|\boldsymbol{\theta}^{t+1} - \boldsymbol{x}^*\|_2^2 - \|\boldsymbol{\theta}^t - \boldsymbol{x}^*\|_2^2 \leq -(1 - \eta D L_u)\left(\|\boldsymbol{\theta}^{t+\frac{1}{2}} - \boldsymbol{\theta}^t\|_2^2 + \|\boldsymbol{\theta}^{t+1} - \boldsymbol{\theta}^{t+\frac{1}{2}}\|_2^2\right). \tag{22}$$

Assume $\left(\|\boldsymbol{\theta}^{t+\frac{1}{2}} - \boldsymbol{\theta}^t\|_2^2 + \|\boldsymbol{\theta}^{t+1} - \boldsymbol{\theta}^{t+\frac{1}{2}}\|_2^2\right)$ do not converge to $0$. Then, from Eq. (22), we have

$$\|\boldsymbol{\theta}^{T+1} - \boldsymbol{x}^*\|_2^2 \leq \|\boldsymbol{\theta}^1 - \boldsymbol{x}^*\|_2^2 - \sum_{t=1}^T (1 - \eta D L_u)\left(\|\boldsymbol{\theta}^{t+\frac{1}{2}} - \boldsymbol{\theta}^t\|_2^2 + \|\boldsymbol{\theta}^{t+1} - \boldsymbol{\theta}^{t+\frac{1}{2}}\|_2^2\right)$$

In addition, since $\eta < \frac{1}{DL_u}$, we have $(1 - \eta D L_u) > 0$. Therefore, as $T \to \infty$, $\|\boldsymbol{\theta}^{T+1} - \boldsymbol{x}^*\|_2^2 \leq \|\boldsymbol{\theta}^1 - \boldsymbol{x}^*\|_2^2 - \sum_{t=1}^T \left(\|\boldsymbol{\theta}^{t+\frac{1}{2}} - \boldsymbol{\theta}^t\|_2^2 + \|\boldsymbol{\theta}^{t+1} - \boldsymbol{\theta}^{t+\frac{1}{2}}\|_2^2\right) = -\infty$, which contracts that $\|\boldsymbol{\theta}^{T+1} - \boldsymbol{x}^*\|_2^2 \geq 0$. Therefore, we have $\left(\|\boldsymbol{\theta}^{t+\frac{1}{2}} - \boldsymbol{\theta}^t\|_2^2 + \|\boldsymbol{\theta}^{t+1} - \boldsymbol{\theta}^{t+\frac{1}{2}}\|_2^2\right)$ as $t \to \infty$.

In addition, from $\eta < \frac{1}{DL_u}$ and Eq. (22), we have

$$\sum_{t=1}^T \left(\|\boldsymbol{\theta}^{t+\frac{1}{2}} - \boldsymbol{\theta}^t\|_2^2 + \|\boldsymbol{\theta}^{t+1} - \boldsymbol{\theta}^{t+\frac{1}{2}}\|_2^2\right) \leq \frac{\|\boldsymbol{\theta}^1 - \boldsymbol{x}^*\|_2^2 - \|\boldsymbol{\theta}^{T+1} - \boldsymbol{x}^*\|_2^2}{1 - \eta D L_u} \leq C,$$

where $C$ is a constant which depends on $\boldsymbol{\theta}^1$, $\boldsymbol{x}^*$, $\eta$, $D$, and $L_u$. Therefore, we get

$$T \min_{t \in T} \left(\|\boldsymbol{\theta}^{t+\frac{1}{2}} - \boldsymbol{\theta}^t\|_2^2 + \|\boldsymbol{\theta}^{t+1} - \boldsymbol{\theta}^{t+\frac{1}{2}}\|_2^2\right) \leq \sum_{t=1}^T \left(\|\boldsymbol{\theta}^{t+\frac{1}{2}} - \boldsymbol{\theta}^t\|_2^2 + \|\boldsymbol{\theta}^{t+1} - \boldsymbol{\theta}^{t+\frac{1}{2}}\|_2^2\right) \leq C,$$

which implies

$$\min_{t \in T} \left(\|\boldsymbol{\theta}^{t+\frac{1}{2}} - \boldsymbol{\theta}^t\|_2^2 + \|\boldsymbol{\theta}^{t+1} - \boldsymbol{\theta}^{t+\frac{1}{2}}\|_2^2\right) \leq \frac{C}{T}.$$

## B.1 PROOF OF LEMMA B.1

**Lemma B.2.** *(Proof is in Appendix B.2) Assume all players follow the update rule of SExRM$^+$, then for any $\boldsymbol{\theta} \in \mathbb{R}_{\geq 1}^{|\boldsymbol{\mathcal{X}}|}$, we have*

$$D_\psi(\boldsymbol{\theta}, \boldsymbol{\theta}^{t+1}) - D_\psi(\boldsymbol{\theta}, \boldsymbol{\theta}^t)$$
$$\leq -\eta\langle \boldsymbol{F}(\boldsymbol{\theta}^{t+\frac{1}{2}}), \boldsymbol{\theta}\rangle + \eta\langle \boldsymbol{F}(\boldsymbol{\theta}^{t+\frac{1}{2}}) - \boldsymbol{F}(\boldsymbol{\theta}^t), \boldsymbol{\theta}^{t+1} - \boldsymbol{\theta}^{t+\frac{1}{2}}\rangle - D_\psi(\boldsymbol{\theta}^{t+1}, \boldsymbol{\theta}^{t+\frac{1}{2}}) - D_\psi(\boldsymbol{\theta}^{t+\frac{1}{2}}, \boldsymbol{\theta}^t).$$

Substituting $\boldsymbol{\theta} = \boldsymbol{x}^* \in \boldsymbol{\mathcal{X}}^*$ into Lemma B.2, we get

$$D_\psi(\boldsymbol{x}^*, \boldsymbol{\theta}^{t+1}) - D_\psi(\boldsymbol{x}^*, \boldsymbol{\theta}^t)$$
$$\leq -\eta\langle \boldsymbol{F}(\boldsymbol{\theta}^{t+\frac{1}{2}}), \boldsymbol{x}^*\rangle - D_\psi(\boldsymbol{\theta}^{t+1}, \boldsymbol{\theta}^{t+\frac{1}{2}}) - D_\psi(\boldsymbol{\theta}^{t+\frac{1}{2}}, \boldsymbol{\theta}^t) + \eta\langle \boldsymbol{F}(\boldsymbol{\theta}^{t+\frac{1}{2}}) - \boldsymbol{F}(\boldsymbol{\theta}^t), \boldsymbol{\theta}^{t+1} - \boldsymbol{\theta}^{t+\frac{1}{2}}\rangle. \tag{23}$$

For the first term of the right-hand side of Eq. (23), we have

$$-\eta \sum_{i \in \mathcal{N}} \langle \boldsymbol{F}_i(\boldsymbol{\theta}^{t+\frac{1}{2}}), \boldsymbol{x}_i^*\rangle = -\eta \sum_{i \in \mathcal{N}} \langle \boldsymbol{\ell}_i^{t+\frac{1}{2}}, \boldsymbol{x}_i^{t+\frac{1}{2}} - \boldsymbol{x}_i^*\rangle = -\eta\langle \boldsymbol{\ell}^{t+\frac{1}{2}}, \boldsymbol{x}^{t+\frac{1}{2}} - \boldsymbol{x}^*\rangle \leq 0. \tag{24}$$

where the last line is from the definition of NE (Section 2.1). For the fourth term of the right-hand side of Eq. (23), we have

$$\eta\langle \boldsymbol{F}(\boldsymbol{\theta}^{t+\frac{1}{2}}) - \boldsymbol{F}(\boldsymbol{\theta}^t), \boldsymbol{\theta}^{t+1} - \boldsymbol{\theta}^{t+\frac{1}{2}}\rangle \leq \eta\|\boldsymbol{F}(\boldsymbol{\theta}^{t+\frac{1}{2}}) - \boldsymbol{F}(\boldsymbol{\theta}^t)\|_2\|\boldsymbol{\theta}^{t+1} - \boldsymbol{\theta}^{t+\frac{1}{2}}\|_2$$
$$\leq \eta D L_u\|\boldsymbol{\theta}^{t+\frac{1}{2}} - \boldsymbol{\theta}^t\|_2\|\boldsymbol{\theta}^{t+1} - \boldsymbol{\theta}^{t+\frac{1}{2}}\|_2, \tag{25}$$

where the last line is from Lemma 5.2 of Farina et al. (2023) ($\|\boldsymbol{F}(\boldsymbol{\theta}) - \boldsymbol{F}(\boldsymbol{\theta}')\|_2 \leq D L_u\|\boldsymbol{\theta} - \boldsymbol{\theta}'\|_2, \forall \boldsymbol{\theta}, \boldsymbol{\theta}' \in \mathbb{R}_{\geq 1}^{|\boldsymbol{\mathcal{X}}|}$, where $D, L_u$ are in Theorem 4.1). Combining Eq. (23), (24), and (25), we get

$$D_\psi(\boldsymbol{x}^*, \boldsymbol{\theta}^{t+1}) - D_\psi(\boldsymbol{x}^*, \boldsymbol{\theta}^t)$$
$$\leq -D_\psi(\boldsymbol{\theta}^{t+1}, \boldsymbol{\theta}^{t+\frac{1}{2}}) - D_\psi(\boldsymbol{\theta}^{t+\frac{1}{2}}, \boldsymbol{\theta}^t) + \eta D L_u\|\boldsymbol{\theta}^{t+\frac{1}{2}} - \boldsymbol{\theta}^t\|_2\|\boldsymbol{\theta}^{t+1} - \boldsymbol{\theta}^{t+\frac{1}{2}}\|_2$$
$$\leq -D_\psi(\boldsymbol{\theta}^{t+1}, \boldsymbol{\theta}^{t+\frac{1}{2}}) - D_\psi(\boldsymbol{\theta}^{t+\frac{1}{2}}, \boldsymbol{\theta}^t) + \eta D L_u\left(D_\psi(\boldsymbol{\theta}^{t+\frac{1}{2}}, \boldsymbol{\theta}^t) + D_\psi(\boldsymbol{\theta}^{t+1}, \boldsymbol{\theta}^{t+\frac{1}{2}})\right)$$
$$\leq -(1 - \eta D L_u)D_\psi\left(D_\psi(\boldsymbol{\theta}^{t+\frac{1}{2}}, \boldsymbol{\theta}^t) + D_\psi(\boldsymbol{\theta}^{t+1}, \boldsymbol{\theta}^{t+\frac{1}{2}})\right).$$

where the second inequality is from $\forall a, b \in \mathbb{R}, ab \leq pa^2/2 + b^2/2p, \forall p > 0$ (in this case, $a = \|\boldsymbol{\theta}^{t+\frac{1}{2}} - \boldsymbol{\theta}^t\|_2, b = \|\boldsymbol{\theta}^{t+1} - \boldsymbol{\theta}^{t+\frac{1}{2}}\|_2$, $p = 1$) and $D_\psi(\boldsymbol{a}, \boldsymbol{b}) = \|\boldsymbol{a} - \boldsymbol{b}\|_2^2/2$, $\forall \boldsymbol{a}, \boldsymbol{b} \in \mathbb{R}^d$ if $\psi(\cdot)$ is the quadratic regularizer.

### B.2 Proof of Lemma B.2

To prove Lemma B.2, we first introduce the following folk theorem (we drop the terms involved $x$ in Eq. (2) and Eq. (3) since they are not used in the following proofs).

**Theorem B.3.** *The Update rule of SExRM$^+$ can be written as*

$$\boldsymbol{\theta}^{t+\frac{1}{2}} \in \underset{\boldsymbol{\theta} \in \times_{i \in \mathcal{N}} \mathbb{R}_{\geq 1}^{|A_i|}}{\arg\min} \{\langle -\boldsymbol{F}(\boldsymbol{\theta}^t), \boldsymbol{\theta}\rangle + \frac{1}{\eta} D_\psi(\boldsymbol{\theta}, \boldsymbol{\theta}^t)\},$$

$$\boldsymbol{\theta}^{t+1} \in \underset{\boldsymbol{\theta} \in \times_{i \in \mathcal{N}} \mathbb{R}_{\geq 1}^{|A_i|}}{\arg\min} \{\langle -\boldsymbol{F}(\boldsymbol{\theta}^{t+\frac{1}{2}}), \boldsymbol{\theta}\rangle + \frac{1}{\eta} D_\psi(\boldsymbol{\theta}, \boldsymbol{\theta}^t)\}, \quad (26)$$

*and the update rule of SPRM$^+$ can be written as*

$$\boldsymbol{\theta}^{t+\frac{1}{2}} \in \underset{\boldsymbol{\theta} \in \times_{i \in \mathcal{N}} \mathbb{R}_{\geq 1}^{|A_i|}}{\arg\min} \{\langle -\boldsymbol{F}(\boldsymbol{\theta}^{t-\frac{1}{2}}), \boldsymbol{\theta}\rangle + \frac{1}{\eta} D_\psi(\boldsymbol{\theta}, \boldsymbol{\theta}^t)\},$$

$$\boldsymbol{\theta}^{t+1} \in \underset{\boldsymbol{\theta} \in \times_{i \in \mathcal{N}} \mathbb{R}_{\geq 1}^{|A_i|}}{\arg\min} \{\langle -\boldsymbol{F}(\boldsymbol{\theta}^{t+\frac{1}{2}}), \boldsymbol{\theta}\rangle + \frac{1}{\eta} D_\psi(\boldsymbol{\theta}, \boldsymbol{\theta}^t)\}, \quad (27)$$

*where $\eta > 0$ is the learning rate.*

Considering Eq. (26), and using Lemma D.2 with $\boldsymbol{a} = \boldsymbol{\theta}^t$, $\boldsymbol{a}' = \boldsymbol{\theta}^{t+1}$, $\boldsymbol{a}^* = \boldsymbol{\theta}$ and $\boldsymbol{g} = -\eta \boldsymbol{F}(\boldsymbol{\theta}^{t+\frac{1}{2}})$ (in this case, $\mathcal{A}$ is $\times_{i \in \mathcal{N}} \mathbb{R}_{\geq 1}^{|A_i|}$), we have

$$\eta\langle -\boldsymbol{F}(\boldsymbol{\theta}^{t+\frac{1}{2}}), \boldsymbol{\theta}^{t+1} - \boldsymbol{\theta}\rangle \leq D_\psi(\boldsymbol{\theta}, \boldsymbol{\theta}^t) - D_\psi(\boldsymbol{\theta}, \boldsymbol{\theta}^{t+1}) - D_\psi(\boldsymbol{\theta}^{t+1}, \boldsymbol{\theta}^t). \quad (28)$$

Similarly, with $\boldsymbol{a} = \boldsymbol{\theta}^t$, $\boldsymbol{a}' = \boldsymbol{\theta}^{t+\frac{1}{2}}$, $\boldsymbol{a}^* = \boldsymbol{\theta}^{t+1}$ and $\boldsymbol{g} = -\eta \boldsymbol{F}(\boldsymbol{\theta}^t)$, we get

$$\eta\langle -\boldsymbol{F}(\boldsymbol{\theta}^t), \boldsymbol{\theta}^{t+\frac{1}{2}} - \boldsymbol{\theta}^{t+1}\rangle \leq D_\psi(\boldsymbol{\theta}^{t+1}, \boldsymbol{\theta}^t) - D_\psi(\boldsymbol{\theta}^{t+1}, \boldsymbol{\theta}^{t+\frac{1}{2}}) - D_\psi(\boldsymbol{\theta}^{t+\frac{1}{2}}, \boldsymbol{\theta}^t). \quad (29)$$

Summing up Eq. (28) and (29), and adding $\eta\langle \boldsymbol{F}(\boldsymbol{\theta}^{t+\frac{1}{2}}) - \boldsymbol{F}(\boldsymbol{\theta}^t), \boldsymbol{\theta}^{t+1} - \boldsymbol{\theta}^{t+\frac{1}{2}}\rangle$ to both sides, we get

$$\eta\langle -\boldsymbol{F}(\boldsymbol{\theta}^{t+\frac{1}{2}}), \boldsymbol{\theta}^{t+\frac{1}{2}} - \boldsymbol{\theta}\rangle$$
$$\leq D_\psi(\boldsymbol{\theta}, \boldsymbol{\theta}^t) - D_\psi(\boldsymbol{\theta}, \boldsymbol{\theta}^{t+1}) - D_\psi(\boldsymbol{\theta}^{t+1}, \boldsymbol{\theta}^{t+\frac{1}{2}}) - D_\psi(\boldsymbol{\theta}^{t+\frac{1}{2}}, \boldsymbol{\theta}^t) + \eta\langle \boldsymbol{F}(\boldsymbol{\theta}^{t+\frac{1}{2}}) - \boldsymbol{F}(\boldsymbol{\theta}^t), \boldsymbol{\theta}^{t+1} - \boldsymbol{\theta}^{t+\frac{1}{2}}\rangle.$$

Arranging the terms, we have

$$D_\psi(\boldsymbol{\theta}, \boldsymbol{\theta}^{t+1}) - D_\psi(\boldsymbol{\theta}, \boldsymbol{\theta}^t)$$
$$\leq \eta\langle \boldsymbol{F}(\boldsymbol{\theta}^{t+\frac{1}{2}}), \boldsymbol{\theta}^{t+\frac{1}{2}} - \boldsymbol{\theta}\rangle + \eta\langle \boldsymbol{F}(\boldsymbol{\theta}^{t+\frac{1}{2}}) - \boldsymbol{F}(\boldsymbol{\theta}^t), \boldsymbol{\theta}^{t+1} - \boldsymbol{\theta}^{t+\frac{1}{2}}\rangle - D_\psi(\boldsymbol{\theta}^{t+1}, \boldsymbol{\theta}^{t+\frac{1}{2}}) - D_\psi(\boldsymbol{\theta}^{t+\frac{1}{2}}, \boldsymbol{\theta}^t)$$
$$\leq -\eta\langle \boldsymbol{F}(\boldsymbol{\theta}^{t+\frac{1}{2}}), \boldsymbol{\theta}\rangle + \eta\langle \boldsymbol{F}(\boldsymbol{\theta}^{t+\frac{1}{2}}) - \boldsymbol{F}(\boldsymbol{\theta}^t), \boldsymbol{\theta}^{t+1} - \boldsymbol{\theta}^{t+\frac{1}{2}}\rangle - D_\psi(\boldsymbol{\theta}^{t+1}, \boldsymbol{\theta}^{t+\frac{1}{2}}) - D_\psi(\boldsymbol{\theta}^{t+\frac{1}{2}}, \boldsymbol{\theta}^t),$$

where the last line comes from $\langle \boldsymbol{F}(\boldsymbol{\theta}^{t+\frac{1}{2}}), \boldsymbol{\theta}^{t+\frac{1}{2}}\rangle = \sum_{i \in \mathcal{N}}\langle \boldsymbol{F}_i(\boldsymbol{\theta}^{t+\frac{1}{2}}), \boldsymbol{\theta}_i^{t+\frac{1}{2}}\rangle = \sum_{i \in \mathcal{N}}\langle\langle \boldsymbol{\ell}_i^{t+\frac{1}{2}}, \boldsymbol{x}_i^{t+\frac{1}{2}}\rangle\mathbf{1} - \boldsymbol{\ell}_i^{t+\frac{1}{2}}, \boldsymbol{\theta}_i^{t+\frac{1}{2}}\rangle = \sum_{i \in \mathcal{N}}\langle \boldsymbol{\ell}_i^{t+\frac{1}{2}}, \boldsymbol{x}_i^{t+\frac{1}{2}}\rangle\langle\mathbf{1}, \boldsymbol{\theta}_i^{t+\frac{1}{2}}\rangle - \langle \boldsymbol{\ell}_i^{t+\frac{1}{2}}, \boldsymbol{\theta}_i^{t+\frac{1}{2}}\rangle = \sum_{i \in \mathcal{N}}\langle \boldsymbol{\ell}_i^{t+\frac{1}{2}}, \frac{\boldsymbol{\theta}_i^{t+\frac{1}{2}}}{\|\boldsymbol{\theta}_i^{t+\frac{1}{2}}\|_1}\rangle\|\boldsymbol{\theta}_i^{t+\frac{1}{2}}\|_1 - \langle \boldsymbol{\ell}_i^{t+\frac{1}{2}}, \boldsymbol{\theta}_i^{t+\frac{1}{2}}\rangle = 0$. It completes the proof.

## C Proof of Theorem 4.3

**Lemma C.1.** *(Proof is in Appendix C.1) Let $\boldsymbol{x}^* \in \mathcal{X}^*$ and $0 < \eta < \frac{1}{8DL_u}$, then for every iteration $t \geq 1$, it holds that*

$$\|\boldsymbol{\theta}^{t+1} - \boldsymbol{x}^*\|_2^2 + \frac{1}{16}\|\boldsymbol{\theta}^{t+1} - \boldsymbol{\theta}^{t+\frac{1}{2}}\|_2^2 \leq \|\boldsymbol{\theta}^t - \boldsymbol{x}^*\|_2^2 + \frac{1}{16}\|\boldsymbol{\theta}^t - \boldsymbol{\theta}^{t-\frac{1}{2}}\|_2^2 - \frac{15}{16}(\|\boldsymbol{\theta}^{t+1} - \boldsymbol{\theta}^{t+\frac{1}{2}}\|_2^2 + \|\boldsymbol{\theta}^t - \boldsymbol{\theta}^{t+\frac{1}{2}}\|_2^2).$$

From Lemma C.1, we have

$$\|\boldsymbol{\theta}^{t+1} - \boldsymbol{x}^*\|_2^2 + \frac{1}{16}\|\boldsymbol{\theta}^{t+1} - \boldsymbol{\theta}^{t+\frac{1}{2}}\|_2^2 \leq \|\boldsymbol{\theta}^t - \boldsymbol{\theta}^*\|_2^2 +$$
$$\frac{1}{16}\|\boldsymbol{\theta}^t - \boldsymbol{\theta}^{t-\frac{1}{2}}\|_2^2 - \frac{15}{16}(\|\boldsymbol{\theta}^{t+1} - \boldsymbol{\theta}^{t+\frac{1}{2}}\|_2^2 + \|\boldsymbol{\theta}^t - \boldsymbol{\theta}^{t+\frac{1}{2}}\|_2^2). \quad (30)$$

Assume $\|\boldsymbol{\theta}^{t+1} - \boldsymbol{\theta}^{t+\frac{1}{2}}\|_2^2 + \|\boldsymbol{\theta}^t - \boldsymbol{\theta}^{t+\frac{1}{2}}\|_2^2$ do not converge to 0. Then, from Eq. (30), we have

$$\|\boldsymbol{\theta}^{T+1} - \boldsymbol{x}^*\|_2^2 + \frac{1}{16}\|\boldsymbol{\theta}^{T+1} - \boldsymbol{\theta}^{T+\frac{1}{2}}\|_2^2$$

$$\leq \|\boldsymbol{\theta}^1 - \boldsymbol{x}^*\|_2^2 + \|\frac{1}{16}\|\boldsymbol{\theta}^1 - \boldsymbol{\theta}^{1-\frac{1}{2}}\|_2^2 - \frac{15}{16}\sum_{t=1}^T (\|\boldsymbol{\theta}^{t+1} - \boldsymbol{\theta}^{t+\frac{1}{2}}\|_2^2 + \|\boldsymbol{\theta}^t - \boldsymbol{\theta}^{t+\frac{1}{2}}\|_2^2).$$

Therefore, as $T \to \infty$, $\|\boldsymbol{\theta}^{T+1} - \boldsymbol{x}^*\|_2^2 + \frac{1}{16}\|\boldsymbol{\theta}^{T+1} - \boldsymbol{\theta}^{T+\frac{1}{2}}\|_2^2 \leq \|\boldsymbol{\theta}^1 - \boldsymbol{\theta}^*\|_2^2 + \|\frac{1}{16}\|\boldsymbol{\theta}^1 - \boldsymbol{\theta}^{1-\frac{1}{2}}\|_2^2 - \frac{15}{16}\sum_{t=1}^T (\|\boldsymbol{\theta}^{t+1} - \boldsymbol{\theta}^{t+\frac{1}{2}}\|_2^2 + \|\boldsymbol{\theta}^t - \boldsymbol{\theta}^{t+\frac{1}{2}}\|_2^2) = -\infty$, which contracts that $\|\boldsymbol{\theta}^{T+1} - \boldsymbol{x}^*\|_2^2 + \frac{1}{16}\|\boldsymbol{\theta}^{T+1} - \boldsymbol{\theta}^{T+\frac{1}{2}}\|_2^2 \geq 0$. Therefore, we have $\|\boldsymbol{\theta}^{t+1} - \boldsymbol{\theta}^{t+\frac{1}{2}}\|_2^2 + \|\boldsymbol{\theta}^t - \boldsymbol{\theta}^{t+\frac{1}{2}}\|_2^2 \to 0$ as $t \to \infty$, which implies $\|\boldsymbol{\theta}^{t+1} - \boldsymbol{\theta}^{t+\frac{1}{2}}\|_2^2 \to 0$ and $\|\boldsymbol{\theta}^t - \boldsymbol{\theta}^{t+\frac{1}{2}}\|_2^2 \to 0$ as $t \to \infty$. It completes the proof.

In addition, from $\eta < \frac{1}{8DL_u}$ and Eq. (30), we have

$$\sum_{t=1}^T \left( \|\boldsymbol{\theta}^{t+\frac{1}{2}} - \boldsymbol{\theta}^t\|_2^2 + \|\boldsymbol{\theta}^{t+1} - \boldsymbol{\theta}^{t+\frac{1}{2}}\|_2^2 \right) \leq C,$$

where $C$ is a constant which depends on $\boldsymbol{\theta}^1$, $\boldsymbol{\theta}^{\frac{1}{2}}$, $\boldsymbol{x}^*$, $\eta$, $D$, and $L_u$. Therefore, we get

$$T \min_{t \in T} \left( \|\boldsymbol{\theta}^{t+\frac{1}{2}} - \boldsymbol{\theta}^t\|_2^2 + \|\boldsymbol{\theta}^{t+1} - \boldsymbol{\theta}^{t+\frac{1}{2}}\|_2^2 \right) \leq \sum_{t=1}^T \left( \|\boldsymbol{\theta}^{t+\frac{1}{2}} - \boldsymbol{\theta}^t\|_2^2 + \|\boldsymbol{\theta}^{t+1} - \boldsymbol{\theta}^{t+\frac{1}{2}}\|_2^2 \right) \leq C,$$

which implies

$$\min_{t \in T} \left( \|\boldsymbol{\theta}^{t+\frac{1}{2}} - \boldsymbol{\theta}^t\|_2^2 + \|\boldsymbol{\theta}^{t+1} - \boldsymbol{\theta}^{t+\frac{1}{2}}\|_2^2 \right) \leq \frac{C}{T}.$$

### C.1 PROOF OF LEMMA C.1

**Lemma C.2.** *(Proof is in Appendix C.2) Assume all players follow the update rule of SPRM$^+$, then for any $\boldsymbol{\theta} \in \mathbb{R}_{\geq 1}^{|\boldsymbol{\mathcal{X}}|}$, we have*

$$D_\psi(\boldsymbol{\theta}, \boldsymbol{\theta}^{t+1}) - D_\psi(\boldsymbol{\theta}, \boldsymbol{\theta}^t)$$

$$\leq -\eta \langle \boldsymbol{F}(\boldsymbol{\theta}^{t+\frac{1}{2}}), \boldsymbol{\theta} \rangle + \eta \langle \boldsymbol{F}(\boldsymbol{\theta}^{t+\frac{1}{2}}) - \boldsymbol{F}(\boldsymbol{\theta}^{t-\frac{1}{2}}), \boldsymbol{\theta}^{t+1} - \boldsymbol{\theta}^{t+\frac{1}{2}} \rangle - D_\psi(\boldsymbol{\theta}^{t+1}, \boldsymbol{\theta}^{t+\frac{1}{2}}) - D_\psi(\boldsymbol{\theta}^{t+\frac{1}{2}}, \boldsymbol{\theta}^t).$$

Substituting $\boldsymbol{\theta} = \boldsymbol{x}^* \in \boldsymbol{\mathcal{X}}^*$ into Lemma C.2, we get

$$D_\psi(\boldsymbol{x}^*, \boldsymbol{\theta}^{t+1}) - D_\psi(\boldsymbol{x}^*, \boldsymbol{\theta}^t)$$

$$\leq -\eta \langle \boldsymbol{F}(\boldsymbol{\theta}^{t+\frac{1}{2}}), \boldsymbol{x}^* \rangle - D_\psi(\boldsymbol{\theta}^{t+1}, \boldsymbol{\theta}^{t+\frac{1}{2}}) - D_\psi(\boldsymbol{\theta}^{t+\frac{1}{2}}, \boldsymbol{\theta}^t) + \eta \langle \boldsymbol{F}(\boldsymbol{\theta}^{t+\frac{1}{2}}) - \boldsymbol{F}(\boldsymbol{\theta}^{t-\frac{1}{2}}), \boldsymbol{\theta}^{t+1} - \boldsymbol{\theta}^{t+\frac{1}{2}} \rangle. \tag{31}$$

According to the analysis in Section 4, for the first term of the right-hand side of Eq. (31), we have

$$-\eta \sum_{i \in \mathcal{N}} \langle \boldsymbol{F}_i(\boldsymbol{\theta}^{t+\frac{1}{2}}), \boldsymbol{x}_i^* \rangle \leq 0.$$

To simply the fourth term of the right-hand side of Eq. (31), we first introduce Lemma C.3, whose proof is in Appendix C.3.

**Lemma C.3.** *Assume all players follow the update rule of SPRM$^+$, then we have*

$$\|\boldsymbol{\theta}^{t+1} - \boldsymbol{\theta}^{t+\frac{1}{2}}\|_2 \leq \eta \|\boldsymbol{F}(\boldsymbol{\theta}^{t+\frac{1}{2}}) - \boldsymbol{F}(\boldsymbol{\theta}^{t-\frac{1}{2}})\|_2.$$

Therefore, for the fourth term of the right-hand side of Eq. (31), we have

$$\eta \langle \boldsymbol{F}(\boldsymbol{\theta}^{t+\frac{1}{2}}) - \boldsymbol{F}(\boldsymbol{\theta}^{t-\frac{1}{2}}), \boldsymbol{\theta}^{t+1} - \boldsymbol{\theta}^{t+\frac{1}{2}} \rangle$$

$$\leq \eta \|\boldsymbol{F}(\boldsymbol{\theta}^{t+\frac{1}{2}}) - \boldsymbol{F}(\boldsymbol{\theta}^{t-\frac{1}{2}})\|_2 \|\boldsymbol{\theta}^{t+1} - \boldsymbol{\theta}^{t+\frac{1}{2}}\|_2$$

$$\leq \eta^2 \|\boldsymbol{F}(\boldsymbol{\theta}^{t+\frac{1}{2}}) - \boldsymbol{F}(\boldsymbol{\theta}^{t-\frac{1}{2}})\|_2^2.$$

where the third line is from Lemma C.3. Then, from Lemma 5.2 of Farina et al. (2023) ($\|\boldsymbol{F}(\boldsymbol{\theta}) - \boldsymbol{F}(\boldsymbol{\theta}')\|_2 \leq DL_u\|\boldsymbol{\theta} - \boldsymbol{\theta}'\|_2, \forall \boldsymbol{\theta}, \boldsymbol{\theta}' \in \mathbb{R}_{\geq 1}^{|\boldsymbol{\mathcal{X}}|}$, where $D, L_u$ are defined in Theorem 4.1) and the choice of $\eta$, we have

$$\eta^2 \|\boldsymbol{F}(\boldsymbol{\theta}^{t+\frac{1}{2}}) - \boldsymbol{F}(\boldsymbol{\theta}^{t-\frac{1}{2}})\|_2^2 \leq \eta^2 D^2 L_u^2 \|\boldsymbol{\theta}^{t+\frac{1}{2}} - \boldsymbol{\theta}^{t-\frac{1}{2}}\|_2^2 \leq \frac{1}{64}\|\boldsymbol{\theta}^{t+\frac{1}{2}} - \boldsymbol{\theta}^{t-\frac{1}{2}}\|_2^2.$$

Continuing from Eq. (31), we then have

$$D_\psi(\boldsymbol{x}^*, \boldsymbol{\theta}^{t+1}) - D_\psi(\boldsymbol{x}^*, \boldsymbol{\theta}^t)$$

$$\leq - D_\psi(\boldsymbol{\theta}^{t+1}, \boldsymbol{\theta}^{t+\frac{1}{2}}) - D_\psi(\boldsymbol{\theta}^{t+\frac{1}{2}}, \boldsymbol{\theta}^t) + \frac{1}{64}\|\boldsymbol{\theta}^{t+\frac{1}{2}} - \boldsymbol{\theta}^{t-\frac{1}{2}}\|_2^2$$

$$\leq - D_\psi(\boldsymbol{\theta}^{t+1}, \boldsymbol{\theta}^{t+\frac{1}{2}}) - D_\psi(\boldsymbol{\theta}^{t+\frac{1}{2}}, \boldsymbol{\theta}^t) + \frac{1}{32}\|\boldsymbol{\theta}^{t+\frac{1}{2}} - \boldsymbol{\theta}^t\|_2^2 + \frac{1}{32}\|\boldsymbol{\theta}^t - \boldsymbol{\theta}^{t-\frac{1}{2}}\|_2^2$$

$$\Leftrightarrow \|\boldsymbol{\theta}^{t+1} - \boldsymbol{x}^*\|_2^2 + \frac{1}{16}\|\boldsymbol{\theta}^{t+1} - \boldsymbol{\theta}^{t+\frac{1}{2}}\|_2^2$$

$$\leq \|\boldsymbol{\theta}^t - \boldsymbol{x}^*\|_2^2 + \frac{1}{16}\|\boldsymbol{\theta}^t - \boldsymbol{\theta}^{t-\frac{1}{2}}\|_2^2 - \frac{15}{16}(\|\boldsymbol{\theta}^{t+1} - \boldsymbol{\theta}^{t+\frac{1}{2}}\|_2^2 + \|\boldsymbol{\theta}^t - \boldsymbol{\theta}^{t+\frac{1}{2}}\|_2^2),$$

where the last line is from $D_\psi(\boldsymbol{a}, \boldsymbol{b}) = \|\boldsymbol{a} - \boldsymbol{b}\|_2^2/2$.

## C.2 PROOF OF LEMMA C.2

Considering Eq. (27), and using Lemma D.2 with $\boldsymbol{a} = \boldsymbol{\theta}^t$, $\boldsymbol{a}' = \boldsymbol{\theta}^{t+1}$, $\boldsymbol{a}^* = \boldsymbol{\theta}$ and $\boldsymbol{g} = -\eta\boldsymbol{F}(\boldsymbol{\theta}^{t+\frac{1}{2}})$ (in this case, $\boldsymbol{\mathcal{A}}$ is $\times_{i\in\mathcal{N}}\mathbb{R}_{\geq 1}^{|A_i|}$), we have

$$\eta\langle -\boldsymbol{F}(\boldsymbol{\theta}^{t+\frac{1}{2}}), \boldsymbol{\theta}^{t+1} - \boldsymbol{\theta}\rangle \leq D_\psi(\boldsymbol{\theta}, \boldsymbol{\theta}^t) - D_\psi(\boldsymbol{\theta}, \boldsymbol{\theta}^{t+1}) - D_\psi(\boldsymbol{\theta}^{t+1}, \boldsymbol{\theta}^t)$$

$$\Leftrightarrow \eta\langle -\boldsymbol{F}(\boldsymbol{\theta}^{t+\frac{1}{2}}), \boldsymbol{\theta}^{t+1} - \boldsymbol{\theta}\rangle \leq D_\psi(\boldsymbol{\theta}, \boldsymbol{\theta}^t) - D_\psi(\boldsymbol{\theta}, \boldsymbol{\theta}^{t+1}) - D_\psi(\boldsymbol{\theta}^{t+1}, \boldsymbol{\theta}^t).$$

(32)

Similarly, with $\boldsymbol{a} = \boldsymbol{\theta}^t$, $\boldsymbol{a}' = \boldsymbol{\theta}^{t+\frac{1}{2}}$, $\boldsymbol{a}^* = \boldsymbol{\theta}^{t+1}$ and $\boldsymbol{g} = -\eta\boldsymbol{F}(\boldsymbol{\theta}^{t-\frac{1}{2}})$, we get

$$\eta\langle -\boldsymbol{F}(\boldsymbol{\theta}^{t-\frac{1}{2}}), \boldsymbol{\theta}^{t+\frac{1}{2}} - \boldsymbol{\theta}^{t+1}\rangle \leq D_\psi(\boldsymbol{\theta}^{t+1}, \boldsymbol{\theta}^t) - D_\psi(\boldsymbol{\theta}^{t+1}, \boldsymbol{\theta}^{t+\frac{1}{2}}) - D_\psi(\boldsymbol{\theta}^{t+\frac{1}{2}}, \boldsymbol{\theta}^t)$$

$$\Leftrightarrow \eta\langle -\boldsymbol{F}(\boldsymbol{\theta}^{t-\frac{1}{2}}), \boldsymbol{\theta}^{t+\frac{1}{2}} - \boldsymbol{\theta}^{t+1}\rangle \leq D_\psi(\boldsymbol{\theta}^{t+1}, \boldsymbol{\theta}^t) - D_\psi(\boldsymbol{\theta}^{t+1}, \boldsymbol{\theta}^{t+\frac{1}{2}}) - D_\psi(\boldsymbol{\theta}^{t+\frac{1}{2}}, \boldsymbol{\theta}^t).$$

(33)

Summing up Eq. (32) and (33), and adding $\eta\langle \boldsymbol{F}(\boldsymbol{\theta}^{t+\frac{1}{2}}) - \boldsymbol{F}(\boldsymbol{\theta}^{t-\frac{1}{2}}), \boldsymbol{\theta}^{t+1} - \boldsymbol{\theta}^{t+\frac{1}{2}}\rangle$ to both sides, we get

$$\eta\langle -\boldsymbol{F}(\boldsymbol{\theta}^{t+\frac{1}{2}}), \boldsymbol{\theta}^{t+\frac{1}{2}} - \boldsymbol{\theta}\rangle$$

$$\leq D_\psi(\boldsymbol{\theta}, \boldsymbol{\theta}^t) - D_\psi(\boldsymbol{\theta}, \boldsymbol{\theta}^{t+1}) - D_\psi(\boldsymbol{\theta}^{t+1}, \boldsymbol{\theta}^{t+\frac{1}{2}}) - D_\psi(\boldsymbol{\theta}^{t+\frac{1}{2}}, \boldsymbol{\theta}^t) + \eta\langle \boldsymbol{F}(\boldsymbol{\theta}^{t+\frac{1}{2}}) - \boldsymbol{F}(\boldsymbol{\theta}^{t-\frac{1}{2}}), \boldsymbol{\theta}^{t+1} - \boldsymbol{\theta}^{t+\frac{1}{2}}\rangle.$$

Arranging the terms, we have

$$D_\psi(\boldsymbol{\theta}, \boldsymbol{\theta}^{t+1}) - D_\psi(\boldsymbol{\theta}, \boldsymbol{\theta}^t)$$

$$\leq \eta\langle \boldsymbol{F}(\boldsymbol{\theta}^{t+\frac{1}{2}}), \boldsymbol{\theta}^{t+\frac{1}{2}} - \boldsymbol{\theta}\rangle + \eta\langle \boldsymbol{F}(\boldsymbol{\theta}^{t+\frac{1}{2}}) - \boldsymbol{F}(\boldsymbol{\theta}^{t-\frac{1}{2}}), \boldsymbol{\theta}^{t+1} - \boldsymbol{\theta}^{t+\frac{1}{2}}\rangle - D_\psi(\boldsymbol{\theta}^{t+1}, \boldsymbol{\theta}^{t+\frac{1}{2}}) - D_\psi(\boldsymbol{\theta}^{t+\frac{1}{2}}, \boldsymbol{\theta}^t)$$

$$\leq - \eta\langle \boldsymbol{F}(\boldsymbol{\theta}^{t+\frac{1}{2}}), \boldsymbol{\theta}\rangle + \eta\langle \boldsymbol{F}(\boldsymbol{\theta}^{t+\frac{1}{2}}) - \boldsymbol{F}(\boldsymbol{\theta}^{t-\frac{1}{2}}), \boldsymbol{\theta}^{t+1} - \boldsymbol{\theta}^{t+\frac{1}{2}}\rangle - D_\psi(\boldsymbol{\theta}^{t+1}, \boldsymbol{\theta}^{t+\frac{1}{2}}) - D_\psi(\boldsymbol{\theta}^{t+\frac{1}{2}}, \boldsymbol{\theta}^t),$$

where the last line comes from $\langle \boldsymbol{F}(\boldsymbol{\theta}^{t+\frac{1}{2}}), \boldsymbol{\theta}^{t+\frac{1}{2}}\rangle = \sum_{i\in\mathcal{N}}\langle \boldsymbol{F}_i(\boldsymbol{\theta}^{t+\frac{1}{2}}), \boldsymbol{\theta}_i^{t+\frac{1}{2}}\rangle = \sum_{i\in\mathcal{N}}\langle\langle \boldsymbol{\ell}_i^{t+\frac{1}{2}}, \boldsymbol{x}_i^{t+\frac{1}{2}}\rangle\boldsymbol{1} - \boldsymbol{\ell}_i^{t+\frac{1}{2}}, \boldsymbol{\theta}_i^{t+\frac{1}{2}}\rangle = \sum_{i\in\mathcal{N}}\langle \boldsymbol{\ell}_i^{t+\frac{1}{2}}, \boldsymbol{x}_i^{t+\frac{1}{2}}\rangle\langle\boldsymbol{1}, \boldsymbol{\theta}_i^{t+\frac{1}{2}}\rangle - \langle \boldsymbol{\ell}_i^{t+\frac{1}{2}}, \boldsymbol{\theta}_i^{t+\frac{1}{2}}\rangle = \sum_{i\in\mathcal{N}}\langle \boldsymbol{\ell}_i^{t+\frac{1}{2}}, \frac{\boldsymbol{\theta}_i^{t+\frac{1}{2}}}{\|\boldsymbol{\theta}_i^{t+\frac{1}{2}}\|_1}\rangle\|\boldsymbol{\theta}_i^{t+\frac{1}{2}}\|_1 - \langle \boldsymbol{\ell}_i^{t+\frac{1}{2}}, \boldsymbol{\theta}_i^{t+\frac{1}{2}}\rangle = 0$. It completes the proof.

## C.3 PROOF OF LEMMA C.3

To prove Lemma C.3, we first introduce Lemma C.4, which is Lemma 11 of Wei et al. (2021)

**Lemma C.4.** *Suppose that $\varphi(\cdot)$ satisfies $D_\varphi(\boldsymbol{b}, \boldsymbol{b}') \geq \frac{1}{2}\|\boldsymbol{b} - \boldsymbol{b}'\|_p^2$ for some $p \geq 1$, and let $\boldsymbol{a}, \boldsymbol{a}_1, \boldsymbol{a}_2 \in \boldsymbol{\mathcal{A}}$ (a convex set) be related by the following:*

$$\boldsymbol{a}_1 \in \arg\min_{\boldsymbol{a}'\in\boldsymbol{\mathcal{A}}}\{\langle \boldsymbol{a}', \boldsymbol{g}_1\rangle + D_\varphi(\boldsymbol{a}', \boldsymbol{a})\},$$

$$\boldsymbol{a}_2 \in \arg\min_{\boldsymbol{a}'\in\boldsymbol{\mathcal{A}}}\{\langle \boldsymbol{a}', \boldsymbol{g}_2\rangle + D_\varphi(\boldsymbol{a}', \boldsymbol{a})\}.$$

*Then, we have*

$$\|\boldsymbol{a}_1 - \boldsymbol{a}_2\|_p \leq \|\boldsymbol{g}_1 - \boldsymbol{g}_2\|_q,$$

*where $q \geq 1$ and $\frac{1}{p} + \frac{1}{q} = 1$.*

Considering Eq. (27) and substituting $\boldsymbol{a}_1 = \boldsymbol{\theta}^{t+1}$, $\boldsymbol{a}_2 = \boldsymbol{\theta}^{t+\frac{1}{2}}$, $\boldsymbol{g}_1 = -\eta \boldsymbol{F}(\boldsymbol{\theta}^{t+\frac{1}{2}})$, $\boldsymbol{g}_2 = -\eta \boldsymbol{F}(\boldsymbol{\theta}^{t-\frac{1}{2}})$ and $\varphi(\cdot) = \psi(\cdot)$ ($\psi(\cdot)$ is the quadratic regularizer, which satisfies $D_\psi(\boldsymbol{b}, \boldsymbol{b}') \geq \frac{1}{2}\|\boldsymbol{b} - \boldsymbol{b}'\|_2^2$) into Lemma C.4 (in this case, $\boldsymbol{\mathcal{A}}$ is $\times_{i \in \mathcal{N}} \mathbb{R}_{\geq 1}^{|A_i|}$), we have

$$\|\boldsymbol{\theta}^{t+1} - \boldsymbol{\theta}^{t+\frac{1}{2}}\|_2 \leq \|\eta \boldsymbol{F}(\boldsymbol{\theta}^{t+\frac{1}{2}}) - \eta \boldsymbol{F}(\boldsymbol{\theta}^{t-\frac{1}{2}})\|_2$$
$$\Leftrightarrow \|\boldsymbol{\theta}^{t+1} - \boldsymbol{\theta}^{t+\frac{1}{2}}\|_2 \leq \eta \|\boldsymbol{F}(\boldsymbol{\theta}^{t+\frac{1}{2}}) - \boldsymbol{F}(\boldsymbol{\theta}^{t-\frac{1}{2}})\|_2,$$

which completes the proof.

## D PROOF OF THEOREM 5.2

**Lemma D.1.** *Let $\boldsymbol{x}^* \in \mathcal{X}^*$ and $0 < \eta < \frac{1}{2DL_u}$, then for every iteration $t \geq 1$, it holds that*

$$\sum_{t=1}^{T} \left(\frac{1}{2} + \frac{2\rho}{\eta} - 2\eta^2 D^2 L_u^2\right) \|\boldsymbol{\theta}^{t+1} - \boldsymbol{\theta}^t\|_2^2 \leq \|\boldsymbol{\theta}^1 - \boldsymbol{x}^*\|_2^2 + \frac{1}{4}\|\boldsymbol{\theta}^{\frac{1}{2}} - \boldsymbol{\theta}^{-\frac{1}{2}}\|_2^2.$$

*where $D = \max_{i \in \mathcal{N}} |A_i|$ and $L_u = \sqrt{2P^2 + 4L^2}$.*

From Lemma D.1, we have

$$\sum_{t=1}^{T} \left(\frac{1}{2} + \frac{2\rho}{\eta} - 2\eta^2 D^2 L_u^2\right) \|\boldsymbol{\theta}^{t+1} - \boldsymbol{\theta}^t\|_2^2 \leq \|\boldsymbol{\theta}^1 - \boldsymbol{x}^*\|_2^2 + \frac{1}{4}\|\boldsymbol{\theta}^{\frac{1}{2}} - \boldsymbol{\theta}^{-\frac{1}{2}}\|_2^2. \tag{34}$$

Now, we first prove that if $\rho > -\frac{1}{12\sqrt{3}DL_u}$, there always exists $0 < \eta < \frac{1}{2DL_u}$ that ensures $\frac{1}{2} + \frac{2\rho}{\eta} - 2\eta^2 D^2 L_u^2 > 0$. Formally, consider this case where $\rho = -\frac{1}{12\sqrt{3}DL_u}$, we can set $\eta = 1/(2\sqrt{3}DL_u)$ that ensures

$$\frac{1}{2} + \frac{2\rho}{\eta} - 2\eta^2 D^2 L_u^2 = \frac{1}{2} - \frac{1}{3} - \frac{1}{6} = 0.$$

Therefore, we can obtain that if $\rho > -\frac{1}{12\sqrt{3}DL_u}$, there always exists $0 < \eta < \frac{1}{2DL_u}$ that ensures $\frac{1}{2} + \frac{2\rho}{\eta} - 2\eta^2 D^2 L_u^2 > 0$.

Assume $\|\boldsymbol{\theta}^{t+1} - \boldsymbol{\theta}^t\|_2^2$ do not converge to $0$. Then, from Eq. (34), we have

$$\sum_{t=1}^{T} \left(\frac{1}{2} + \frac{2\rho}{\eta} - 2\eta^2 D^2 L_u^2\right) \|\boldsymbol{\theta}^{t+1} - \boldsymbol{\theta}^t\|_2^2 \geq O(T),$$

which contracts that

$$\sum_{t=1}^{T} \left(\frac{1}{2} + \frac{2\rho}{\eta} - 2\eta^2 D^2 L_u^2\right) \|\boldsymbol{\theta}^{t+1} - \boldsymbol{\theta}^t\|_2^2 \leq \|\boldsymbol{\theta}^1 - \boldsymbol{x}^*\|_2^2 + \frac{1}{4}\|\boldsymbol{\theta}^{\frac{1}{2}} - \boldsymbol{\theta}^{-\frac{1}{2}}\|_2^2.$$

Therefore, $\|\boldsymbol{\theta}^{t+1} - \boldsymbol{\theta}^t\|_2^2 \to 0$.

In addition, from $\frac{1}{2} + \frac{2\rho}{\eta} - 2\eta^2 D^2 L_u^2 > 0$ and Eq. (34), we have

$$\sum_{t=1}^{T} \|\boldsymbol{\theta}^{t+1} - \boldsymbol{\theta}^t\|_2^2 \leq \frac{\|\boldsymbol{\theta}^1 - \boldsymbol{x}^*\|_2^2 + \frac{1}{4}\|\boldsymbol{\theta}^{\frac{1}{2}} - \boldsymbol{\theta}^{-\frac{1}{2}}\|_2^2}{\left(\frac{1}{2} + \frac{2\rho}{\eta} - 2\eta^2 D^2 L_u^2\right)} = C.$$

Since $\boldsymbol{\theta}^1, \boldsymbol{\theta}^{\frac{1}{2}}, \boldsymbol{\theta}^{-\frac{1}{2}}, \boldsymbol{x}^*, \eta, \rho, D$, and $L_u$ is fixed, $C$ must be a constant. Therefore, we get

$$T \min_{t \in T} \|\boldsymbol{\theta}^{t+1} - \boldsymbol{\theta}^t\|_2^2 \leq \sum_{t=1}^{T} \|\boldsymbol{\theta}^{t+1} - \boldsymbol{\theta}^t\|_2^2 \leq C,$$

which implies

$$\min_{t \in T} \|\boldsymbol{\theta}^{t+1} - \boldsymbol{\theta}^t\|_2^2 \leq \frac{C}{T}.$$

### D.1 PROOF OF LEMMA D.1

**Lemma D.2.** *(Lemma 10 of Wei et al. (2021)) Let $\mathcal{A}$ as a convex set and $\boldsymbol{a}' \in \arg\min_{\boldsymbol{a}' \in \mathcal{A}}\{\langle \boldsymbol{a}', \boldsymbol{g}\rangle + D_\psi(\boldsymbol{a}', \boldsymbol{a})\}$. Then for any $\boldsymbol{a}^* \in \mathcal{A}$,*

$$\langle \boldsymbol{a}' - \boldsymbol{a}^*, \boldsymbol{g}\rangle \le D_\psi(\boldsymbol{a}^*, \boldsymbol{a}) - D_\psi(\boldsymbol{a}^*, \boldsymbol{a}') - D_\psi(\boldsymbol{a}', \boldsymbol{a}).$$

**Lemma D.3.** *(Adapted from Lemma A.2 of Hsieh et al. (2019)) Assume all players follow the update rule of SOGRM$^+$, then for any $\boldsymbol{\theta}_i \in \mathbb{R}_{\ge 1}^{|\mathcal{X}_i|}$, we have*

$$D_\psi(\boldsymbol{\theta}_i, \boldsymbol{\theta}_i^{t+1}) - D_\psi(\boldsymbol{\theta}_i, \boldsymbol{\theta}_i^t)$$
$$\le \langle \boldsymbol{\theta}_i^t - \boldsymbol{\theta}_i^{t+\frac{1}{2}} + \eta \boldsymbol{F}_i(\boldsymbol{\theta}^{t-\frac{1}{2}}) - \eta \boldsymbol{F}_i(\boldsymbol{\theta}^{t+\frac{1}{2}}), \boldsymbol{\theta}_i - \boldsymbol{\theta}_i^{t+\frac{1}{2}}\rangle + D_\psi(\boldsymbol{\theta}_i^{t+1}, \boldsymbol{\theta}_i^{t+\frac{1}{2}}) - D_\psi(\boldsymbol{\theta}_i^{t+\frac{1}{2}}, \boldsymbol{\theta}_i^t).$$

Considering Eq. (9) and using Lemma D.2 with $\boldsymbol{a} = \boldsymbol{\theta}_i^t$, $\boldsymbol{a}' = \boldsymbol{\theta}_i^{t+\frac{1}{2}}$, $\boldsymbol{a}^* = \boldsymbol{x}_i^{t+\frac{1}{2}}$, and $\boldsymbol{g} = -\eta \boldsymbol{F}_i(\boldsymbol{\theta}^{t-\frac{1}{2}})$, we have

$$0 \le \langle \eta \boldsymbol{F}_i(\boldsymbol{\theta}^{t-\frac{1}{2}}), \boldsymbol{\theta}_i^{t+\frac{1}{2}} - \boldsymbol{x}_i^{t+\frac{1}{2}}\rangle + D_\psi(\boldsymbol{x}_i^{t+\frac{1}{2}}, \boldsymbol{\theta}_i^t) - D_\psi(\boldsymbol{x}_i^{t+\frac{1}{2}}, \boldsymbol{\theta}_i^{t+\frac{1}{2}}) - D_\psi(\boldsymbol{\theta}_i^{t+\frac{1}{2}}, \boldsymbol{\theta}_i^t)$$
$$\Leftrightarrow 0 \le \langle \eta \boldsymbol{F}_i(\boldsymbol{\theta}^{t-\frac{1}{2}}), \boldsymbol{\theta}_i^{t+\frac{1}{2}} - \boldsymbol{x}_i^{t+\frac{1}{2}}\rangle + \langle \boldsymbol{\theta}_i^t - \boldsymbol{\theta}_i^{t+\frac{1}{2}}, \boldsymbol{\theta}_i^{t+\frac{1}{2}} - \boldsymbol{x}_i^{t+\frac{1}{2}}\rangle,$$
(35)

where the second line comes from

$$D_\psi(\boldsymbol{x}_i^{t+\frac{1}{2}}, \boldsymbol{\theta}_i^t) - D_\psi(\boldsymbol{x}_i^{t+\frac{1}{2}}, \boldsymbol{\theta}_i^{t+\frac{1}{2}}) - D_\psi(\boldsymbol{\theta}_i^{t+\frac{1}{2}}, \boldsymbol{\theta}_i^t)$$
$$= \frac{\|\boldsymbol{x}_i^{t+\frac{1}{2}}\|_2^2}{2} - \langle \boldsymbol{x}_i^{t+\frac{1}{2}}, \boldsymbol{\theta}_i^t\rangle + \frac{\|\boldsymbol{\theta}_i^t\|_2^2}{2} - \frac{\|\boldsymbol{x}_i^{t+\frac{1}{2}}\|_2^2}{2} + \langle \boldsymbol{x}_i^{t+\frac{1}{2}}, \boldsymbol{\theta}_i^{t+\frac{1}{2}}\rangle - \frac{\|\boldsymbol{\theta}_i^{t+\frac{1}{2}}\|_2^2}{2} - \frac{\|\boldsymbol{\theta}_i^{t+\frac{1}{2}}\|_2^2}{2} + \langle \boldsymbol{\theta}_i^{t+\frac{1}{2}}, \boldsymbol{\theta}_i^t\rangle - \frac{\|\boldsymbol{\theta}_i^t\|_2^2}{2}$$
$$= \langle \boldsymbol{\theta}_i^t - \boldsymbol{\theta}_i^{t+\frac{1}{2}}, \boldsymbol{\theta}_i^{t+\frac{1}{2}} - \boldsymbol{x}_i^{t+\frac{1}{2}}\rangle.$$

Substituting $\boldsymbol{\theta}_i = \boldsymbol{x}_i^* \in \mathcal{X}^*$ and Eq. (35) into Lemma D.3, and using the fact that $\langle \boldsymbol{F}_i(\boldsymbol{\theta}^{t+\frac{1}{2}}), \boldsymbol{\theta}_i^{t+\frac{1}{2}}\rangle = \langle \langle \boldsymbol{\ell}^{t+\frac{1}{2}}, \boldsymbol{x}^{t+\frac{1}{2}}\rangle \mathbf{1} - \boldsymbol{\ell}^{t+\frac{1}{2}}, \boldsymbol{\theta}_i^{t+\frac{1}{2}}\rangle = 0$ ($\boldsymbol{x}_i^{t+\frac{1}{2}} = \boldsymbol{\theta}_i^{t+\frac{1}{2}}/\|\boldsymbol{\theta}_i^{t+\frac{1}{2}}\|_1$) and $\langle \boldsymbol{F}_i(\boldsymbol{\theta}^{t+\frac{1}{2}}), \boldsymbol{x}_i^{t+\frac{1}{2}}\rangle = \langle \langle \boldsymbol{\ell}^{t+\frac{1}{2}}, \boldsymbol{x}^{t+\frac{1}{2}}\rangle \mathbf{1} - \boldsymbol{\ell}^{t+\frac{1}{2}}, \boldsymbol{x}_i^{t+\frac{1}{2}}\rangle = 0$, we get

$$D_\psi(\boldsymbol{\theta}_i, \boldsymbol{\theta}_i^{t+1}) - D_\psi(\boldsymbol{\theta}_i, \boldsymbol{\theta}_i^t)$$
$$\le \langle \boldsymbol{\theta}_i^t - \boldsymbol{\theta}_i^{t+\frac{1}{2}} + \eta \boldsymbol{F}_i(\boldsymbol{\theta}^{t-\frac{1}{2}}) - \eta \boldsymbol{F}_i(\boldsymbol{\theta}^{t+\frac{1}{2}}), \boldsymbol{x}_i^* - \boldsymbol{x}_i^{t+\frac{1}{2}}\rangle + D_\psi(\boldsymbol{\theta}_i^{t+1}, \boldsymbol{\theta}_i^{t+\frac{1}{2}}) - D_\psi(\boldsymbol{\theta}_i^{t+\frac{1}{2}}, \boldsymbol{\theta}_i^t).$$
(36)

Since $\boldsymbol{\theta}_i^{t+1} = \boldsymbol{\theta}_i^{t+\frac{1}{2}} - \eta \boldsymbol{F}_i(\boldsymbol{\theta}^{t-\frac{1}{2}}) + \eta \boldsymbol{F}_i(\boldsymbol{\theta}^{t+\frac{1}{2}})$, we have

$$\frac{\boldsymbol{\theta}_i^t - \boldsymbol{\theta}_i^{t+1}}{\eta} = \frac{\boldsymbol{\theta}_i^t - \boldsymbol{\theta}_i^{t+\frac{1}{2}}}{\eta} + \boldsymbol{F}_i(\boldsymbol{\theta}^{t-\frac{1}{2}}) - \boldsymbol{F}_i(\boldsymbol{\theta}^{t+\frac{1}{2}}).$$
(37)

From Eq. (12), we have

$$\frac{\boldsymbol{\theta}^t - \boldsymbol{\theta}^{t+1}}{\eta} - \boldsymbol{\ell}^{t+\frac{1}{2}} \in \mathcal{N}_{\mathcal{X}}(\boldsymbol{x}^{t+\frac{1}{2}}).$$
(38)

From the definition of weak MVI ($\langle \boldsymbol{\ell}^{\boldsymbol{x}} + \boldsymbol{z}, \boldsymbol{x} - \boldsymbol{x}^*\rangle \ge \rho\|\boldsymbol{\ell}^{\boldsymbol{x}} + \boldsymbol{z}\|_2^2, \forall \boldsymbol{z} \in \mathcal{N}_{\mathcal{X}}(\boldsymbol{x})$) and setting $\boldsymbol{x} = \boldsymbol{x}^{t+\frac{1}{2}}$ and $\boldsymbol{z} = \frac{\boldsymbol{\theta}_i^t - \boldsymbol{\theta}_i^{t+1}}{\eta} - \boldsymbol{\ell}^{t+\frac{1}{2}} \in \mathcal{N}_{\mathcal{X}}(\boldsymbol{x}^{t+\frac{1}{2}})$ (Eq. (38)), we have

$$\langle \boldsymbol{\theta}^t - \boldsymbol{\theta}^{t+1}, \boldsymbol{x}^* - \boldsymbol{x}^{t+\frac{1}{2}}\rangle = \eta \langle \boldsymbol{\ell}^{t+\frac{1}{2}} + \frac{\boldsymbol{\theta}^t - \boldsymbol{\theta}^{t+1}}{\eta} - \boldsymbol{\ell}^{t+\frac{1}{2}}, \boldsymbol{x}^* - \boldsymbol{x}^{t+\frac{1}{2}}\rangle$$
$$\le -\rho\eta\|\frac{\boldsymbol{\theta}^t - \boldsymbol{\theta}^{t+1}}{\eta}\|_2^2 \le -\frac{2\rho}{\eta}D_\psi(\boldsymbol{\theta}^{t+1}, \boldsymbol{\theta}^t).$$
(39)

Now, we define $c = \frac{1}{2} - 2\eta^2 D^2 L_u^2 > 0$. Combining Eq. (36), (37) and (39), we have

$$D_\psi(\boldsymbol{\theta}, \boldsymbol{\theta}^{t+1}) - D_\psi(\boldsymbol{\theta}, \boldsymbol{\theta}^t) \le D_\psi(\boldsymbol{\theta}^{t+1}, \boldsymbol{\theta}^{t+\frac{1}{2}}) - D_\psi(\boldsymbol{\theta}^{t+\frac{1}{2}}, \boldsymbol{\theta}^t) - \frac{2\rho}{\eta}D_\psi(\boldsymbol{\theta}^{t+1}, \boldsymbol{\theta}^t)$$
$$\le D_\psi(\boldsymbol{\theta}^{t+1}, \boldsymbol{\theta}^{t+\frac{1}{2}}) - D_\psi(\boldsymbol{\theta}^{t+\frac{1}{2}}, \boldsymbol{\theta}^t) + c D_\psi(\boldsymbol{\theta}^{t+1}, \boldsymbol{\theta}^t) - (\frac{2\rho}{\eta} + c)D_\psi(\boldsymbol{\theta}^{t+1}, \boldsymbol{\theta}^t)$$
(40)
$$\le (1+2c)D_\psi(\boldsymbol{\theta}^{t+1}, \boldsymbol{\theta}^{t+\frac{1}{2}}) - (1-2c)D_\psi(\boldsymbol{\theta}^{t+\frac{1}{2}}, \boldsymbol{\theta}^t) - (\frac{2\rho}{\eta} + c)D_\psi(\boldsymbol{\theta}^{t+1}, \boldsymbol{\theta}^t),$$

where the last line comes from $D_\psi(\boldsymbol{\theta}^{t+1}, \boldsymbol{\theta}^t) \le 2D_\psi(\boldsymbol{\theta}^{t+1}, \boldsymbol{\theta}^{t+\frac{1}{2}}) + 2D_\psi(\boldsymbol{\theta}^{t+\frac{1}{2}}, \boldsymbol{\theta}^t)$.

Using the fact that $\boldsymbol{\theta}_i^{t+1} = \boldsymbol{\theta}_i^{t+\frac{1}{2}} - \eta \boldsymbol{F}_i(\boldsymbol{\theta}^{t-\frac{1}{2}}) + \eta \boldsymbol{F}_i(\boldsymbol{\theta}^{t+\frac{1}{2}})$, Lemma 5.2 of Farina et al. (2023) $(\|\boldsymbol{F}(\boldsymbol{\theta}) - \boldsymbol{F}(\boldsymbol{\theta}')\|_2 \leq DL_u \|\boldsymbol{\theta} - \boldsymbol{\theta}'\|_2, \forall \boldsymbol{\theta}, \boldsymbol{\theta}' \in \mathbb{R}_{\geq 1}^{|\boldsymbol{\mathcal{X}}|}$, where $D, L_u$ are defined in Theorem 4.1), and $D_\psi(\boldsymbol{a}, \boldsymbol{b}) = \|\boldsymbol{a} - \boldsymbol{b}\|_2^2/2$, we have

$$D_\psi(\boldsymbol{\theta}^{t+1}, \boldsymbol{\theta}^{t+\frac{1}{2}}) = D_\psi(\eta \boldsymbol{F}(\boldsymbol{\theta}^{t-\frac{1}{2}}), \eta \boldsymbol{F}(\boldsymbol{\theta}^{t+\frac{1}{2}})) \leq \eta^2 D^2 L_u^2 D_\psi(\boldsymbol{\theta}^{t-\frac{1}{2}}, \boldsymbol{\theta}^{t+\frac{1}{2}}). \tag{41}$$

Using Eq. (41), we get

$$D_\psi(\boldsymbol{\theta}^{t+\frac{1}{2}}, \boldsymbol{\theta}^{t-\frac{1}{2}}) \leq 2D_\psi(\boldsymbol{\theta}^{t+\frac{1}{2}}, \boldsymbol{\theta}^t) + 2D_\psi(\boldsymbol{\theta}^t, \boldsymbol{\theta}^{t-\frac{1}{2}}) \leq 2D_\psi(\boldsymbol{\theta}^{t+\frac{1}{2}}, \boldsymbol{\theta}^t) + 2\eta^2 D^2 L_u^2 D_\psi(\boldsymbol{\theta}^{t-\frac{1}{2}}, \boldsymbol{\theta}^{t-\frac{3}{2}}),$$

which implies

$$D_\psi(\boldsymbol{\theta}^{t+\frac{1}{2}}, \boldsymbol{\theta}^t) \geq \frac{1}{2} D_\psi(\boldsymbol{\theta}^{t+\frac{1}{2}}, \boldsymbol{\theta}^{t-\frac{1}{2}}) - \eta^2 D^2 L_u^2 D_\psi(\boldsymbol{\theta}^{t-\frac{1}{2}}, \boldsymbol{\theta}^{t-\frac{3}{2}}), \tag{42}$$

Combining Eq. (40), (41), and (42), we have

$$D_\psi(\boldsymbol{\theta}, \boldsymbol{\theta}^{t+1}) - D_\psi(\boldsymbol{\theta}, \boldsymbol{\theta}^t)$$

$$\leq (1 + 2c) D_\psi(\boldsymbol{\theta}^{t+1}, \boldsymbol{\theta}^{t+\frac{1}{2}}) - (1 - 2c) D_\psi(\boldsymbol{\theta}^{t+\frac{1}{2}}, \boldsymbol{\theta}^t) - (\frac{2\rho}{\eta} + c) D_\psi(\boldsymbol{\theta}^{t+1}, \boldsymbol{\theta}^t)$$

$$\leq -(\frac{1}{2} - c - (1 + 2c)\eta^2 D^2 L_u^2) D_\psi(\boldsymbol{\theta}^{t+\frac{1}{2}}, \boldsymbol{\theta}^{t-\frac{1}{2}}) + (1 - 2c)\eta^2 D^2 L_u^2 D_\psi(\boldsymbol{\theta}^{t-\frac{1}{2}}, \boldsymbol{\theta}^{t-\frac{3}{2}}) -$$

$$(\frac{2\rho}{\eta} + c) D_\psi(\boldsymbol{\theta}^{t+1}, \boldsymbol{\theta}^t)$$

$$\leq -(\frac{2\rho}{\eta} + c) D_\psi(\boldsymbol{\theta}^{t+1}, \boldsymbol{\theta}^t) + 4\eta^4 D^4 L_u^4 \left( D_\psi(\boldsymbol{\theta}^{t-\frac{1}{2}}, \boldsymbol{\theta}^{t-\frac{3}{2}}) - D_\psi(\boldsymbol{\theta}^{t+\frac{1}{2}}, \boldsymbol{\theta}^{t-\frac{1}{2}}) \right).$$

Telescoping the above inequality, and using $c = \frac{1}{2} - 2\eta^2 D^2 L_u^2$ with $D_\psi(\boldsymbol{a}, \boldsymbol{b}) = \|\boldsymbol{a} - \boldsymbol{b}\|_2^2/2$, we have

$$\sum_{t=1}^T \left( \frac{1}{2} + \frac{2\rho}{\eta} - 2\eta^2 D^2 L_u^2 \right) \|\boldsymbol{\theta}^{t+1} - \boldsymbol{\theta}^t\|_2^2 \leq \|\boldsymbol{\theta}^1 - \boldsymbol{x}^*\|_2^2 + 4\eta^4 D^4 L_u^4 \|\boldsymbol{\theta}^{\frac{1}{2}} - \boldsymbol{\theta}^{-\frac{1}{2}}\|_2^2$$

$$\leq \|\boldsymbol{\theta}^1 - \boldsymbol{x}^*\|_2^2 + \frac{1}{4} \|\boldsymbol{\theta}^{\frac{1}{2}} - \boldsymbol{\theta}^{-\frac{1}{2}}\|_2^2,$$

where the last line comes from $4\eta^4 D^4 L_u^4 \leq \frac{1}{4}$ (note that $c > 0$, which implies $2\eta^2 D^2 L_u^2 < \frac{1}{2}$, thus $4\eta^4 D^4 L_u^4 \leq \frac{1}{4}$).

## E  PROOF OF LEMMA 5.3

From the definition of $\sum_{i \in \mathcal{N}} \langle \boldsymbol{\ell}_i^{t-\frac{1}{2}} - \frac{\boldsymbol{\theta}^t - \boldsymbol{\theta}^{t+\frac{1}{2}}}{\eta}, \boldsymbol{x}_i - \boldsymbol{x}_i^{t+\frac{1}{2}} \rangle$, we have

$$\sum_{i \in \mathcal{N}} \langle \boldsymbol{\ell}_i^{t-\frac{1}{2}} - \frac{\boldsymbol{\theta}^t - \boldsymbol{\theta}^{t+\frac{1}{2}}}{\eta}, \boldsymbol{x}_i - \boldsymbol{x}_i^{t+\frac{1}{2}} \rangle$$

$$= \sum_{i \in \mathcal{N}} \langle \boldsymbol{\ell}_i^{t-\frac{1}{2}} - \frac{\boldsymbol{\theta}^t - \boldsymbol{\theta}^{t+1}}{\eta} + \boldsymbol{F}_i(\boldsymbol{\theta}^{t-\frac{1}{2}}) - \boldsymbol{F}_i(\boldsymbol{\theta}^{t+\frac{1}{2}}), \boldsymbol{x}_i - \boldsymbol{x}_i^{t+\frac{1}{2}} \rangle$$

$$= \sum_{i \in \mathcal{N}} \langle \boldsymbol{\ell}_i^{t-\frac{1}{2}} - \frac{\boldsymbol{\theta}^t - \boldsymbol{\theta}^{t+1}}{\eta} + \langle \boldsymbol{\ell}_i^{t-\frac{1}{2}}, \boldsymbol{x}_i^{t-\frac{1}{2}} \rangle \boldsymbol{1} - \boldsymbol{\ell}_i^{t-\frac{1}{2}} - \langle \boldsymbol{\ell}_i^{t+\frac{1}{2}}, \boldsymbol{x}_i^{t+\frac{1}{2}} \rangle \boldsymbol{1} + \boldsymbol{\ell}_i^{t+\frac{1}{2}}, \boldsymbol{x}_i - \boldsymbol{x}_i^{t+\frac{1}{2}} \rangle$$

$$= \sum_{i \in \mathcal{N}} \langle \boldsymbol{\ell}_i^{t+\frac{1}{2}} - \frac{\boldsymbol{\theta}^t - \boldsymbol{\theta}^{t+1}}{\eta}, \boldsymbol{x}_i - \boldsymbol{x}_i^{t+\frac{1}{2}} \rangle,$$

where the last line is from $\langle \langle \boldsymbol{\ell}_i^{t-\frac{1}{2}}, \boldsymbol{x}_i^{t-\frac{1}{2}} \rangle \boldsymbol{1}, \boldsymbol{x}_i - \boldsymbol{x}_i^{t+\frac{1}{2}} \rangle = 0$ and $\langle \langle \boldsymbol{\ell}_i^{t+\frac{1}{2}}, \boldsymbol{x}_i^{t+\frac{1}{2}} \rangle \boldsymbol{1}, \boldsymbol{x}_i - \boldsymbol{x}_i^{t+\frac{1}{2}} \rangle = 0$.

## F  HOW TO OBTAIN EQ. (10), (7), AND (8) VIA THE ANALYSIS IN SECTION 3

Now, we provide the details of obtaining Eq. (10), (7), and (8) from Eq. (9), (2), and (3) via the analysis in Section 3. From the analysis in Section 3, we have that $\forall \boldsymbol{\theta}_i^{t_2}, \boldsymbol{\theta}_i^{t_1}, \boldsymbol{\theta}_i^{t_0} \in \mathbb{R}_{\geq 1}^{|A_i|}, \eta > 0$ and

$\psi(\cdot)$ as the quadratic regularizer, the update rule in Eq. (43) can be written as the form in Eq. (44).

$$
\begin{cases}
\boldsymbol{x}_i^{t_2} = \dfrac{\boldsymbol{\theta}_i^{t_2}}{\|\boldsymbol{\theta}_i^{t_2}\|_1}, \ \boldsymbol{\theta}_i^{t_2} \in \underset{\boldsymbol{\theta}_i \in \mathbb{R}_{\geq 1}^{|A_i|}}{\arg\min}\{\langle -\boldsymbol{F}_i(\boldsymbol{\theta}^{t_1}), \boldsymbol{\theta}_i\rangle + \dfrac{1}{\eta}D_\psi(\boldsymbol{\theta}_i, \boldsymbol{\theta}_i^{t_0})\}, \\[2mm]
\boldsymbol{F}_i(\boldsymbol{\theta}^{t_1}) = \langle \dfrac{\boldsymbol{\theta}_i^{t_1}}{\|\boldsymbol{\theta}_i^{t_1}\|_1}, \boldsymbol{\ell}_i^{\boldsymbol{\theta^{t_1}}}\rangle \mathbf{1} - \boldsymbol{\ell}_i^{\boldsymbol{\theta^{t_1}}},
\end{cases}
\tag{43}
$$

$$
\begin{cases}
\boldsymbol{x}_i^{t_2} \in \underset{\boldsymbol{x}_i \in \mathcal{X}_i}{\arg\min}\{\langle \boldsymbol{\ell}_i^{\boldsymbol{\theta^{t_1}}}, \boldsymbol{x}_i\rangle + f_i(\boldsymbol{x}_i) + D_{h_i}(\boldsymbol{x}_i, \boldsymbol{x}_i^{t_0})\}, \\[2mm]
h_i(\boldsymbol{x}_i) + f_i(\boldsymbol{x}_i) = \dfrac{\|\boldsymbol{\theta}_i^{t_2}\|_1}{\eta}\psi(\boldsymbol{x}_i), \ h_i(\boldsymbol{x}_i) = \dfrac{\|\boldsymbol{\theta}_i^{t_0}\|_1}{\eta}\psi(\boldsymbol{x}_i),
\end{cases}
\tag{44}
$$

where $\boldsymbol{x}_i^{t_2} = \boldsymbol{\theta}_i^{t_2}/\|\boldsymbol{\theta}_i^{t_2}\|_1$, $\boldsymbol{x}_i^{t_0} = \boldsymbol{\theta}_i^{t_0}/\|\boldsymbol{\theta}_i^{t_0}\|_1$.

Consider the update rule of SOGRM$^+$ as shown in the following

$$
\boldsymbol{\theta}_i^{t+\frac{1}{2}} \in \underset{\boldsymbol{\theta}_i \in \mathbb{R}_{\geq 1}^{|A_i|}}{\arg\min}\{\langle -\boldsymbol{F}_i(\boldsymbol{\theta}^{t-\frac{1}{2}}), \boldsymbol{\theta}_i\rangle + \frac{1}{\eta}D_\psi(\boldsymbol{\theta}_i, \boldsymbol{\theta}_i^t)\}, \ \boldsymbol{x}_i^{t+\frac{1}{2}} = \frac{\boldsymbol{\theta}_i^{t+\frac{1}{2}}}{\|\boldsymbol{\theta}_i^{t+\frac{1}{2}}\|_1},
\tag{45}
$$

$$
\boldsymbol{\theta}_i^{t+1} = \boldsymbol{\theta}_i^{t+\frac{1}{2}} - \eta\boldsymbol{F}_i(\boldsymbol{\theta}^{t-\frac{1}{2}}) + \eta\boldsymbol{F}_i(\boldsymbol{\theta}^{t+\frac{1}{2}}).
$$

Substituting $\boldsymbol{\theta}_i^{t_2} = \boldsymbol{\theta}_i^{t+\frac{1}{2}}, \boldsymbol{\theta}_i^{t_1} = \boldsymbol{\theta}_i^{t-\frac{1}{2}}, \boldsymbol{\theta}_i^{t_0} = \boldsymbol{\theta}_i^t$ into Eq. (43), we have that

$$
\boldsymbol{\theta}_i^{t+\frac{1}{2}} \in \underset{\boldsymbol{\theta}_i \in \mathbb{R}_{\geq 1}^{|A_i|}}{\arg\min}\{\langle -\boldsymbol{F}_i(\boldsymbol{\theta}^{t-\frac{1}{2}}), \boldsymbol{\theta}_i\rangle + \frac{1}{\eta}D_\psi(\boldsymbol{\theta}_i, \boldsymbol{\theta}_i^t)\}, \ \boldsymbol{x}_i^{t+\frac{1}{2}} = \frac{\boldsymbol{\theta}_i^{t+\frac{1}{2}}}{\|\boldsymbol{\theta}_i^{t+\frac{1}{2}}\|_1},
$$

$$
\boldsymbol{F}_i(\boldsymbol{\theta}^{t-\frac{1}{2}}) = \langle \frac{\boldsymbol{\theta}_i^{t-\frac{1}{2}}}{\|\boldsymbol{\theta}_i^{t-\frac{1}{2}}\|_1}, \boldsymbol{\ell}_i^{\boldsymbol{\theta^{t-\frac{1}{2}}}}\rangle \mathbf{1} - \boldsymbol{\ell}_i^{\boldsymbol{\theta^{t-\frac{1}{2}}}},
$$

which is consistent with the first prox-mapping operator in Eq. (45). Therefore, according to the relationship between Eq. (43) and Eq. (44), we have that the first prox-mapping operator in Eq. (45) and $\boldsymbol{x}_i^{t+\frac{1}{2}} = \boldsymbol{\theta}_i^{t+\frac{1}{2}}/\|\boldsymbol{\theta}_i^{t+\frac{1}{2}}\|_1$ can be rewrite as

$$
\boldsymbol{x}_i^{t+\frac{1}{2}} \in \underset{\boldsymbol{x}_i \in \mathcal{X}_i}{\arg\min}\{\langle \boldsymbol{\ell}_i^{t-\frac{1}{2}}, \boldsymbol{x}_i\rangle + q_i^{t-\frac{1}{2}}(\boldsymbol{x}_i) + D_{q_i^{0:t-1}}(\boldsymbol{x}_i, \boldsymbol{x}_i^t)\},
$$

$$
q_i^{0:t-1}(\boldsymbol{x}_i) = \frac{\|\boldsymbol{\theta}_i^t\|_1}{\eta}\psi(\boldsymbol{x}_i), \quad q_i^{0:t-1}(\boldsymbol{x}_i) + q_i^{t-\frac{1}{2}}(\boldsymbol{x}_i) = \frac{\|\boldsymbol{\theta}_i^{t+\frac{1}{2}}\|_1}{\eta}\psi(\boldsymbol{x}_i).
$$

In this case, $h_i(\boldsymbol{x}_i), f_i(\boldsymbol{x}_i)$ in Eq. (44) are $q_i^{0:t-1}(\boldsymbol{x}_i)$ and $q_i^{t-\frac{1}{2}}(\boldsymbol{x}_i)$, respectively. Therefore, we get Eq. (10).

Consider the update rule of SExRM$^+$ as shown in the following

$$
\boldsymbol{\theta}_i^{t+\frac{1}{2}} \in \underset{\boldsymbol{\theta}_i \in \mathbb{R}_{\geq 1}^{|A_i|}}{\arg\min}\{\langle -\boldsymbol{F}_i(\boldsymbol{\theta}^{t-\frac{1}{2}}), \boldsymbol{\theta}_i\rangle + \frac{1}{\eta}D_\psi(\boldsymbol{\theta}_i, \boldsymbol{\theta}_i^t)\}, \ \boldsymbol{x}_i^{t+\frac{1}{2}} = \frac{\boldsymbol{\theta}_i^{t+\frac{1}{2}}}{\|\boldsymbol{\theta}_i^{t+\frac{1}{2}}\|_1},
$$

$$
\boldsymbol{\theta}_i^{t+1} \in \underset{\boldsymbol{\theta}_i \in \mathbb{R}_{\geq 1}^{|A_i|}}{\arg\min}\{\langle -\boldsymbol{F}_i(\boldsymbol{\theta}^{t+\frac{1}{2}}), \boldsymbol{\theta}_i\rangle + \frac{1}{\eta}D_\psi(\boldsymbol{\theta}_i, \boldsymbol{\theta}_i^t)\}, \ \boldsymbol{x}_i^{t+1} = \frac{\boldsymbol{\theta}_i^{t+1}}{\|\boldsymbol{\theta}_i^{t+1}\|_1}.
\tag{46}
$$

Substituting $\boldsymbol{\theta}_i^{t_2} = \boldsymbol{\theta}_i^{t+\frac{1}{2}}, \boldsymbol{\theta}_i^{t_1} = \boldsymbol{\theta}_i^{t-\frac{1}{2}}, \boldsymbol{\theta}_i^{t_0} = \boldsymbol{\theta}_i^t$ into Eq. (43), we have that

$$
\boldsymbol{\theta}_i^{t+\frac{1}{2}} \in \underset{\boldsymbol{\theta}_i \in \mathbb{R}_{\geq 1}^{|A_i|}}{\arg\min}\{\langle -\boldsymbol{F}_i(\boldsymbol{\theta}^t), \boldsymbol{\theta}_i\rangle + \frac{1}{\eta}D_\psi(\boldsymbol{\theta}_i, \boldsymbol{\theta}_i^t)\}, \ \boldsymbol{x}_i^{t+\frac{1}{2}} = \frac{\boldsymbol{\theta}_i^{t+\frac{1}{2}}}{\|\boldsymbol{\theta}_i^{t+\frac{1}{2}}\|_1},
$$

$$
\boldsymbol{F}_i(\boldsymbol{\theta}^t) = \langle \frac{\boldsymbol{\theta}_i^t}{\|\boldsymbol{\theta}_i^t\|_1}, \boldsymbol{\ell}_i^{\boldsymbol{\theta^t}}\rangle \mathbf{1} - \boldsymbol{\ell}_i^{\boldsymbol{\theta^t}},
$$

which is consistent with the first prox-mapping operator in Eq. (46). Therefore, according to the relationship between Eq. (43) and Eq. (44), we have that the first prox-mapping operator in Eq. (46) and $\boldsymbol{x}_i^{t+\frac{1}{2}} = \boldsymbol{\theta}_i^{t+\frac{1}{2}}/\|\boldsymbol{\theta}_i^{t+\frac{1}{2}}\|_1$ can be rewrite as

$$\boldsymbol{x}_i^{t+\frac{1}{2}} \in \arg\min_{\boldsymbol{x}_i \in \boldsymbol{\mathcal{X}}_i}\{\langle \boldsymbol{\ell}_i^{t-\frac{1}{2}}, \boldsymbol{x}_i\rangle + q_i^{t-\frac{1}{2}}(\boldsymbol{x}_i) + D_{q_i^{0:t-1}}(\boldsymbol{x}_i, \boldsymbol{x}_i^t)\},$$

$$q_i^{0:t-1}(\boldsymbol{x}_i) = \frac{\|\boldsymbol{\theta}_i^t\|_1}{\eta}\psi(\boldsymbol{x}_i), \quad q_i^{0:t-1}(\boldsymbol{x}_i) + q_i^{t-\frac{1}{2}}(\boldsymbol{x}_i) = \frac{\|\boldsymbol{\theta}_i^{t+\frac{1}{2}}\|_1}{\eta}\psi(\boldsymbol{x}_i). \tag{47}$$

In this case, $h_i(\boldsymbol{x}_i)$, $f_i(\boldsymbol{x}_i)$ in Eq. (44) are $q_i^{0:t-1}(\boldsymbol{x}_i)$ and $q_i^{t-\frac{1}{2}}(\boldsymbol{x}_i)$, respectively. Similarly, substituting $\boldsymbol{\theta}_i^{t_2} = \boldsymbol{\theta}_i^{t+1}, \boldsymbol{\theta}_i^{t_1} = \boldsymbol{\theta}_i^{t+\frac{1}{2}}, \boldsymbol{\theta}_i^{t_0} = \boldsymbol{\theta}_i^t$ into Eq. (43), we have that

$$\boldsymbol{\theta}_i^{t+1} \in \arg\min_{\boldsymbol{\theta}_i \in \mathbb{R}_{\geq 1}^{|A_i|}}\{\langle -\boldsymbol{F}_i(\boldsymbol{\theta}^{t+\frac{1}{2}}), \boldsymbol{\theta}_i\rangle + \frac{1}{\eta}D_\psi(\boldsymbol{\theta}_i, \boldsymbol{\theta}_i^t)\}, \quad \boldsymbol{x}_i^{t+1} = \frac{\boldsymbol{\theta}_i^{t+1}}{\|\boldsymbol{\theta}_i^{t+1}\|_1},$$

$$\boldsymbol{F}_i(\boldsymbol{\theta}^{t+\frac{1}{2}}) = \langle \frac{\boldsymbol{\theta}_i^{t+\frac{1}{2}}}{\|\boldsymbol{\theta}_i^{t+\frac{1}{2}}\|_1}, \boldsymbol{\ell}_i^{\boldsymbol{\theta}^{t+\frac{1}{2}}}\rangle\mathbf{1} - \boldsymbol{\ell}_i^{\boldsymbol{\theta}^{t+\frac{1}{2}}},$$

which is consistent with the second prox-mapping operator in Eq. (46). Therefore, according to the relationship between Eq. (43) and Eq. (44), we have that the second prox-mapping operator in Eq. (46) and $\boldsymbol{x}_i^{t+1} = \boldsymbol{\theta}_i^{t+1}/\|\boldsymbol{\theta}_i^{t+1}\|_1$ can be rewrite as

$$\boldsymbol{x}_i^{t+1} \in \arg\min_{\boldsymbol{x}_i \in \boldsymbol{\mathcal{X}}_i}\{\langle \boldsymbol{\ell}_i^{t+\frac{1}{2}}, \boldsymbol{x}_i\rangle + q_i^t(\boldsymbol{x}_i) + D_{q_i^{0:t-1}}(\boldsymbol{x}_i, \boldsymbol{x}_i^t)\},$$

$$q_i^{0:t-1}(\boldsymbol{x}_i) = \frac{\|\boldsymbol{\theta}_i^t\|_1}{\eta}\psi(\boldsymbol{x}_i), \quad q_i^{0:t-1}(\boldsymbol{x}_i) + q_i^t(\boldsymbol{x}_i) = \frac{\|\boldsymbol{\theta}_i^{t+1}\|_1}{\eta}\psi(\boldsymbol{x}_i). \tag{48}$$

In this case, $h_i(\boldsymbol{x}_i)$, $f_i(\boldsymbol{x}_i)$ in Eq. (44) are $q_i^{0:t-1}(\boldsymbol{x}_i)$ and $q_i^t(\boldsymbol{x}_i)$, respectively. Combining Eq. (47) with (48), we get Eq. (7).

Consider the update rule of SPRM$^+$ as shown in the following

$$\boldsymbol{\theta}_i^{t+\frac{1}{2}} \in \arg\min_{\boldsymbol{\theta}_i \in \mathbb{R}_{\geq 1}^{|A_i|}}\{\langle -\boldsymbol{F}_i(\boldsymbol{\theta}^{t-\frac{1}{2}}), \boldsymbol{\theta}_i\rangle + \frac{1}{\eta}D_\psi(\boldsymbol{\theta}_i, \boldsymbol{\theta}_i^t)\}, \quad \boldsymbol{x}_i^{t+\frac{1}{2}} = \frac{\boldsymbol{\theta}_i^{t+\frac{1}{2}}}{\|\boldsymbol{\theta}_i^{t+\frac{1}{2}}\|_1},$$

$$\boldsymbol{\theta}_i^{t+1} \in \arg\min_{\boldsymbol{\theta}_i \in \mathbb{R}_{\geq 1}^{|A_i|}}\{\langle -\boldsymbol{F}_i(\boldsymbol{\theta}^{t+\frac{1}{2}}), \boldsymbol{\theta}_i\rangle + \frac{1}{\eta}D_\psi(\boldsymbol{\theta}_i, \boldsymbol{\theta}_i^t)\}, \quad \boldsymbol{x}_i^{t+1} = \frac{\boldsymbol{\theta}_i^{t+1}}{\|\boldsymbol{\theta}_i^{t+1}\|_1}. \tag{49}$$

Substituting $\boldsymbol{\theta}_i^{t_2} = \boldsymbol{\theta}_i^{t+\frac{1}{2}}, \boldsymbol{\theta}_i^{t_1} = \boldsymbol{\theta}_i^{t-\frac{1}{2}}, \boldsymbol{\theta}_i^{t_0} = \boldsymbol{\theta}_i^t$ into Eq. (43), we have that

$$\boldsymbol{\theta}_i^{t+\frac{1}{2}} \in \arg\min_{\boldsymbol{\theta}_i \in \mathbb{R}_{\geq 1}^{|A_i|}}\{\langle -\boldsymbol{F}_i(\boldsymbol{\theta}^{t-\frac{1}{2}}), \boldsymbol{\theta}_i\rangle + \frac{1}{\eta}D_\psi(\boldsymbol{\theta}_i, \boldsymbol{\theta}_i^t)\}, \quad \boldsymbol{x}_i^{t+\frac{1}{2}} = \frac{\boldsymbol{\theta}_i^{t+\frac{1}{2}}}{\|\boldsymbol{\theta}_i^{t+\frac{1}{2}}\|_1},$$

$$\boldsymbol{F}_i(\boldsymbol{\theta}^{t-\frac{1}{2}}) = \langle \frac{\boldsymbol{\theta}_i^{t-\frac{1}{2}}}{\|\boldsymbol{\theta}_i^{t-\frac{1}{2}}\|_1}, \boldsymbol{\ell}_i^{\boldsymbol{\theta}^{t-\frac{1}{2}}}\rangle\mathbf{1} - \boldsymbol{\ell}_i^{\boldsymbol{\theta}^{t-\frac{1}{2}}},$$

which is consistent with the first prox-mapping operator in Eq. (49). Therefore, according to the relationship between Eq. (43) and Eq. (44), we have that the first prox-mapping operator in Eq. (49) and $\boldsymbol{x}_i^{t+\frac{1}{2}} = \boldsymbol{\theta}_i^{t+\frac{1}{2}}/\|\boldsymbol{\theta}_i^{t+\frac{1}{2}}\|_1$ can be rewrite as

$$\boldsymbol{x}_i^{t+\frac{1}{2}} \in \arg\min_{\boldsymbol{x}_i \in \boldsymbol{\mathcal{X}}_i}\{\langle \boldsymbol{\ell}_i^{t-\frac{1}{2}}, \boldsymbol{x}_i\rangle + q_i^{t-\frac{1}{2}}(\boldsymbol{x}_i) + D_{q_i^{0:t-1}}(\boldsymbol{x}_i, \boldsymbol{x}_i^t)\},$$

$$q_i^{0:t-1}(\boldsymbol{x}_i) = \frac{\|\boldsymbol{\theta}_i^t\|_1}{\eta}\psi(\boldsymbol{x}_i), \quad q_i^{0:t-1}(\boldsymbol{x}_i) + q_i^{t-\frac{1}{2}}(\boldsymbol{x}_i) = \frac{\|\boldsymbol{\theta}_i^{t+\frac{1}{2}}\|_1}{\eta}\psi(\boldsymbol{x}_i). \tag{50}$$

In this case, $h_i(\boldsymbol{x}_i)$, $f_i(\boldsymbol{x}_i)$ in Eq. (44) are $q_i^{0:t-1}(\boldsymbol{x}_i)$ and $q_i^{t-\frac{1}{2}}(\boldsymbol{x}_i)$, respectively. Similarly, substituting $\boldsymbol{\theta}_i^{t_2} = \boldsymbol{\theta}_i^{t+1}, \boldsymbol{\theta}_i^{t_1} = \boldsymbol{\theta}_i^{t+\frac{1}{2}}, \boldsymbol{\theta}_i^{t_0} = \boldsymbol{\theta}_i^t$ into Eq. (43), we have that

$$\boldsymbol{\theta}_i^{t+1} \in \arg\min_{\boldsymbol{\theta}_i \in \mathbb{R}_{\geq 1}^{|A_i|}}\{\langle -\boldsymbol{F}_i(\boldsymbol{\theta}^{t+\frac{1}{2}}), \boldsymbol{\theta}_i \rangle + \frac{1}{\eta}D_\psi(\boldsymbol{\theta}_i, \boldsymbol{\theta}_i^t)\}, \quad \boldsymbol{x}_i^{t+1} = \frac{\boldsymbol{\theta}_i^{t+1}}{\|\boldsymbol{\theta}_i^{t+1}\|_1},$$

$$\boldsymbol{F}_i(\boldsymbol{\theta}^{t+\frac{1}{2}}) = \langle \frac{\boldsymbol{\theta}_i^{t+\frac{1}{2}}}{\|\boldsymbol{\theta}_i^{t+\frac{1}{2}}\|_1}, \boldsymbol{\ell}_i^{\boldsymbol{\theta}^{t+\frac{1}{2}}} \rangle \boldsymbol{1} - \boldsymbol{\ell}_i^{\boldsymbol{\theta}^{t+\frac{1}{2}}},$$

which is consistent with the second prox-mapping operator in Eq. (49). Therefore, according to the relationship between Eq. (43) and Eq. (44), we have that the second prox-mapping operator in Eq. (49) and $\boldsymbol{x}_i^{t+1} = \boldsymbol{\theta}_i^{t+1}/\|\boldsymbol{\theta}_i^{t+1}\|_1$ can rewrite as

$$\boldsymbol{x}_i^{t+1} \in \arg\min_{\boldsymbol{x}_i \in \boldsymbol{\mathcal{X}}_i}\{\langle \boldsymbol{\ell}_i^{t+\frac{1}{2}}, \boldsymbol{x}_i \rangle + q_i^t(\boldsymbol{x}_i) + D_{q_i^{0:t-1}}(\boldsymbol{x}_i, \boldsymbol{x}_i^t)\},$$

$$q_i^{0:t-1}(\boldsymbol{x}_i) = \frac{\|\boldsymbol{\theta}_i^t\|_1}{\eta}\psi(\boldsymbol{x}_i), \quad q_i^{0:t-1}(\boldsymbol{x}_i) + q_i^t(\boldsymbol{x}_i) = \frac{\|\boldsymbol{\theta}_i^{t+1}\|_1}{\eta}\psi(\boldsymbol{x}_i). \tag{51}$$

In this case, $h_i(\boldsymbol{x}_i)$, $f_i(\boldsymbol{x}_i)$ in Eq. (44) are $q_i^{0:t-1}(\boldsymbol{x}_i)$ and $q_i^t(\boldsymbol{x}_i)$, respectively. Combining Eq. (50) with (51), we get Eq. (8).

## G    EXAMPLE OF DIFFERENT GAME TYPES

In this section, we provide examples of smooth games that satisfy the MVI and weak MVI, respectively. We do not provide the example of smooth games satisfying monotonicity as any two-player zero-sum matrix game is a smooth game and satisfies monotonicity. Note that in this section, we focus on two-player normal-form game, whose utility function is convex and represented by payoff matrices. Note that any two-player normal-form game is a smooth game. For each two-player normal-form game, the utility functions of player 0 and 1 are presented by payoff matrices $\boldsymbol{A}$ and $\boldsymbol{B}$, respectively. Formally, $u_0(\boldsymbol{x}) = \boldsymbol{x}_0^\mathrm{T}\boldsymbol{A}\boldsymbol{x}_1$ and $u_1(\boldsymbol{x}) = \boldsymbol{x}_1^\mathrm{T}\boldsymbol{B}^\mathrm{T}\boldsymbol{x}_0$, which implies $\boldsymbol{\ell}_0^{\boldsymbol{x}} = -\boldsymbol{A}\boldsymbol{x}_1$ and $\boldsymbol{\ell}_1^{\boldsymbol{x}} = -\boldsymbol{B}^\mathrm{T}\boldsymbol{x}_0$.

### G.1    EXAMPLE OF GAMES SATISFYING THE MVI

The example is defined as following

$$\boldsymbol{A} = \begin{pmatrix} 2 & 0 \\ 0 & 0 \end{pmatrix}, \quad \boldsymbol{B} = \begin{pmatrix} 2 & 0 \\ 0 & 0 \end{pmatrix}.$$

This game violates monotonicity when

$$\boldsymbol{x}_0 = \begin{pmatrix} 0 \\ 1 \end{pmatrix}, \quad \boldsymbol{x}_1 = \begin{pmatrix} 0 \\ 1 \end{pmatrix}, \quad \boldsymbol{x}_0' = \begin{pmatrix} 0.1 \\ 0.9 \end{pmatrix}, \quad \boldsymbol{x}_1' = \begin{pmatrix} 0.1 \\ 0.9 \end{pmatrix}.$$

Formally, in this case, we have

$$\boldsymbol{\ell}_0^{\boldsymbol{x}} = \begin{pmatrix} 0 \\ 0 \end{pmatrix}, \quad \boldsymbol{\ell}_1^{\boldsymbol{x}} = \begin{pmatrix} -2 \\ 0 \end{pmatrix}, \quad \boldsymbol{\ell}_0^{\boldsymbol{x}'} = \begin{pmatrix} 0 \\ 0 \end{pmatrix}, \quad \boldsymbol{\ell}_1^{\boldsymbol{x}'} = \begin{pmatrix} -2 \\ 0 \end{pmatrix}.$$

$$\langle \boldsymbol{\ell}^{\boldsymbol{x}} - \boldsymbol{\ell}^{\boldsymbol{x}'}, \boldsymbol{x} - \boldsymbol{x}' \rangle = \begin{pmatrix} 2 \\ 0 \end{pmatrix} \cdot \begin{pmatrix} -0.1 \\ 0.1 \end{pmatrix} + \begin{pmatrix} 2 \\ 0 \end{pmatrix} \cdot \begin{pmatrix} -0.1 \\ 0.1 \end{pmatrix} = -0.4 < 0$$

which violates monotonicity.

Now, we show that the provided example satisfies the MVI. The unique NE of this game (learned by "Nashpy" (Knight & Campbell, 2018)) is

$$\boldsymbol{x}_0^* = \begin{pmatrix} 1 \\ 0 \end{pmatrix}, \quad \boldsymbol{x}_1^* = \begin{pmatrix} 1 \\ 0 \end{pmatrix}.$$

We define the strategies of players as following

$$\boldsymbol{x}_0 = \begin{pmatrix} a \\ 1-a \end{pmatrix}, \quad \boldsymbol{x}_1 = \begin{pmatrix} b \\ 1-b \end{pmatrix},$$

where $0 \leq a \leq 1$ and $0 \leq b \leq 1$. The loss gradient $\boldsymbol{\ell}_i^{\boldsymbol{x}}$ of player $i$ is

$$\boldsymbol{\ell}_0^{\boldsymbol{x}} = -\boldsymbol{A}\boldsymbol{x}_1, \quad \boldsymbol{\ell}_1^{\boldsymbol{x}} = -\boldsymbol{B}^{\mathrm{T}}\boldsymbol{x}_0.$$

Formally, for player 0, we have

$$\boldsymbol{A}\boldsymbol{x}_1 = \begin{pmatrix} 2 & 0 \\ 0 & 0 \end{pmatrix} \begin{pmatrix} b \\ 1-b \end{pmatrix} = \begin{pmatrix} 2b \\ 0 \end{pmatrix},$$

$$\boldsymbol{\ell}_0^{\boldsymbol{x}} = -\boldsymbol{A}\boldsymbol{x}_1 = \begin{pmatrix} -2b \\ 0 \end{pmatrix}.$$

Similarly, for player 1, we have

$$\boldsymbol{B}^{\mathrm{T}} = \begin{pmatrix} 2 & 0 \\ 0 & 0 \end{pmatrix},$$

$$\boldsymbol{B}^{\mathrm{T}}\boldsymbol{x}_0 = \begin{pmatrix} 2 & 0 \\ 0 & 0 \end{pmatrix} \begin{pmatrix} a \\ 1-a \end{pmatrix} = \begin{pmatrix} -2a \\ 0 \end{pmatrix},$$

$$\boldsymbol{\ell}_1^{\boldsymbol{x}} = -\boldsymbol{B}^{\mathrm{T}}\boldsymbol{x}_0 = \begin{pmatrix} -2a \\ 0 \end{pmatrix}.$$

In this case, we have

$$\langle \boldsymbol{\ell}^{\boldsymbol{x}}, \boldsymbol{x} - \boldsymbol{x}^* \rangle = \begin{pmatrix} -2b \\ 0 \end{pmatrix} \cdot \begin{pmatrix} a-1 \\ 1-a \end{pmatrix} + \begin{pmatrix} -2a \\ 0 \end{pmatrix} \cdot \begin{pmatrix} b-1 \\ 1-b \end{pmatrix} = -4ab + 2a + 2b = (-4a+2)b + 2a.$$

We can find that $(-4a+2)b+2a$ is linear function w.r.t $b$ given fixed $a$. If $1 \geq a \geq \frac{1}{2}$, $(-4a+2)b+2a$ decreases as $b$ increases. Therefore, given $1 \geq a \geq \frac{1}{2}$, $\min_{0 \leq b \leq 1}(-4a+2)b+2a = (-4a+2)+2a = 2-2a \geq 0$. Similarly, if $0 \leq a < \frac{1}{2}$ $(-4a+2)b+2a$ decreases as $b$ decreases. Therefore, given $0 \leq a < \frac{1}{2}$, $\min_{0 \leq b \leq 1}(-4a+2)b+2a = 2a \geq 0$. Hence, we get $-4ab + 2b + 2a \geq 0$, which implies $-4ab + 2a + 2b \geq 0$. Therefore, we get

$$\langle \boldsymbol{\ell}^{\boldsymbol{x}}, \boldsymbol{x} - \boldsymbol{x}^* \rangle = -4ab + 2a + 2b \geq 0.$$

Then, we have $\langle \boldsymbol{\ell}^{\boldsymbol{x}}, \boldsymbol{x} - \boldsymbol{x}^* \rangle \geq 0, \forall \boldsymbol{x} \in \boldsymbol{\mathcal{X}}$ and $\exists \boldsymbol{x}^* \in \boldsymbol{\mathcal{X}}^*$, which means the MVI holds in this game.

## G.2 EXAMPLE OF GAMES SATISFYING THE WEAK MVI

The example is defined as following

$$\boldsymbol{A} = \begin{pmatrix} 1 & 0 \\ -1 & 1 \end{pmatrix}, \quad \boldsymbol{B} = \begin{pmatrix} 0 & 1 \\ -1 & 1 \end{pmatrix}.$$

The unique NE of this game (learned by "Nashpy" (Knight & Campbell, 2018)) is

$$\boldsymbol{x}_0^* = \begin{pmatrix} 0 \\ 1 \end{pmatrix}, \quad \boldsymbol{x}_1^* = \begin{pmatrix} 0 \\ 1 \end{pmatrix}.$$

This game violates the MVI when

$$\boldsymbol{x}_0 = \begin{pmatrix} 0.7 \\ 0.3 \end{pmatrix}, \quad \boldsymbol{x}_1 = \begin{pmatrix} 0.9 \\ 0.1 \end{pmatrix}.$$

Formally, in this case, we have

$$\langle \boldsymbol{\ell}^{\boldsymbol{x}}, \boldsymbol{x} - \boldsymbol{x}^* \rangle = \begin{pmatrix} -0.9 \\ 0.8 \end{pmatrix} \cdot \begin{pmatrix} 0.7 - 0 \\ 0.3 - 1 \end{pmatrix} + \begin{pmatrix} 0.3 \\ -1 \end{pmatrix} \cdot \begin{pmatrix} 0.9 - 0 \\ 0.1 - 1 \end{pmatrix} = -0.02 < 0,$$

which violates the MVI.

Now, we show that the provided example satisfies the weak MVI. Adapted from Lemma 2 of Cai et al. (2022b) (although the original statement of this lemma is established under monotone games, it can be naturally extended to the smooth games considered in our work. This extension is achieved

using $\langle \boldsymbol{\ell^x}, \boldsymbol{x} - \boldsymbol{x}' \rangle \leq \langle \boldsymbol{\ell^x}, \boldsymbol{x} - \boldsymbol{x}' \rangle + \langle \boldsymbol{z}, \boldsymbol{x} - \boldsymbol{x}' \rangle \leq \|\boldsymbol{\ell^x} + \boldsymbol{z}\|_2 \|\boldsymbol{x} - \boldsymbol{x}'\|_2$, where $\boldsymbol{z} \in \mathcal{N}_{\boldsymbol{\mathcal{X}}}(\boldsymbol{x}))$, for any smooth game, we have

$$r^{dg}(\boldsymbol{x}) = \max_{\boldsymbol{x}' \in \mathcal{X}} \langle \boldsymbol{\ell^x}, \boldsymbol{x} - \boldsymbol{x}' \rangle \leq C_1 r^{tan}(\boldsymbol{x}) = C_1 \min_{\boldsymbol{z} \in \mathcal{N}_{\boldsymbol{\mathcal{X}}}(\boldsymbol{x})} \|\boldsymbol{\ell^x} + \boldsymbol{z}\|_2,$$

where $C_1$ is a game-dependent constant. Recall the definition of the weak MVI

$$\langle \boldsymbol{\ell^x} + \boldsymbol{z}, \boldsymbol{x} - \boldsymbol{x}^* \rangle \geq \rho \|\boldsymbol{\ell^x} + \boldsymbol{z}\|_2^2, \forall \boldsymbol{z} \in \mathcal{N}_{\boldsymbol{\mathcal{X}}}(\boldsymbol{x}).$$

Therefore, if we can show that

$$\langle \boldsymbol{\ell^x}, \boldsymbol{x} - \boldsymbol{x}^* \rangle \geq -(r^{dg}(\boldsymbol{x}))^2,$$

we can always find a $\rho = -C_1^2 < 0$ to ensure the weak MVI holds since $\forall \boldsymbol{z} \in \mathcal{N}_{\boldsymbol{\mathcal{X}}}(\boldsymbol{x})$,

$$\langle \boldsymbol{\ell^x}, \boldsymbol{x} - \boldsymbol{x}^* \rangle \geq -(r^{dg}(\boldsymbol{x}))^2 = -(\max_{\boldsymbol{x}' \in \mathcal{X}} \langle \boldsymbol{\ell^x}, \boldsymbol{x} - \boldsymbol{x}' \rangle)^2 \geq -(C_1 r^{tan}(\boldsymbol{x}))^2 = -C_1^2 \min_{\boldsymbol{z}' \in \mathcal{N}_{\boldsymbol{\mathcal{X}}}(\boldsymbol{x})} \|\boldsymbol{\ell^x} + \boldsymbol{z}'\|_2^2 \geq -C_1^2 \|\boldsymbol{\ell^x} + \boldsymbol{z}\|_2^2,$$

and

$$\langle \boldsymbol{z}, \boldsymbol{x} - \boldsymbol{x}^* \rangle \geq 0.$$

Now, we show that $\langle \boldsymbol{\ell^x}, \boldsymbol{x} - \boldsymbol{x}^* \rangle \geq -(r^{dg}(\boldsymbol{x}))^2$ holds in this game. We define the strategies of players as following

$$\boldsymbol{x}_0 = \begin{pmatrix} a \\ 1-a \end{pmatrix}, \quad \boldsymbol{x}_1 = \begin{pmatrix} b \\ 1-b \end{pmatrix},$$

where $0 \leq a \leq 1$ and $0 \leq b \leq 1$. The loss gradient $\boldsymbol{\ell}_i^x$ of player $i$ is

$$\boldsymbol{\ell}_0^x = -\boldsymbol{A}\boldsymbol{x}_1, \quad \boldsymbol{\ell}_1^x = -\boldsymbol{B}^{\mathrm{T}}\boldsymbol{x}_0.$$

Formally, for player 0, we have

$$\boldsymbol{A}\boldsymbol{x}_1 = \begin{pmatrix} 1 & 0 \\ -1 & 1 \end{pmatrix} \begin{pmatrix} b \\ 1-b \end{pmatrix} = \begin{pmatrix} b \\ -b+1-b \end{pmatrix} = \begin{pmatrix} b \\ 1-2b \end{pmatrix},$$

$$\boldsymbol{\ell}_0^x = -\boldsymbol{A}\boldsymbol{x}_1 = \begin{pmatrix} -b \\ -(1-2b) \end{pmatrix} = \begin{pmatrix} -b \\ 2b-1 \end{pmatrix}.$$

Similarly, for player 1, we have

$$\boldsymbol{B}^{\mathrm{T}} = \begin{pmatrix} 0 & -1 \\ 1 & 1 \end{pmatrix},$$

$$\boldsymbol{B}^{\mathrm{T}}\boldsymbol{x}_0 = \begin{pmatrix} 0 & -1 \\ 1 & 1 \end{pmatrix} \begin{pmatrix} a \\ 1-a \end{pmatrix} = \begin{pmatrix} -(1-a) \\ a+(1-a) \end{pmatrix} = \begin{pmatrix} a-1 \\ 1 \end{pmatrix},$$

$$\boldsymbol{\ell}_1^x = -\boldsymbol{B}^{\mathrm{T}}\boldsymbol{x}_0 = \begin{pmatrix} 1-a \\ -1 \end{pmatrix}.$$

Now, we show $\langle \boldsymbol{\ell^x}, \boldsymbol{x} - \boldsymbol{x}^* \rangle \geq -(r^{dg}(\boldsymbol{x}))^2$ by showing $\langle \boldsymbol{\ell^x}, \boldsymbol{x} - \boldsymbol{x}^* \rangle + (r^{dg}(\boldsymbol{x}))^2 \geq 0$ holds. We first compute $\langle \boldsymbol{\ell^x}, \boldsymbol{x} - \boldsymbol{x}^* \rangle$. Formally, we get

$$\boldsymbol{x}_0 - \boldsymbol{x}_0^* = \begin{pmatrix} a \\ 1-a \end{pmatrix} - \begin{pmatrix} 0 \\ 1 \end{pmatrix} = \begin{pmatrix} a \\ -a \end{pmatrix},$$

$$\boldsymbol{x}_1 - \boldsymbol{x}_1^* = \begin{pmatrix} b \\ 1-b \end{pmatrix} - \begin{pmatrix} 0 \\ 1 \end{pmatrix} = \begin{pmatrix} b \\ -b \end{pmatrix}.$$

Next, calculate the dot products

$$\langle \boldsymbol{\ell}_0^x, \boldsymbol{x}_0 - \boldsymbol{x}_0^* \rangle = \begin{pmatrix} -b \\ 2b-1 \end{pmatrix} \cdot \begin{pmatrix} a \\ -a \end{pmatrix} = -ab - a(2b-1) = -3ab + a,$$

$$\langle \boldsymbol{\ell}_1^x, \boldsymbol{x}_1 - \boldsymbol{x}_1^* \rangle = \begin{pmatrix} 1-a \\ -1 \end{pmatrix} \cdot \begin{pmatrix} b \\ -b \end{pmatrix} = b(1-a) + b = b(1-a+1) = b(2-a).$$

Combine the results

$$\langle \boldsymbol{\ell^x}, \boldsymbol{x} - \boldsymbol{x}^* \rangle = \langle \boldsymbol{\ell}_0^x, \boldsymbol{x}_0 - \boldsymbol{x}_0^* \rangle + \langle \boldsymbol{\ell}_1^x, \boldsymbol{x}_1 - \boldsymbol{x}_1^* \rangle = -3ab + a + b(2-a).$$

This simplifies to:

$$\langle \boldsymbol{\ell^x}, \boldsymbol{x} - \boldsymbol{x}^* \rangle = -3ab + a + 2b - ab = -4ab + 2b + a.$$

Similarly, for $r^{dg}(\boldsymbol{x}) = \max_{\boldsymbol{x}' \in \mathcal{X}} \langle \boldsymbol{\ell^x}, \boldsymbol{x} - \boldsymbol{x}' \rangle$, we get

$$\max_{\boldsymbol{x}' \in \mathcal{X}} \langle \boldsymbol{\ell^x}, \boldsymbol{x} - \boldsymbol{x}' \rangle = \langle \boldsymbol{\ell_0^x}, \boldsymbol{x}_0 \rangle - \min(\boldsymbol{\ell_0^x}[0], \boldsymbol{\ell_0^x}[1]) + \langle \boldsymbol{\ell_1^x}, \boldsymbol{x}_1 \rangle - \min(\boldsymbol{\ell_1^x}[0], \boldsymbol{\ell_1^x}[1]),$$

which results in

$$\max_{\boldsymbol{x}' \in \mathcal{X}} \langle \boldsymbol{\ell^x}, \boldsymbol{x} - \boldsymbol{x}' \rangle = -4ab + 4b - 2 + a - \min(-b, 2b-1) - \min(1-a, -1) = -4ab + 4b - 1 + a - \min(-b, 2b-1)$$

**Case 1:** If $0 \le b \le \frac{1}{3}$,

$$\langle \boldsymbol{\ell^x}, \boldsymbol{x} - \boldsymbol{x}^* \rangle + (r^{dg}(\boldsymbol{x}))^2 = -4ab + 2b + a + (-4ab + 4b - 1 + a - 2b + 1)^2 = -4ab + 2b + a + (-4ab + 2b + a)^2.$$

It is obviously if $-4ab + 2b + a \ge 0$, $-4ab + 2b + a + (-4ab + 4b - 1 + a - 2b + 1)^2 = -4ab + 2b + a + (-4ab + 2b + a)^2 \ge 0$. Now, we show $-4ab + 2b + a \ge 0$. Formally, we get

$$-4ab + 2b + a = (-4a + 2)b + a.$$

We can find that $(-4a + 2)b + a$ is linear function w.r.t $b$ given fixed $a$. If $1 \ge a \ge \frac{1}{2}$, $(-4a + 2)b + a$ decreases as $b$ increases. Therefore, given $1 \ge a \ge \frac{1}{2}$, $\min_{0 \le b \le \frac{1}{3}}(-4a + 2)b + a = (-4a + 2)\frac{1}{3} + a = \frac{2}{3} - \frac{a}{3} \ge \frac{1}{3}$. Similarly, if $0 \le a < \frac{1}{2}$, $(-4a + 2)b + a$ decreases as $b$ decreases. Therefore, given $0 \le a < \frac{1}{2}$, $\min_{0 \le b \le \frac{1}{3}}(-4a + 2)b + a = a \ge 0$. Hence, we get $-4ab + 2b + a \ge 0$, which implies $-4ab + 2b + a + (-4ab + 4b - 1 + a - 2b + 1)^2 = -4ab + 2b + a + (-4ab + 2b + a)^2 \ge 0$.

**Case 2:** If $\frac{1}{3} \le b \le 1$,

$$\langle \boldsymbol{\ell^x}, \boldsymbol{x} - \boldsymbol{x}^* \rangle + (r^{dg}(\boldsymbol{x}))^2 = -4ab + 2b + a + (-4ab + 4b - 1 + a + b)^2 = -4ab + 2b + a + (-4ab + 5b + a - 1)^2.$$

Now, we simplify the expression

$$-4ab + 2b + a + (-4ab + 5b + a - 1)^2.$$

Then,

$$(-4ab + 5b + a - 1)^2$$
$$= (-4ab + 5b + a - 1)(-4ab + 5b + a - 1)$$
$$= (-4ab)^2 + (5b)^2 + a^2 + (-1)^2 + 2(-4ab \cdot 5b) + 2(-4ab \cdot a) + 2(-4ab \cdot -1) + 2(5b \cdot a) + 2(5b \cdot -1) + 2(a \cdot -1)$$
$$= 16a^2b^2 + 25b^2 + a^2 + 1 - 40ab^2 - 8a^2b + 8ab + 10ab - 10b - 2a$$
$$= 16a^2b^2 + 25b^2 + a^2 - 40ab^2 - 8a^2b + 18ab - 10b - 2a + 1.$$

So the full expression is

$$-4ab + 2b + a + 16a^2b^2 + 25b^2 + a^2 - 40ab^2 - 8a^2b + 18ab - 10b - 2a + 1.$$

Therefore, we define

$$f(a) = (16b^2 - 8b + 1)a^2 + (-40b^2 + 14b - 1)a + 25b^2 - 8b + 1.$$

For $f(a)$, given a fixed $b$, it is a quadratic function with respect to $a$. For the term $32b^2 - 16b + 2$, as it takes the minimum value when $b = \frac{16}{64} = \frac{1}{4}$, we have that the value of $32b^2 - 16b + 2$ increases as $b$ increases when $\frac{1}{3} \le b \le 1$. Therefore, the minimum and maximum values of $32b^2 - 16b + 2$ when $\frac{1}{3} \le b \le 1$ are $32\frac{1}{9} - \frac{16}{3} + 2 = \frac{2}{9}$ and $32 - 16 + 2 = 18$, respectively. As $32b^2 - 16b + 2 > 0$, for $f(a)$, given a fixed $b$, so it takes the minimum value in the following case

$$a = \frac{40b^2 - 14b + 1}{32b^2 - 16b + 2} = \frac{32b^2 - 16b + 2 + 8b^2 + 2b - 1}{32b^2 - 16b + 2} = 1 + \frac{8b^2 + 2b - 1}{32b^2 - 16b + 2}.$$

For the term $8b^2 + 2b - 1$, as it takes the minimum value when $b = \frac{-2}{16} \le 0$, we have that the value of $8b^2 + 2b - 1$ increases as $b$ increases when $\frac{1}{3} \le b \le 1$. Therefore, the minimum value of $8b^2 + 2b - 1$

when $\frac{1}{3} \le b \le 1$ is $8\frac{1}{9} + \frac{2}{3} - 1 = \frac{5}{9}$. Combining $8b^2 + 2b - 1 \ge \frac{5}{9}$ and $18 \ge 32b^2 - 16b + 2 \ge \frac{2}{9}$, we have

$$1 + \frac{8b^2 + 2b - 1}{32b^2 - 16b + 2} \ge 1.$$

Therefore, given a fixed $b$, $f(a)$ takes the minimum value when $a = 1$. Therefore, we get

$$f(1) = 16b^2 - 8b + 1 - 40b^2 + 14b - 1 + 25b^2 - 8b + 1 = b^2 - 2b + 1 \ge 0, \forall \frac{1}{3} \le b \le 1.$$

**Conclusion:** Combining the results in **Case 1** and **Case 2**, we have

$$\langle \boldsymbol{\ell^x}, \boldsymbol{x} - \boldsymbol{x}^* \rangle + (r^{dg}(\boldsymbol{x}))^2 \ge 0.$$

Therefore, we get $\forall \boldsymbol{z} \in \mathcal{N}_{\boldsymbol{\mathcal{X}}}(\boldsymbol{x})$,

$$\langle \boldsymbol{\ell^x}, \boldsymbol{x} - \boldsymbol{x}^* \rangle \ge -(r^{dg}(\boldsymbol{x}))^2 = -(\max_{\boldsymbol{x}' \in \mathcal{X}} \langle \boldsymbol{\ell^x}, \boldsymbol{x} - \boldsymbol{x}' \rangle)^2 \ge -(C_1 r^{tan}(\boldsymbol{x}))^2 = -C_1^2 \min_{\boldsymbol{z}' \in \mathcal{N}_{\boldsymbol{\mathcal{X}}}(\boldsymbol{x})} \|\boldsymbol{\ell^x} + \boldsymbol{z}'\|_2^2$$
$$\ge -C_1^2 \|\boldsymbol{\ell^x} + \boldsymbol{z}\|_2^2.$$

In addition, from the definition of the normal cone, we have

$$\langle \boldsymbol{z}, \boldsymbol{x} - \boldsymbol{x}^* \rangle \ge 0, \forall \boldsymbol{z} \in \mathcal{N}_{\boldsymbol{\mathcal{X}}}(\boldsymbol{x}).$$

Combining the above results, we obtain

$$\langle \boldsymbol{\ell^x} + \boldsymbol{z}, \boldsymbol{x} - \boldsymbol{x}^* \rangle \ge -(r^{dg}(\boldsymbol{x}))^2 \ge -C_1^2 \|\boldsymbol{\ell^x} + \boldsymbol{z}\|_2^2, \forall \boldsymbol{z} \in \mathcal{N}_{\boldsymbol{\mathcal{X}}}(\boldsymbol{x}),$$

which means the weak MVI holds in this game with $\rho = -C_1^2$.

# H   PSEUDOCODE OF RM$^+$ VARIANTS MENTIONED IN THIS PAPER

Now, we provide the pseudocode of RM$^+$ variants mentioned in this paper. Specifically, the pseudocode of RM$^+$, SExRM$^+$, SPRM$^+$, and SOGRM$^+$ are shown in Algorithm algorithm 1, 2, 3, and 4, respectively.

---

**Algorithm 1** RM$^+$

---

**Require:** Step size $\eta \in (0, \infty)$.
1: Initialize: $\boldsymbol{\theta}_i^1 \leftarrow \mathbf{1}/|A_i|, \forall i \in \mathcal{N}$
2: **for** $t = 1, 2, \dots$ **do**
3:     **for** $i \in \mathcal{N}$ **do**
4:         $\boldsymbol{\theta}_i^{t+1} = [\boldsymbol{\theta}_i^t + \eta \boldsymbol{F}_i^t(\boldsymbol{\theta}^t)]^+$
5:     **end for**
6: **end for**

---

**Algorithm 2** SExRM$^+$

---

**Require:** Step size $\eta \in \left(0, \frac{1}{DL_u}\right)$.
1: Initialize: $\boldsymbol{\theta}_i^1 \leftarrow \mathbf{1}/|A_i|, \forall i \in \mathcal{N}$
2: **for** $t = 1, 2, \dots$ **do**
3:     **for** $i \in \mathcal{N}$ **do**
4:         $\boldsymbol{\theta}_i^{t+\frac{1}{2}} \in \arg\min_{\boldsymbol{\theta}_i \in \mathbb{R}_{\ge 1}^{|A_i|}} \{\langle -\boldsymbol{F}_i(\boldsymbol{\theta}^t), \boldsymbol{\theta}_i \rangle + \frac{1}{\eta} D_{\psi}(\boldsymbol{\theta}_i, \boldsymbol{\theta}_i^t)\}, \ \boldsymbol{x}_i^{t+\frac{1}{2}} = \frac{\boldsymbol{\theta}_i^{t+\frac{1}{2}}}{\|\boldsymbol{\theta}_i^{t+\frac{1}{2}}\|_1}$
5:     **end for**
6:     **for** $i \in \mathcal{N}$ **do**
7:         $\boldsymbol{\theta}_i^{t+1} \in \arg\min_{\boldsymbol{\theta}_i \in \mathbb{R}_{\ge 1}^{|A_i|}} \{\langle -\boldsymbol{F}_i(\boldsymbol{\theta}^{t+\frac{1}{2}}), \boldsymbol{\theta}_i \rangle + \frac{1}{\eta} D_{\psi}(\boldsymbol{\theta}_i, \boldsymbol{\theta}_i^t)\}, \ \boldsymbol{x}_i^{t+1} = \frac{\boldsymbol{\theta}_i^{t+1}}{\|\boldsymbol{\theta}_i^{t+1}\|_1}$
8:     **end for**
9: **end for**

---

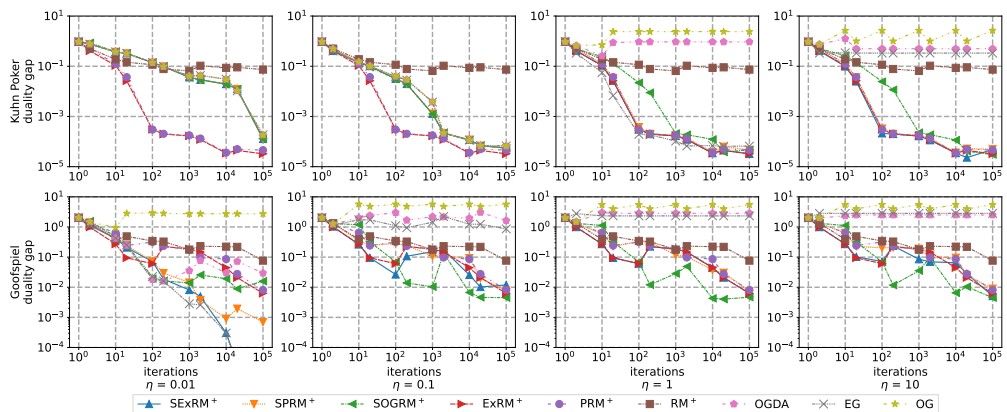

Figure 2: Performance of different algorithms in Kuhn Poker (top) and Goofspiel (bottom).

---

**Algorithm 3** SPRM$^+$

---

**Require:** Step size $\eta \in \left(0, \frac{1}{8DL_u}\right)$.

1: Initialize: $\boldsymbol{\theta}_i^{\frac{1}{2}} \leftarrow \mathbf{1}/|A_i|, \boldsymbol{\theta}_i^1 \leftarrow \mathbf{1}/|A_i|, \forall i \in \mathcal{N}$
2: **for** $t = 1, 2, \ldots$ **do**
3:      **for** $i \in \mathcal{N}$ **do**
4:         $\boldsymbol{\theta}_i^{t+\frac{1}{2}} \in \arg\min_{\boldsymbol{\theta}_i \in \mathbb{R}_{\geq 1}^{|A_i|}} \{\langle -\boldsymbol{F}_i(\boldsymbol{\theta}^{t-\frac{1}{2}}), \boldsymbol{\theta}_i \rangle + \frac{1}{\eta} D_\psi(\boldsymbol{\theta}_i, \boldsymbol{\theta}_i^t)\}, \boldsymbol{x}_i^{t+\frac{1}{2}} = \frac{\boldsymbol{\theta}_i^{t+\frac{1}{2}}}{\|\boldsymbol{\theta}_i^{t+\frac{1}{2}}\|_1}$
5:      **end for**
6:      **for** $i \in \mathcal{N}$ **do**
7:         $\boldsymbol{\theta}_i^{t+1} \in \arg\min_{\boldsymbol{\theta}_i \in \mathbb{R}_{\geq 1}^{|A_i|}} \{\langle -\boldsymbol{F}_i(\boldsymbol{\theta}^{t+\frac{1}{2}}), \boldsymbol{\theta}_i \rangle + \frac{1}{\eta} D_\psi(\boldsymbol{\theta}_i, \boldsymbol{\theta}_i^t)\}, \boldsymbol{x}_i^{t+1} = \frac{\boldsymbol{\theta}_i^{t+1}}{\|\boldsymbol{\theta}_i^{t+1}\|_1}$
8:      **end for**
9: **end for**

---

**Algorithm 4** SOGRM$^+$

---

**Require:** Step size $\eta \in \left(0, \frac{1}{2DL_u}\right)$.

1: Initialize: $\boldsymbol{\theta}_i^{\frac{1}{2}} \leftarrow \mathbf{1}/|A_i|, \boldsymbol{\theta}_i^1 \leftarrow \mathbf{1}/|A_i|, \forall i \in \mathcal{N}$
2: **for** $t = 1, 2, \ldots$ **do**
3:      **for** $i \in \mathcal{N}$ **do**
4:         $\boldsymbol{\theta}_i^{t+\frac{1}{2}} \in \arg\min_{\boldsymbol{\theta}_i \in \mathbb{R}_{\geq 1}^{|A_i|}} \{\langle -\boldsymbol{F}_i(\boldsymbol{\theta}^{t-\frac{1}{2}}), \boldsymbol{\theta}_i \rangle + \frac{1}{\eta} D_\psi(\boldsymbol{\theta}_i, \boldsymbol{\theta}_i^t)\}, \boldsymbol{x}_i^{t+\frac{1}{2}} = \frac{\boldsymbol{\theta}_i^{t+\frac{1}{2}}}{\|\boldsymbol{\theta}_i^{t+\frac{1}{2}}\|_1}$
5:      **end for**
6:      **for** $i \in \mathcal{N}$ **do**
7:         $\boldsymbol{\theta}_i^{t+1} = \boldsymbol{\theta}_i^{t+\frac{1}{2}} - \eta \boldsymbol{F}_i(\boldsymbol{\theta}^{t-\frac{1}{2}}) + \eta \boldsymbol{F}_i(\boldsymbol{\theta}^{t+\frac{1}{2}})$
8:      **end for**
9: **end for**

---

# I  ADDITIONAL EXPERIMENTAL RESULTS

In this section, we present experimental results on (i) the normal-form representation of two extensive-form games, Kuhn Poker and Goofspiel, and (ii) randomly generated three-player zero-sum poly-matrix games with sizes $[10, 20, 50]$. Notably, polymatrix games are classical games with satisfying monotonicity (Pérolat et al., 2021). The normal-form representations of the two extensive-form games are derived from the open-source code provided by Cai et al. (2023) (https://openreview.net/forum?id=LWeVVPuIx0&noteId=4vbVJryMNi&referrer=%5BTasks%5D(%2Ftasks)). The payoff matrices for Kuhn Poker and Goofspiel are of sizes $[27, 64]$ and $[72, 7808]$, respectively. For the randomly generated three-player zero-sum polymatrix games, as did in Section 6, we generate 20

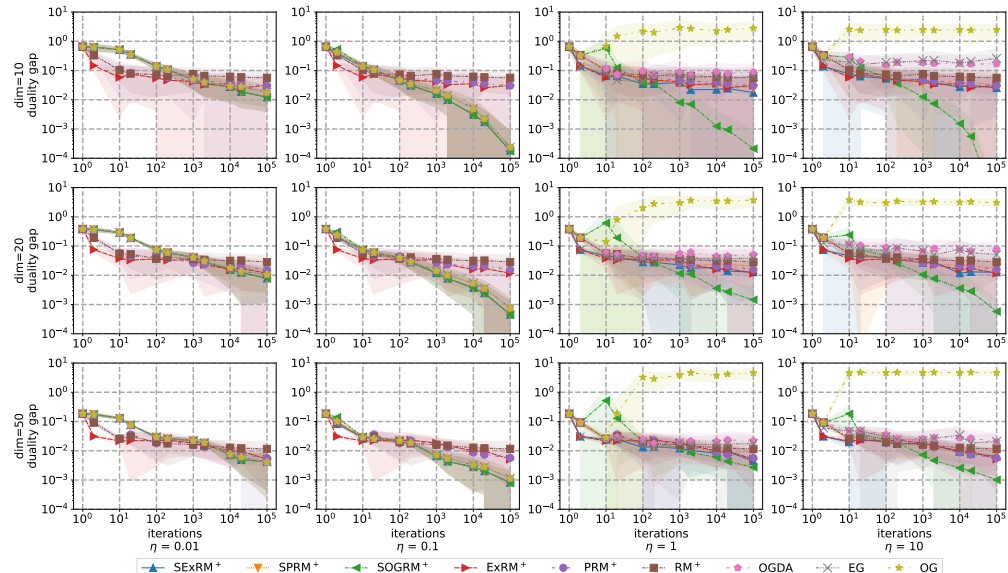

Figure 3: Performance of different algorithms in $10 \times 10$ (top), $20 \times 20$ (middle), $50 \times 50$ (bottom) randomly generated three-player zero-sum polymatrix games.

instances for each size. In the randomly generated three-player zero-sum, the payoff matrix for each pair of players is a diagonal matrix, with each diagonal element sampled from a standard normal distribution.

**Convergence Performance on Kuhn Poker and Goofspiel.** The results are shown in Figure 2 and consistent with those presented in Section 6. OGDA, EG, and OG exhibit poorer convergence performance and higher sensitivity to hyperparameters compared to their corresponding smooth $RM^+$ variants ($SPRM^+$, $SExRM^+$, and $SOGRM^+$, respectively). Moreover, we observe that OG fails to converge in Goofspiel for any set of parameters. We hypothesize that this is due to the significantly larger scale of Goofspiel compared to the other games tested, requiring OG to use a much smaller learning rate $\eta$ for convergence. In contrast, $SOGRM^+$ demonstrates lower sensitivity to hyperparameters, consistently exhibiting convergence across all parameter settings.

**Convergence Performance on randomly generated three-player zero-sum polymatrix games.** The experimental results are shown in Figure 3. Consistent with the results in Figure 1 and Figure 2, the smooth $RM^+$ variants generally exhibit superior convergence performance and reduced sensitivity to hyperparameters compared to their corresponding OMD algorithms. However, we also observe that the OG tends to diverge significantly when $\eta \geq 1$. In contrast, $SOGRM^+$, consistent with previous experimental findings, demonstrates low sensitivity to parameters and retains strong convergence even for $\eta \geq 1$.

## J DISCUSSION OF THE REASON WHY $SOGRM^+$ ALLOWS LARGE $\eta$ COMPARED TO OTHER $RM^+$ VARIANTS

For the reason why $SOGRM^+$ allows large $\eta$ compared to other $RM^+$ variants, we hypothesize that it arises because our proposed algorithm, $SOGRM^+$, performs only a single prox-mapping operator per update step, unlike other smooth $RM^+$ algorithms, which involve two prox-mapping operations at each iteration (the first occurrence of the prox-mapping operator is in the introduction of OMD, Section 2).

Specifically, the prox-mapping operator in smooth $RM^+$ variants (such as $SExRM^+$, $SPRM^+$, and $SOGRM^+$) involves a projection onto the simplex at sometimes (Farina et al., 2023) (not always as in OMD algorithms), which may lead to significant changes in $\theta$ depending on the choice of $\eta$. In contrast, the update rule of $SOGRM^+$ (in the second line) omits this prox-mapping operator and instead relies solely on simple addition and subtraction operations. As a result, the initial parameter $\theta_0$ may become negligible compared to the term $\eta F_i(\theta)$. Thus, the values of $\theta_i$ in

SOGRM$^+$ are likely to vary in direct proportion to $\eta$, and the resulting strategy $x_i = \theta_i/\|\theta_i\|_1$ will exhibit a more stable behavior with respect to changes in $\eta$. Therefore, for different values of $\eta$, the sequence of strategies generated by SOGRM$^+$ exhibits small differences. Moreover, when $\eta$ is small, Theorem 5.1 guarantees thatthe sequence of strategies produced by SOGRM$^+$ converges to the set of NE. Consequently, SOGRM$^+$ permits the use of larger $\eta$ values compared to other algorithms.

To validate our statement, as demonstrated in Section 6, we conducted evaluations on 20 randomly generated 10-dimensional two-player zero-sum matrix games. Specifically, we analyzed the strategies of Player 0 output by SExRM$^+$, SPRM$^+$, and SOGRM$^+$ at iterations 1, 10, 100, 1000, and 10,000. To mitigate randomness, we averaged the strategies across the 20 instances. The results clearly show that for different values of $\eta$, the sequence of strategies generated by SOGRM$^+$ exhibits minimal variation. Notably, when $\eta \geq 1$ and the number of iterations $\geq 1000$, the strategies produced by SOGRM$^+$ are nearly identical across different values of $\eta$. This behavior is not observed in the other two RM$^+$ variants.

Table 2: The sequence of strategies generated by SExRM$^+$.

```
eta=0.01-------------------------------------------------------
iteration 1
[0.100000, 0.100000, 0.100000, 0.100000, 0.100000, 0.100000, 0.100000, 0.100000, 0.100000, 0.100000]
iteration 10
[0.098491, 0.100056, 0.109717, 0.106483, 0.096661, 0.091065, 0.094224, 0.101697, 0.107110, 0.094496]
iteration 100
[0.115888, 0.047996, 0.172965, 0.159585, 0.078764, 0.017972, 0.063801, 0.116942, 0.204308, 0.021778]
iteration 1000
[0.247108, 0.000000, 0.114474, 0.070481, 0.000000, 0.000000, 0.082859, 0.185139, 0.299941, 0.000000]
iteration 10000
[0.269960, 0.000000, 0.144571, 0.049592, 0.024670, 0.000000, 0.080797, 0.145601, 0.284809, 0.000000]
iteration 100000
[0.277022, 0.000000, 0.141075, 0.056758, 0.011324, 0.000000, 0.081395, 0.152016, 0.280411, 0.000000]
eta=0.1--------------------------------------------------------
iteration 1
[0.100000, 0.100000, 0.100000, 0.100000, 0.100000, 0.100000, 0.100000, 0.100000, 0.100000, 0.100000]
iteration 10
[0.115382, 0.045286, 0.170913, 0.159142, 0.080181, 0.018597, 0.065633, 0.117780, 0.205713, 0.021372]
iteration 100
[0.244384, 0.000000, 0.121762, 0.065640, 0.000000, 0.000000, 0.088998, 0.180417, 0.298800, 0.000000]
iteration 1000
[0.271157, 0.000000, 0.144291, 0.052641, 0.017717, 0.000000, 0.084207, 0.145175, 0.284811, 0.000000]
iteration 10000
[0.271794, 0.000000, 0.142475, 0.054832, 0.012438, 0.000000, 0.083277, 0.150720, 0.284463, 0.000000]
iteration 100000
[0.271736, 0.000000, 0.142474, 0.054823, 0.012477, 0.000000, 0.083266, 0.150734, 0.284490, 0.000000]
eta=1----------------------------------------------------------
iteration 1
[0.100000, 0.100000, 0.100000, 0.100000, 0.100000, 0.100000, 0.100000, 0.100000, 0.100000, 0.100000]
iteration 10
[0.250238, 0.000000, 0.141786, 0.047503, 0.000000, 0.000000, 0.071202, 0.174680, 0.314590, 0.000000]
iteration 100
[0.272172, 0.000000, 0.107236, 0.031325, 0.012185, 0.000000, 0.098432, 0.173866, 0.304784, 0.000000]
iteration 1000
[0.271736, 0.000000, 0.149928, 0.056319, 0.012477, 0.000000, 0.078212, 0.149291, 0.282037, 0.000000]
iteration 10000
[0.271736, 0.000000, 0.151644, 0.057013, 0.012477, 0.000000, 0.076993, 0.148728, 0.281409, 0.000000]
iteration 100000
[0.271736, 0.000000, 0.142474, 0.054823, 0.012477, 0.000000, 0.083266, 0.150734, 0.284490, 0.000000]
eta=10---------------------------------------------------------
iteration 1
[0.100000, 0.100000, 0.100000, 0.100000, 0.100000, 0.100000, 0.100000, 0.100000, 0.100000, 0.100000]
iteration 10
[0.269329, 0.000000, 0.000000, 0.040992, 0.028389, 0.000000, 0.153818, 0.197411, 0.310061, 0.000000]
iteration 100
[0.267860, 0.000000, 0.142353, 0.054205, 0.019992, 0.000000, 0.066032, 0.178721, 0.270838, 0.000000]
iteration 1000
[0.282710, 0.000000, 0.142474, 0.054823, 0.006415, 0.000000, 0.095782, 0.118767, 0.299029, 0.000000]
iteration 10000
[0.270484, 0.000000, 0.142474, 0.054823, 0.013252, 0.000000, 0.082598, 0.153161, 0.283207, 0.000000]
iteration 100000
[0.280316, 0.000000, 0.142474, 0.054823, 0.006891, 0.000000, 0.095685, 0.118429, 0.301383, 0.000000]
```

Table 3: The sequence of strategies generated by SPRM$^+$.

```
eta=0.01-------------------------------------------------------
iteration 1
[0.100000, 0.100000, 0.100000, 0.100000, 0.100000, 0.100000, 0.100000, 0.100000, 0.100000, 0.100000]
iteration 10
[0.098491, 0.100056, 0.109718, 0.106484, 0.096661, 0.091065, 0.094224, 0.101697, 0.107110, 0.094496]
iteration 100
[0.115888, 0.047996, 0.172965, 0.159585, 0.078764, 0.017972, 0.063801, 0.116942, 0.204308, 0.021778]
iteration 1000
[0.247107, 0.000000, 0.114474, 0.070481, 0.000000, 0.000000, 0.082859, 0.185139, 0.299941, 0.000000]
iteration 10000
[0.269960, 0.000000, 0.144571, 0.049592, 0.024670, 0.000000, 0.080797, 0.145601, 0.284810, 0.000000]
iteration 100000
[0.277022, 0.000000, 0.141075, 0.056758, 0.011324, 0.000000, 0.081394, 0.152016, 0.280411, 0.000000]
eta=0.1-------------------------------------------------------
iteration 1
[0.100000, 0.100000, 0.100000, 0.100000, 0.100000, 0.100000, 0.100000, 0.100000, 0.100000, 0.100000]
iteration 10
[0.115372, 0.045123, 0.170807, 0.159184, 0.080205, 0.018616, 0.065781, 0.117766, 0.205911, 0.021235]
iteration 100
[0.244321, 0.000000, 0.121870, 0.065578, 0.000000, 0.000000, 0.089146, 0.180253, 0.298833, 0.000000]
iteration 1000
[0.271172, 0.000000, 0.144289, 0.052646, 0.017681, 0.000000, 0.084196, 0.145199, 0.284817, 0.000000]
iteration 10000
[0.271796, 0.000000, 0.142475, 0.054831, 0.012437, 0.000000, 0.083275, 0.150721, 0.284465, 0.000000]
iteration 100000
[0.271736, 0.000000, 0.142474, 0.054823, 0.012477, 0.000000, 0.083266, 0.150734, 0.284490, 0.000000]
eta=1-------------------------------------------------------
iteration 1
[0.100000, 0.100000, 0.100000, 0.100000, 0.100000, 0.100000, 0.100000, 0.100000, 0.100000, 0.100000]
iteration 10
[0.237655, 0.000000, 0.000000, 0.000000, 0.000000, 0.000000, 0.236226, 0.203216, 0.322903, 0.000000]
iteration 100
[0.275397, 0.000000, 0.137955, 0.053723, 0.009448, 0.000000, 0.085151, 0.122804, 0.315521, 0.000000]
iteration 1000
[0.261815, 0.000000, 0.142474, 0.054823, 0.019802, 0.000000, 0.068201, 0.186105, 0.266779, 0.000000]
iteration 10000
[0.265631, 0.000000, 0.142474, 0.054823, 0.016390, 0.000000, 0.069668, 0.180080, 0.270934, 0.000000]
iteration 100000
[0.266516, 0.000000, 0.142474, 0.054823, 0.015538, 0.000000, 0.072873, 0.173694, 0.274082, 0.000000]
eta=10-------------------------------------------------------
iteration 1
[0.100000, 0.100000, 0.100000, 0.100000, 0.100000, 0.100000, 0.100000, 0.100000, 0.100000, 0.100000]
iteration 10
[0.227614, 0.000000, 0.000000, 0.000000, 0.035267, 0.000000, 0.172873, 0.288558, 0.275688, 0.000000]
iteration 100
[0.271127, 0.000000, 0.138072, 0.053705, 0.000000, 0.000000, 0.098440, 0.122633, 0.316024, 0.000000]
iteration 1000
[0.272132, 0.000000, 0.142474, 0.054823, 0.013019, 0.000000, 0.083662, 0.145232, 0.288658, 0.000000]
iteration 10000
[0.265599, 0.000000, 0.142474, 0.054823, 0.016434, 0.000000, 0.069693, 0.180184, 0.270793, 0.000000]
iteration 100000
[0.267444, 0.000000, 0.142474, 0.054823, 0.015058, 0.000000, 0.079408, 0.161766, 0.279027, 0.000000]
```

Table 4: The sequence of strategies generated by SOGRM$^+$.

```
eta=0.01------------------------------------------------------
iteration 1
[0.100000, 0.100000, 0.100000, 0.100000, 0.100000, 0.100000, 0.100000, 0.100000, 0.100000, 0.100000]
iteration 10
[0.098649, 0.100078, 0.108744, 0.105828, 0.096983, 0.091963, 0.094790, 0.101526, 0.106359, 0.095079]
iteration 100
[0.115373, 0.048574, 0.172920, 0.159312, 0.079077, 0.018314, 0.063757, 0.117009, 0.203262, 0.022401]
iteration 1000
[0.247102, 0.000000, 0.114400, 0.070508, 0.000000, 0.000000, 0.082918, 0.185222, 0.299849, 0.000000]
iteration 10000
[0.269951, 0.000000, 0.144571, 0.049631, 0.024661, 0.000000, 0.080760, 0.145596, 0.284830, 0.000000]
iteration 100000
[0.277013, 0.000000, 0.141074, 0.056758, 0.011325, 0.000000, 0.081401, 0.152020, 0.280409, 0.000000]
eta=0.1-------------------------------------------------------
iteration 1
[0.100000, 0.100000, 0.100000, 0.100000, 0.100000, 0.100000, 0.100000, 0.100000, 0.100000, 0.100000]
iteration 10
[0.109695, 0.054811, 0.168579, 0.154984, 0.082590, 0.023255, 0.067513, 0.117496, 0.194750, 0.026325]
iteration 100
[0.244066, 0.000000, 0.121463, 0.065806, 0.000000, 0.000000, 0.089626, 0.180912, 0.298127, 0.000000]
iteration 1000
[0.271160, 0.000000, 0.144287, 0.052636, 0.017740, 0.000000, 0.084047, 0.145153, 0.284977, 0.000000]
iteration 10000
[0.271809, 0.000000, 0.142475, 0.054831, 0.012428, 0.000000, 0.083271, 0.150721, 0.284465, 0.000000]
iteration 100000
[0.271736, 0.000000, 0.142474, 0.054823, 0.012477, 0.000000, 0.083266, 0.150734, 0.284490, 0.000000]
eta=1---------------------------------------------------------
iteration 1
[0.100000, 0.100000, 0.100000, 0.100000, 0.100000, 0.100000, 0.100000, 0.100000, 0.100000, 0.100000]
iteration 10
[0.143554, 0.000000, 0.000000, 0.019365, 0.081302, 0.059547, 0.165077, 0.274010, 0.246277, 0.010868]
iteration 100
[0.272928, 0.000000, 0.140918, 0.055096, 0.011850, 0.000000, 0.079330, 0.152495, 0.287383, 0.000000]
iteration 1000
[0.271736, 0.000000, 0.142474, 0.054823, 0.012477, 0.000000, 0.083267, 0.150733, 0.284489, 0.000000]
iteration 10000
[0.271736, 0.000000, 0.142474, 0.054823, 0.012477, 0.000000, 0.083266, 0.150734, 0.284490, 0.000000]
iteration 100000
[0.271736, 0.000000, 0.142474, 0.054823, 0.012477, 0.000000, 0.083266, 0.150734, 0.284490, 0.000000]
eta=10--------------------------------------------------------
iteration 1
[0.100000, 0.100000, 0.100000, 0.100000, 0.100000, 0.100000, 0.100000, 0.100000, 0.100000, 0.100000]
iteration 10
[0.139452, 0.013191, 0.089558, 0.134699, 0.099862, 0.016938, 0.047250, 0.174526, 0.271009, 0.013515]
iteration 100
[0.267804, 0.000000, 0.142767, 0.053852, 0.014696, 0.000000, 0.083433, 0.150929, 0.286519, 0.000000]
iteration 1000
[0.271736, 0.000000, 0.142474, 0.054823, 0.012477, 0.000000, 0.083266, 0.150734, 0.284490, 0.000000]
iteration 10000
[0.271736, 0.000000, 0.142474, 0.054823, 0.012477, 0.000000, 0.083266, 0.150734, 0.284490, 0.000000]
iteration 100000
[0.271736, 0.000000, 0.142474, 0.054823, 0.012477, 0.000000, 0.083266, 0.150734, 0.284490, 0.000000]
```

