# OpenReview forum: "Last-Iterate Convergence of Smooth Regret Matching$^+$ Variants in Learning Nash Equilibria"
_ICLR.cc/2025/Conference — Submitted to ICLR 2025_

### Official Review · Reviewer_HPtD · 2024-10-27

**Soundness:** 3
**Presentation:** 2
**Contribution:** 3
**Rating:** 6
**Confidence:** 2

**Summary:**

This paper studies last-iterate convergence of Regret Matching (RM+) variants to Nash equilibria. It introduces a proof method leveraging the equivalence between RM+ and Online Mirror Descent to restore essential properties like monotonicity. Then, the authors show that smooth RM+ variants, including a new algorithm, achieve convergence under MVI condition. Experiments show that the proposed new algorithm perform better than other algorithms.

**Strengths:**

The structure of this paper is clear and easy to follow. The literature review is done thoroughly. The paper seems to introduce an innovative proof paradigm that effectively addresses gaps in achieving last-iterate convergence for RM+ variants. It establishes last-iterate convergence to games under a weak condition, making it applicable to a wider class of games.

**Weaknesses:**

Although the paper shows SOGRM+’s effectiveness, the experiments focus on two-player zero-sum matrix games. Additional tests across different game structures would strengthen its applicability claims.

**Questions:**

Why the proposed algorithm allows large \eta compared to other methods? Can you provide some intuitions behind this?

---

> ### Author Response · Authors · 2024-11-17
>
> We sincerely appreciate your thoughtful evaluation of our manuscript. Below, we outline our responses to the points you have raised.
>
> **Q1**: Although the paper shows SOGRM+’s effectiveness, the experiments focus on two-player zero-sum matrix games. Additional tests across different game structures would strengthen its applicability claims.
>
> **A**: We have included experimental results on three-player zero-sum polymatrix games, a classical type of game satisfying monotonicity [Pérolat et al., 2021], in our revisions (Appendix I, highlighted in red). Similar to the results on two-player zero-sum matrix games, SOGRM+ demonstrates reduced sensitivity to hyperparameters, consistently exhibiting strong convergence across all parameters.
>
>
>
>
>
>
>
> **Q2**: Why the proposed algorithm allows large $\eta$ compared to other methods? Can you provide some intuitions behind this?
>
> **A**: In response to the phenomenon you mentioned, we hypothesize that it arises because our proposed algorithm, SOGRM+, performs only a single prox-mapping operator per update step, unlike other smooth RM+ algorithms, which involve two prox-mapping operations at each iteration (the first occurrence of the prox-mapping  operator is in line 189). Then the  sequence of strategy profiles generated by SOGRM+ exhibits small differences.
>
> Specifically, the prox-mapping operator in smooth RM+ variants (such as SExRM+, SPRM+, and SOGRM+) involves a projection onto the simplex at sometimes [Farina et al., 2023] (not alwasy as in OMD algorithms), which may lead to significant changes in $\theta$ depending on the choice of $\eta$. In contrast, the update rule of SOGRM+ (in the second line) omits this prox-mapping operator and instead relies solely on simple addition and subtraction operations. As a result, the initial parameter $\theta_0$ may become negligible compared to the term $\eta F_i(\theta)$. Thus, the values of $\theta_i$ in SOGRM+ are likely to vary in direct proportion to $\eta$, and the resulting strategy $x_i = \frac{\theta_i}{\Vert \theta_i \Vert_1}$ will exhibit a more stable behavior with respect to changes in $\eta$. Therefore, for different values of $\eta$, the sequence of strategy profiles generated by SOGRM+ exhibits small differences. Moreover, when $\eta$ is small, Theorem 5.2 guarantees that the sequence of strategy profiles produced by SOGRM+ converges to the set of NE. Consequently, SOGRM+ permits the use of larger $\eta$ values compared to other algorithms.
>
> To validate our statement, as demonstrated in Section 6 Experiments, we conducted evaluations on 20 randomly generated 10-dimensional two-player zero-sum matrix games. Specifically, we analyzed the strategies of Player 0 output by SExRM+, SPRM+, and SOGRM+ at iterations 1, 10, 100, 1000, and 10,000. To mitigate randomness, we averaged the strategies across the 20 instances. The results clearly show that for different values of $\eta$, the sequence of strategies generated by SOGRM+ exhibits minimal variation. Notably, when $\eta \geq 1$ and the number of iterations $\geq 1000$, the strategies produced by SOGRM+ are nearly identical across different values of $\eta$. This behavior is not observed in the other two RM+ variants.

---

> ### Author Response · Authors · 2024-11-17
>
> **The sequence of strategies generated by SOGRM+** (The Sequence of strategies generated by SExRM+ and SPRM+ are in Appendix J of our revision, highlighted in red)
>
>
> SOGRM+ eta=0.01-----------------
>
> ```
> [iteration 1]          0.1000, 0.1000, 0.1000, 0.1000, 0.1000, 0.1000, 0.1000, 0.1000, 0.1000, 0.1000
> ```
>
> ```
> [iteration 10]         0.0986, 0.1001, 0.1087, 0.1058, 0.0970, 0.0920, 0.0948, 0.1015, 0.1064, 0.0951
> ```
>
> ```
> [iteration 100]        0.1154, 0.0486, 0.1729, 0.1593, 0.0791, 0.0183, 0.0638, 0.1170, 0.2033, 0.0224
> ```
>
> ```
> [iteration 1000]       0.2471, 0.0000, 0.1144, 0.0705, 0.0000, 0.0000, 0.0829, 0.1852, 0.2998, 0.0000
> ```
>
> ```
> [iteration 10000]      0.2700, 0.0000, 0.1446, 0.0496, 0.0247, 0.0000, 0.0808, 0.1456, 0.2848, 0.0000
> ```
>
> ```
> [iteration 100000]     0.2770, 0.0000, 0.1411, 0.0568, 0.0113, 0.0000, 0.0814, 0.1520, 0.2804, 0.0000
> ```
>
> SOGRM+ eta=0.1-----------------
>
> ```
> [iteration 1]          0.1000, 0.1000, 0.1000, 0.1000, 0.1000, 0.1000, 0.1000, 0.1000, 0.1000, 0.1000
> ```
>
> ```
> [iteration 10]         0.1097, 0.0548, 0.1686, 0.1550, 0.0826, 0.0233, 0.0675, 0.1175, 0.1948, 0.0263
> ```
>
> ```
> [iteration 100]        0.2441, 0.0000, 0.1215, 0.0658, 0.0000, 0.0000, 0.0896, 0.1809, 0.2981, 0.0000
> ```
>
> ```
> [iteration 1000]       0.2712, 0.0000, 0.1443, 0.0526, 0.0177, 0.0000, 0.0840, 0.1452, 0.2850, 0.0000
> ```
>
> ```
> [iteration 10000]      0.2718, 0.0000, 0.1425, 0.0548, 0.0124, 0.0000, 0.0833, 0.1507, 0.2845, 0.0000
> ```
>
> ```
> [iteration 100000]     0.2717, 0.0000, 0.1425, 0.0548, 0.0125, 0.0000, 0.0833, 0.1507, 0.2845, 0.0000
> ```
>
> SOGRM+ eta=1-----------------
>
> ```
> [iteration 1]          0.1000, 0.1000, 0.1000, 0.1000, 0.1000, 0.1000, 0.1000, 0.1000, 0.1000, 0.1000
> ```
>
> ```
> [iteration 10]         0.1436, 0.0000, 0.0000, 0.0194, 0.0813, 0.0595, 0.1651, 0.2740, 0.2463, 0.0109
> ```
>
> ```
> [iteration 100]        0.2729, 0.0000, 0.1409, 0.0551, 0.0118, 0.0000, 0.0793, 0.1525, 0.2874, 0.0000
> ```
>
> ```
> [iteration 1000]       0.2717, 0.0000, 0.1425, 0.0548, 0.0125, 0.0000, 0.0833, 0.1507, 0.2845, 0.0000
> ```
>
> ```
> [iteration 10000]      0.2717, 0.0000, 0.1425, 0.0548, 0.0125, 0.0000, 0.0833, 0.1507, 0.2845, 0.0000
> ```
>
> ```
> [iteration 100000]     0.2717, 0.0000, 0.1425, 0.0548, 0.0125, 0.0000, 0.0833, 0.1507, 0.2845, 0.0000
> ```
>
> SOGRM+ eta=10-----------------
>
> ```
> [iteration 1]          0.1000, 0.1000, 0.1000, 0.1000, 0.1000, 0.1000, 0.1000, 0.1000, 0.1000, 0.1000
> ```
>
> ```
> [iteration 10]         0.1395, 0.0132, 0.0896, 0.1347, 0.0999, 0.0169, 0.0472, 0.1745, 0.2710, 0.0135
> ```
>
> ```
> [iteration 100]        0.2678, 0.0000, 0.1428, 0.0539, 0.0147, 0.0000, 0.0834, 0.1509, 0.2865, 0.0000
> ```
>
> ```
> [iteration 1000]       0.2717, 0.0000, 0.1425, 0.0548, 0.0125, 0.0000, 0.0833, 0.1507, 0.2845, 0.0000
> ```
>
> ```
> [iteration 10000]      0.2717, 0.0000, 0.1425, 0.0548, 0.0125, 0.0000, 0.0833, 0.1507, 0.2845, 0.0000
> ```
>
> ```
> [iteration 100000]     0.2717, 0.0000, 0.1425, 0.0548, 0.0125, 0.0000, 0.0833, 0.1507, 0.2845, 0.0000
> ```

---

### Official Review · Reviewer_kPj9 · 2024-11-04

**Soundness:** 3
**Presentation:** 3
**Contribution:** 3
**Rating:** 6
**Confidence:** 4

**Summary:**

This paper studies the last-iterate convergence properties of smooth variants of the Regret Matching+ (RM+) algorithm. RM+ type algorithms differ from Extragradient (EG)/Optimistic Gradient (OG) algorithms since the feedback is not the loss function and can be non-monotone. The paper's main contribution is a new proof paradigm that addresses this challenge: (1) they show an equivalence between RM+ and Online Mirror Descent (OMD) with Euclidean regularizer; (2) using the equivalence, they relate the tangent residual to the distance between consecutive accumulated regrets, which can be easily bounded. By this proof paradigm and existing results for OG, they propose SOGRM+ and show asymptotic last-iterate convergence in the duality gap for games with weak MVI. They also show SPRM+ and SEXRM+ have asymptotic last-iterate convergence in the duality gap for monotone games. They also conduct numerical experiments on small random games to demonstrate the effectiveness of their algorithms.

**Strengths:**

1. This paper studies the last-iterate convergence properties of RM-type algorithms in games. This question has been studied only recently, and these first results only focus on zero-sum games. This paper extends previous results and presents asymptotic last-iterate convergence in the duality gap in monotone games and games that satisfy weak MVI.
2. This paper's proof paradigm incorporates several ideas in learning and games and is novel. It uses the existing idea of equivalence between RM+ and OMD to show that the operator's non-monotonicity can be addressed by rewriting the update rule in the OMD style. Then, by elegantly using the tangent residual, they show that the tangent residual can be bounded by the distance between consecutive accumulative regret vectors. This distance can be further bounded by extending existing results for OMD-type algorithms. This analysis is simple but powerful.

**Weaknesses:**

1. The paper's presentation could be improved. For the experimental results, for example, whether the algorithm is alternating or simultaneous should be provided. The OMD algorithm given in the paper seems non-standard: a standard OMD update (prox-mapping) does not contain a $ qt$ term, and the Bregman divergence is fixed with one generating function $q$. This should be explained in detail to reduce confusion.
2. The meaning of asymptotic last-iterate convergence is confusing in this paper. Since the proof paradigm uses the tangent residual as a proxy, it gives asymptotic last-iterate convergence in the duality gap (i.e., $\lim_{t\rightarrow \infty} Gap(x^t) = 0$). But it does not give the stronger notion of point convergence of the iterates to a Nash equilibrium (i.e., $\lim_{t\rightarrow \infty} = x^*$ is a Nash equilibrium), which is proved in [1] for SExRM+ and SPRM+ in zero-sum games. This difference should be clarified in the text and acknowledged in Table 1.

[1] Yang Cai, Gabriele Farina, Julien Grand-Clément, Christian Kroer, Chung-Wei Lee, Haipeng Luo, and Weiqiang Zheng. Last-iterate convergence properties of regret-matching algorithms in games, 2023.

**Questions:**

1. It has been shown that restarting SExRM+ and SPRM+ has a game-dependent linear last-iterate convergence in zero-sum games. Do you think similar results hold for the proposed algorithm SOGRM+ and (non-restarting version) of SPRM+ and SExRM+?
2. Is it possible to prove point convergence of the iterates to a Nash equilibrium of these algorithms in monotone games?

---

> ### Author Response · Authors · 2024-11-17
>
> Thank you for your thoughtful feedback on our paper. In the following, we address the questions and concerns you have highlighted.
>
> **Q1**: The meaning of asymptotic last-iterate convergence is confusing in this paper. Since the proof paradigm uses the tangent residual as a proxy, it gives asymptotic last-iterate convergence in the duality gap (i.e., limt→∞Gap(xt)=0). But it does not give the stronger notion of point convergence of the iterates to a Nash equilibrium (i.e., limt→∞=x∗ is a Nash equilibrium), which is proved in [1] for SExRM+ and SPRM+ in zero-sum games. This difference should be clarified in the text and acknowledged in Table 1.
>
> **A**: We received similar comments before the ICLR submission deadline. We apologize for not making the changes based on those comments before the ICLR submission deadline. The reason is that we did not fully understand the distinction between "point convergence of the iterates" and "convergence of the tangent residual." To avoid introducing potential errors, we refrained from making modifications.
>
> Fortunately, we have come across the ICLR 2025 version of [Cai et al., 2023], which provides a detailed explanation of both "point convergence of the iterates" and "convergence of the tangent residual" (this version became available online only after we submitted our paper to ICLR 2025). Specifically, "convergence of the tangent residual" guarantees convergence to the set of NE, but the sequence of the strategy profiles does not converge to one point. It is possible for this sequence to cycle around the
> set of NE without ever converging to a single point. On the other hand, "point convergence of the iterates" not only guarantees convergence to the set of NE but also ensures convergence to a specific point within that set.
>
> We have revised our paper accordingly and acknowledge that the last-iterate convergence of SExRM+ and SPRM+ provided by [Cai et al., 2023] is a stronger convergence concept than the last-iterate convergence presented in our paper. The specific changes can be found in lines 58-60 and lines 168-180, which are highlighted in red. Notably, in the main text, we utilize "last-iterate convergence of the iterates" rather than "point convergence of the iterates" since [Cai et al., 2023] employ "last-iterate convergence of the iterates" in their paper, and we follow their methodology.
>
> We greatly appreciate your feedback, as it significantly enhances the soundness of our paper.
>
> **Q2**: whether the algorithm is alternating or simultaneous?
>
> **A**: We employ simultaneous updates because to the best of our knowledge, the theoretical analysis of existing work on last-iterate convergence is based on simultaneous updates and no work has provided a last-iterate convergence guarantee for algorithms that use alternating updates. This clarification is provided in our revision (lines 495-497, highlighted in red).
>
> **Q3**: The OMD algorithm given in the paper seems non-standard.
>
> **A**: We utilize the OMD algorithm defined in [Joulani et al., 2017] and [Liu et al., 2021], which is a more generalized version of the standard OMD algorithm and encompasses the standard OMD as a special case. This clarification has been incorporated in the updated version of our manuscript (lines 194-197, highlighted in red).
>
> **Q4**: It has been shown that restarting SExRM+ and SPRM+ has a game-dependent linear last-iterate convergence in zero-sum games. Do you think similar results hold for the proposed algorithm SOGRM+ and (non-restarting version) of SPRM+ and SExRM+.
>
> **A**: We think these results hold because we observed linear last-iterate convergence rates of SOGRM+, SPRM+, and SExRM+ in two-player zero-sum games during our experiments. We will consider proving this as one of our future research directions.

---

> ### Author Response · Authors · 2024-11-17
>
> **Q5**: Is it possible to prove point convergence of the iterates to a Nash equilibrium of these algorithms in monotone games?
>
> **A**: For SExRM+ and SPRM+, we think this is feasible because, as shown in Eq. (22) and (30), we can bound $\theta^t$ and $\theta^{t+1/2}$ within a bounded space (also mentioned in line 286 of the ICLR 2025 version of [Cai et al., 2023]). Subsequently, we can apply traditional analysis from OMD (Section 6.1 of [Lee et al., 2021]) to establish the point convergence of the iterates.
>
> Using SExRM+ as an example, we assume there exists a convergent subsequence $\\{ \theta^{\tau+1/2}|\tau \in \kappa \\}$, where $\theta^{\tau+1/2}$ converges to a limit point $\hat{\theta}^{\infty}$. As stated in the ICLR 2025 version of [Cai et al., 2023] (line 904-907), such subsequence must be exsit and $\hat{x}^{\infty} = \\{ \hat{x}^{\infty}_i \mid \hat{x}^{\infty}_i = \hat{\theta}^{\infty}_i / \Vert \hat{\theta}^{\infty}_i \Vert_1 \\}$ is an NE. From Appendix B.1 of our paper, we have
>
> $$
> D\_{\psi}(\hat{\theta}^{\infty}_i, \theta^{t+1}_i) \leq D\_{\psi}(\hat{\theta}^{\infty}_i, \theta^{t}_i) - \eta \langle F_i(\theta^{t+1/2}), \hat{\theta}^{\infty}_i \rangle - D\_{\psi}(\theta^{t+1}_i, \theta^{t+1/2}_i) - D\_{\psi}(\theta^{t+1/2}_i, \theta^{t}_i) + \eta D L_u \Vert \theta^{t+1/2} - \theta^{t} \Vert_2 \Vert \theta^{t+1} - \theta^{t+1/2} \Vert_2
> $$
>
> For the term $\langle F_i(\theta^{t+1/2}), \hat{\theta}^{\infty}_i \rangle$, we can express it as:
>
> $$
> \langle F_i(\theta^{t+1/2}), \hat{\theta}^{\infty}_i \rangle = \Vert \hat{\theta}^{\infty}_i \Vert_1 \langle \ell^{t+1/2}_i, x^{t+1/2}_i - \hat{x}^{\infty}_i \rangle
> $$
>
> Subsequently, we obtain the following inequality:
>
> $$
> D\_{\psi}(\hat{\theta}^{\infty}_i, \theta^{t+1}_i) \leq D\_{\psi}(\hat{\theta}^{\infty}_i, \theta^{t}_i) - \eta \Vert \hat{\theta}^{\infty}_i \Vert_1 \langle \ell^{t+1/2}_i, x^{t+1/2}_i - \hat{x}^{\infty}_i \rangle - D\_{\psi}(\theta^{t+1}_i, \theta^{t+1/2}_i) - D\_{\psi}(\theta^{t+1/2}_i, \theta^{t}_i) + \eta D L_u \Vert \theta^{t+1/2} - \theta^{t} \Vert_2 \Vert \theta^{t+1} - \theta^{t+1/2} \Vert_2
> $$
>
> Dividing both sides by $\Vert \hat{\theta}^{\infty}_i \Vert_1$:
>
> $$
> \frac{D\_{\psi}(\hat{\theta}^{\infty}_i, \theta^{t+1}_i)}{\Vert \hat{\theta}^{\infty}_i \Vert_1} \leq \frac{D\_{\psi}(\hat{\theta}^{\infty}_i, \theta^{t}_i)}{\Vert \hat{\theta}^{\infty}_i \Vert_1} - \eta \langle \ell^{t+1/2}_i, x^{t+1/2}_i - \hat{x}^{\infty}_i \rangle - \frac{D\_{\psi}(\theta^{t+1}_i, \theta^{t+1/2}_i) + D\_{\psi}(\theta^{t+1/2}_i, \theta^{t}_i)}{\Vert \hat{\theta}^{\infty}_i \Vert_1} + \frac{\eta D L_u}{\Vert \hat{\theta}^{\infty}_i \Vert_1} \Vert \theta^{t+1/2} - \theta^{t} \Vert_2 \Vert \theta^{t+1} - \theta^{t+1/2} \Vert_2
> $$
>
> Since both $\theta^t$ and $\theta^{t+1/2}$ are constrained within a bounded space, it follows that $\Vert \hat{\theta}^{\infty}_i \Vert_1$ must have both an upper and a lower bound (larger than 1 since $\theta \in R^{|A_i|}\_{\geq 1}$). Therefore, there must exist a value for $\eta$ such that the following inequality holds:
>
> $$
> \sum\_{i} \frac{D\_{\psi}(\hat{\theta}^{\infty}i, \theta^{t+1}_i)}{\Vert \hat{\theta}^{\infty}_i \Vert_1} \leq \sum\_{i} \frac{D\_{\psi}(\hat{\theta}^{\infty}_i, \theta^{t}_i)}{\Vert \hat{\theta}^{\infty}_i \Vert_1} - C(D\_{\psi}(\theta^{t+1}, \theta^{t+1/2}) + D\_{\psi}(\theta^{t+1/2}, \theta^{t})), \tag{1}
> $$
>
> where C is a game-dependent constant. Combining this with Theorem 4.2, where $\Vert \theta^{t+1/2} - \theta^{t} \Vert_2 \to 0$ and $\Vert \theta^{t+1} - \theta^{t+1/2} \Vert_2 \to 0$ as $t \to \infty$, we conclude that the point convergence of the iterates. It is important to note that the value of $\eta$ may need to be reduced further compared to the value specified in Theorem 4.1 of our paper.
>
> This idea is conceived during the rebuttal process, inspired by your reference to the "point convergence of the iterates." We would greatly appreciate any feedback you may have and would be delighted to discuss this idea further.
>
> Regarding SOGRM+, we sincerely regret our inability to prove its point convergence of the iterates, as we were unable to derive an inequality analogous to Eq. (1) for SOGRM+. If you have any suggestions or insights, we would be more than happy to discuss SOGRM+'s  point convergence of the iterates with you.
>
>
>
> Additional References:
>
> [1] Joulani, Pooria, András György, and Csaba Szepesvári. "A modular analysis of adaptive (non-) convex optimization: Optimism, composite objectives, and variational bounds." In *International Conference on Algorithmic Learning Theory*, pp. 681-720. PMLR, 2017.
>
> [2] Lee, Chung-Wei, Christian Kroer, and Haipeng Luo. "Last-iterate convergence in extensive-form games." *Advances in Neural Information Processing Systems* 34 (2021): 14293-14305.

---

> > ### Comment · Reviewer_kPj9 · 2024-11-24
> >
> > I thank the authors for their responses. My concerns about the presentation have been resolved, and I appreciate the point convergence results provided in the response. I have increased my score accordingly.
> >
> > A typo I noticed when reading the paper:
> > Line 1230-1231, the last $\ell^{t-1/2}_i$ in the equation should be $\ell^{t+1/2}_i$.

---

> > > ### Author Response · Authors · 2024-11-25
> > >
> > > Thank you for acknowledging our revisions and responses. We have corrected the typo (highlighted in red) based on your suggestion.

---

### Official Review · Reviewer_oJYd · 2024-11-04

**Soundness:** 3
**Presentation:** 3
**Contribution:** 3
**Rating:** 6
**Confidence:** 3

**Summary:**

The authors study last-iterate convergence  for smooth variants of Regret Matching+ introduced by [Farina et al.,2023], Smooth Extragradient RM+ (SExRM+) and Smooth Predictive RM+(SPRM+) in smooth games. They show asymptotic last-iterate convergence. They also show ($1/\sqrt{t})$ best-iterate convergence.

References:
Farina, Gabriele, et al. "Regret matching+:(in) stability and fast convergence in games." Advances in Neural Information Processing Systems 36 (2024).

**Strengths:**

1) Their main techniques capitalize and build on the equivalence to Optimistic Mirror Dsecent and then using this they are able to recover monotonocity and then by measuring the distance to the NE via the tangent residual [Cai et al., 2022] and show that this is the distance between accumulated regrets.

2) The established equivalence to OMD is definitely useful.

3) Using this equivalence they propose a new algorithm Smooth Optimistic Gradient Regret Matching+ (SOGRM+), which has last-iterate convergence guarantees and seems to perform well in their conducted experiments, better than SExRM+ and SPRM+, which were proposed in [Farina et al., 2023].

**Weaknesses:**

1) Their paper is restricted to the class of smooth games. It would be great to get a clarity of how far these techniques can be stretched without the smoothness assumption?

2) Is it possible to obtain asymptotic rates of convergence for the last-iterate analysis shown here?

3) Could the authors also compare their algorithm SOGRM+ on games such as Kuhn Poker and Goofspiel etc, to get a better idea of the performance of their proposed algorithm, aside from the random zero-sum games?

Minor comment:
To make the paper self-contained, please describe the algorithms being analyzed as they are quite recent developments.

**Questions:**

See weaknesses.

---

> ### Author Response · Authors · 2024-11-17
>
> We greatly appreciate your recognition of our work. We now provide responses to the issues you have raised.
>
> **Q1**: Their paper is restricted to the class of smooth games. It would be great to get a clarity of how far these techniques can be stretched without the smoothness assumption?
>
> **A**: We agree with your point that extending our proof paradigm beyond smooth games is an interesting direction for future research. However, in non-smooth games, establishing the relationship between loss and strategy proflie is considerably more challenging than in smooth games. This is one of the key issues we aim to address in our future work. Without the smoothness assumption, we cannot prove the last-iterate convergence by the current approach.
>
> **Q2**: Is it possible to obtain asymptotic rates of convergence for the last-iterate analysis shown here?
>
> **A**: We apologize for that we are not entirely sure we understand the question. We assume you are referring to "non-asymptotic rates." Indeed, we have investigated the non-asymptotic rates of RM+ variants. Unfortunately, we were unable to derive such rates due to the inherent challenge that RM+ type algorithms update $\theta$ rather than $x$, yet the final output is $x$. This mismatch between the update mechanism and the output makes the algorithm analysis exceptionally difficult. For instance, as mentioned in our paper, even in analyzing the convergence of SOGRM+, we have to "use the definition of the inner product to apply the weak MVI and tangent residual, rather than directly transforming variables using equalities as in OG."
>
> **Q3**: Could the authors also compare their algorithm SOGRM+ on games such as Kuhn Poker and Goofspiel etc, to get a better idea of the performance of their proposed algorithm, aside from the random zero-sum games?
>
> **A**: The results on Kuhn Poker and Goofspiel have already been included in our revision (Appendix I, highlighted in red). The game payoff matrices of Kuhn Poker and Goofspiel are from the open-source code provided in the ICLR 2025 version of [Cai et al., 2023]. Consistent with the results presented in Section 6 (Experiments), SOGRM+ demonstrates lower sensitivity to hyperparameters compared to other algorithms, showing strong convergence performance across all tested parameters.
>
> **Q4**: To make the paper self-contained, please describe the algorithms being analyzed as they are quite recent developments.
>
> **A**: In our original submission, we introduced the key ideas and update rules of the relevant algorithms, such as RM+, SExRM+, and SPRM+, in Section 2 (Preliminaries). For SOGRM+, its key ideas and update rules can be found in the second line of Section 5 (Our Algorithm: SOGRM+).
>
> In our revision, we have added an introduction to the Blackwell approachability framework, upon which the RM+ variants are based (lines 194-197, highlighted in red). This addition aims to help you better understand the RM+ variants. Note that vanilla RM+ was initially proposed by [Bowling et al., 2015], and later incorporated into the Blackwell approachability framework by [Farina et al., 2021]. Subsequent works have built upon the Blackwell approachability framework to improve RM+ and theoretically analyze the convergence of RM+ variants [Farina et al., 2023; Meng et al., 2023; Cai et al., 2023]. Additionally, in our revision, we have provided the pseudocode for RM+, SExRM+, SPRM+, and SOGRM+ (Appendix H, highlighted in red).

---

> > ### Comment · Reviewer_oJYd · 2024-11-26
> > **Additional Clarifications**
> >
> > I thank the authors for their efforts to respond to the comments and I am generally satisfied with my responses. I looked at the additional experiments for SOGRM+. The performance of SExRM+, ExRM+ and PRM+ seem to match or is better than SOGRM+, for Kuhn poker and Goofspiel (at least for some hyperparameters).
> >
> > While I understand these are just specific examples, maybe it helps in understanding what is the motivation for their algorithm SOGRM+, when algorithms such as SExRM+ and SPRM+ may exhibit similar performance? Have authors identified certain regimes or specific cases when SOGRM+ is better or natural to use compared to the existing algorithms? Or for instance, it would be perhaps good to highlight if it has better theoretical guarantees etc? This will really help place their contributions in this sea of algorithms. Note that I do recognize the main contribution, which is establishing last-iterate convergence for RM+ in smooth games.

---

> ### Author Response · Authors · 2024-11-26
>
> Thank you for recognizing the contributions of our work. Regarding your new questions, we provide the following clarifications:
>
> 1. As demonstrated in our theoretical analysis, SExRM+ and SPRM+ achieve last-iterate convergence only in games with monotonicity (e.g., two-player zero-sum matrix games and convex-concave games) (Theorem 4.1). In contrast, SOGRM+ exhibits last-iterate convergence in games satisfying the weak MVI, which is a broader set of games that includes those with monotonicity (Theorem 5.1). Based on existing techniques, we cannot provide a proof that SExRM and SOGRM+ achieve last-iterate convergence in games satisfying the weak MVI.
> 2. Our experimental results further indicate that SOGRM+ outperforms both SExRM+ and SPRM+ in certain games with monotonicity, such as randomly generated two-player zero-sum matrix games (Figure 1) and randomly generated three-player zero-sum polymatrix games (Figure 3).
>
> In summary, SOGRM+ is an effective algorithm that provides stronger theoretical guarantees than both SExRM+ and SPRM+. Moreover, it demonstrates superior empirical performance in certain games compared to SExRM+ and SPRM+.

---

### Official Review · Reviewer_AL6a · 2024-11-05

**Soundness:** 3
**Presentation:** 2
**Contribution:** 3
**Rating:** 6
**Confidence:** 2

**Summary:**

This paper provides a proof to show the last-iterate convergence of SExRM+ and SPRM+ algorithms, which are smoothed variants of RM+ in general monotone games, and proposes a new algorithm SOGRM+ that further has last-iterate convergence in games with Weak MVI.

**Strengths:**

- The result in this paper generalizes those in previous works by Meng et al. (2023) and Cai et al. (2024). The new technique in their analysis might be interesting for future work.

**Weaknesses:**

- In terms of presentation, I think more space should be used to explain the high-level ideas and comparison with previous work, especially Cai et al. (2023).
- I do not understand the strong claim in the technical novelty section saying that "Neither of these techniques can be used alone to prove the last-iterate convergence of RM+ variants."  For example, there remains the possibility to prove similar results without going through tangent residual argument.
- The OMD update in this paper seems non-standard due to the f_i(x_i) term in Eq.(5).  Calling it OMD might be misleading.

**Questions:**

See the weakness section.

---

> ### Author Response · Authors · 2024-11-17
>
> We sincerely appreciate your positive feedback on our manuscript. Below, we address the points you raised.
>
> **Q1**: In terms of presentation, I think more space should be used to explain the high-level ideas and comparison with previous work, especially Cai et al. (2023).
>
> **High-Level Ideas**
>
> In the Paragraph Contributions of Secion 1 Introduction (lines 88–96), we provide a detailed description of the high-level ideas behind our proof paradigm. Our key insight is to first recover monotonicity or the weak MVI by leveraging the strong equivalence between RM$^+$ and OMD. Specifically, in the OMD equivalents of RM$^+$ variants, the feedback corresponds to the loss gradient of the vanilla games, which inherently satisfies monotonicity or the weak MVI. Then, our proof paradigm  measures the distance of RM$^+$ variants to NE via the tangent residual, which in turn demonstrates that this distance is related to the distance between accumulated regrets, thereby proving last-iterate convergence. This is because the distance between accumulated regrets can be shown to converge to zero.
>
> **Comparison with Previous Work**
>
> In the Paragraph Discussions of the Secion 1 Introduction (lines 105–119), we compare our theoretical results with those of previous works.
>
> - Our proofs significantly diverge from those of previous works, which either analyze the dynamics of limit points [Cai et al., 2023] or rely on strong monotonicity [Meng et al., 2023].
> - The last-iterate convergence results in [Cai et al., 2023] and [Meng et al., 2023] cannot be extended to games satisfying monotonicity (let alone weak MVI). The reason is that their results require additional assumptions beyond monotonicity, such as the existence of a strict NE (the results of [Cai et al., 2023] regarding RM+), the interchangeability of NEs (the results of [Cai et al., 2023] regarding SExRM+ & SPRM+), Saddle-Point Metric Subregularity (the results of [Cai et al., 2023] regarding RS-SExRM+ & RS-SPRM+), or even strong monotonicity (the results of [Meng et al., 2023] regarding RM+).
> - Our proof paradigm allows the application of existing last-iterate convergence results for OMD-based algorithms to RM$^+$ variants. In contrast, the proof in [Cai et al., 2023] cannot achieve this because their motivation assumes that the feedback in RM$^+$ variants only satisfies MVI (which is weaker than monotonicity but stronger than weak MVI, as defined in Section 2.1), even when the loss gradient of vanilla games satisfies monotonicity.
> - [Cai et al., 2023] must employ a different approach to prove best-iterate convergence, whereas we utilize the same proof paradigm. Additionally, [Meng et al., 2023] do not investigate best-iterate convergence.
> - The best-iterate convergence results in [Cai et al., 2023] are limited to two-player zero-sum matrix games, as their results depend on the definition of the duality gap in these games. In contrast, our best-iterate convergence results are valid for all games satisfying monotonicity, or even just weak MVI.

---

> > ### Author Response · Authors · 2024-11-17
> >
> > **Q2**: I do not understand the strong claim in the technical novelty section saying that "Neither of these techniques can be used alone to prove the last-iterate convergence of RM+ variants." For example, there remains the possibility to prove similar results without going through tangent residual argument.
> >
> > **A**: We apologize for the lack of clarity in our previous statement. What we intended to convey is the following: to the best of our knowledge, prior to our submission to ICLR 2025, neither of these techniques had been used independently to establish the last-iterate convergence of RM+ variants. While it is possible that alternative methods exist to prove the last-iterate convergence of RM+ variants without utilizing the tangent residual, no prior work, as of our submission to ICLR 2025, had used either of these techniques in isolation to provide such a convergence guarantee for RM+ variants. We have revised this statement in our new manuscript (lines 123-124, highlighted in red).
> >
> > **Q3**: The OMD update in this paper seems non-standard due to the f_i(x_i) term in Eq.(5). Calling it OMD might be misleading.
> >
> > **A**: We utilize the OMD algorithm defined in [Joulani et al., 2017] and [Liu et al., 2021], which is a more generalized version of the standard OMD algorithm and encompasses the standard OMD as a special case. This clarification has been incorporated in the updated version of our manuscript (lines 194-197, highlighted in red).
> >
> >
> >
> > Additional References:
> >
> > [1] Joulani, Pooria, András György, and Csaba Szepesvári. "A modular analysis of adaptive (non-) convex optimization: Optimism, composite objectives, and variational bounds." In *International Conference on Algorithmic Learning Theory*, pp. 681-720. PMLR, 2017.

---

### Meta-Review · Area_Chair_xYs1 · 2024-12-20

**Metareview:**

The paper focuses on the last iterate convergence of variants of RM+, going beyond the two-player zero-sum setting (i.e., settings where weak MVI holds). The authors effectively use a recent connection between RM+ and online MD (which was established in a paper by Farina et al), and techniques from last iterate convergence results of Extra Gradient and Optimistic GD based on tangent residuals (appeared in Cai et al) to show last-iterate convergence of two existing smooth RM variants. The reviewers and the AC believe that the paper is interesting, it has merits, but it is a borderline paper. Due to the fact it seems the techniques are not novel, i.e., are heavily based on the two aforementioned papers and the results are not enough (not surprising or so important) to make the paper pass the bar, we recommend rejection.

**Additional Comments On Reviewer Discussion:**

The rebuttal did not change the reviewers' or AC's opinion.

---

### Decision · Program_Chairs · 2025-01-22

Reject